# BATCHED STOCHASTIC MATCHING BANDITS

## ABSTRACT

In this study, we introduce a novel bandit framework for stochastic matching based on the Multi-nomial Logit (MNL) choice model. In our setting, $N$ agents on one side are assigned to $K$ arms on the other side, where each arm stochastically selects an agent from its assigned pool according to an unknown preference and yields a corresponding reward. The objective is to minimize regret by maximizing the cumulative revenue from successful matches across all agents. This task requires solving a combinatorial optimization problem based on estimated preferences, which is NP-hard and leads a naive approach to incur a computational cost of $O(K^N)$ per round. To address this challenge, we propose batched algorithms that limit the frequency of matching updates, thereby reducing the amortized computational cost—i.e., the average cost per round—to $O(1)$ while still achieving a regret bound of $\tilde{O}(\sqrt{T})$.

## 1 INTRODUCTION

In recent years, the rapid growth of matching markets—such as ride-hailing platforms, online job boards, and labor marketplaces—has underscored the importance of *maximizing revenue* from successful matches. For example, in ride-hailing services, the platform seeks to match riders (agents) with drivers (arms) in a way that maximizes total revenue generated from completed rides.

This demand has led to extensive research on online bipartite matching problems (Karp et al., 1990; Mehta et al., 2007; 2013; Gamlath et al., 2019; Fuchs et al., 2005; Kesselheim et al., 2013), where two sets of vertices are considered and one side is revealed sequentially. These studies primarily focus on maximizing the number of matches. However, a significant gap remains between these theoretical models and practical scenarios for maximizing revenue under latent reward functions. Specifically, these models generally assume one-to-one assignments under deterministic matching and focus solely on match count, without incorporating *learning mechanisms* that adapt to observed reward feedback or aim to maximize cumulative revenue.

More recently, the concept of *matching bandits* has emerged to better capture online learning dynamics in matching markets (Liu et al., 2020; 2021; Sankararaman et al., 2020; Basu et al., 2021; Zhang et al., 2022; Kong & Li, 2023). In this framework, agents are assigned to arms in each round, and arms select one agent to match, generating stochastic reward feedback. The goal is typically to learn reward distributions to eventually identify stable matchings (McVitie & Wilson, 1971).

Despite introducing online learning, existing matching bandit models rely on structural assumptions that restrict their practical applicability. Specifically, prior work generally assumes that arms select agents *deterministically* according to known or fixed preference orders, resulting in what we refer to as deterministic matching. However, in many real-world settings—such as ride-hailing services and online freelancer marketplace—arms often make *stochastic* choices reflecting unknown or latent preferences. For example, when a dispatch system offers a driver multiple rider requests, the driver may select among them probabilistically, reflecting personal preferences, rather than following a fixed or deterministic rule.

In this work, we propose a novel and practical online matching framework, termed *stochastic matching bandits* (SMB), designed to model such stochastic choice behavior under unknown preferences. SMB permits *multiple agents* to be simultaneously assigned to the same arm, with the arm stochastically selecting one agent from the assigned pool. This formulation departs from both traditional online matching and prior matching bandit frameworks by explicitly modeling *probabilistic arm behavior*, thereby addressing a different yet practically motivated objective.

While our framework captures important aspects of real-world matching systems that are not fully addressed by prior models, it represents a different modeling perspective rather than a direct replacement for existing approaches. Specifically, our work focuses on a practically significant setting where the goal is to *learn to maximize revenue* under *stochastic arm behavior with unknown preferences*. By explicitly modeling stochastic choice dynamics and allowing multiple simultaneous proposals, our framework expands the scope of matching bandit research toward more realistic and revenue-driven applications.

However, realizing this goal comes with substantial computational challenges: determining the optimal assignment in each round requires solving a combinatorial optimization problem that is NP-hard, making naive implementations impractical in large-scale systems. This raises the following fundamental question:

*Can we maximize revenue in stochastic matching bandits*
*while ensuring (amortized) computational efficiency?*

To address this challenge, we propose *batched* algorithms for the SMB framework that strategically limit the frequency of matching assignment updates. These algorithms achieve no-regret performance while substantially reducing the amortized computational cost—that is, the average computation required per round. Below, we summarize our main contributions.

**Summary of Our Contributions.**

- We introduce a novel and practical framework of stochastic matching bandits (SMB), which incorporates the stochastic behavior of arms under latent preferences. However, naive approaches suffer from significant computational overhead, incurring an amortized cost of $O(K^N)$ per round, where $N$ agents are matched to $K$ arms.

- Under SMB, we first develop a batched algorithm that balances exploration and exploitation with limited matching updates. Assuming knowledge of a non-linearity parameter $\kappa$, the algorithm achieves $\widetilde{\mathcal{O}}(\sqrt{T})$ regret using only minimal matching updates of $\Theta(\log \log T)$—and thus $\mathcal{O}(1)$ amortized computational cost for a large enough $T$.

- We further propose our second algorithm to eliminate the requirement of knowing $\kappa$, retaining the same $\widetilde{\mathcal{O}}(\sqrt{T})$ regret still with only $\Theta(\log \log T)$ updates and low amortized computational cost of $\mathcal{O}(1)$.

- Finally, through empirical evaluations, we demonstrate that our algorithms achieve improved or comparable regret while significantly reducing computational cost compared to existing methods, highlighting their practical effectiveness.

## 2 RELATED WORK

**Matching Bandits.** We review the literature on matching bandits, which studies regret minimization in matching markets. This line of work was initiated by Liu et al. (2020) and extended by Sankararaman et al. (2020); Liu et al. (2021); Basu et al. (2021); Zhang et al. (2022); Kong & Li (2023), focusing on finding optimal stable matchings through stochastic reward feedback. However, these studies are largely limited to the standard multi-armed bandit setting, without considering feature-based preferences or structural generalizations. Moreover, they universally assume that the number of agents does not exceed the number of arms ($N \leq K$).

Our proposed *Stochastic Matching Bandits (SMB)* framework departs from this literature in several key ways. First, while prior work assumes that arms select agents *deterministically* based on known preferences, SMB models arms as making *stochastic* choices based on unknown, latent preferences that must be learned over time. This shifts the objective from identifying a stable matching to maximizing cumulative reward through adaptive learning. Second, SMB captures richer preference structures by modeling utilities as functions of agent-side features. Third, it removes structural restrictions on the market size, allowing both $N \leq K$ and $N \geq K$ scenarios. While SMB represents, in principle, a distinct modeling perspective, these advances make SMB applicable to a broader range of real-world systems, such as ride-hailing and online marketplaces, where preferences are stochastic, feature-driven, and market sizes vary across applications.

**MNL-Bandits.** In our study, we adopt the Multi-nomial Logit (MNL) model for arms' choice preferences in matching bandits. As the first MNL bandit method, Agrawal et al. (2017a) proposed

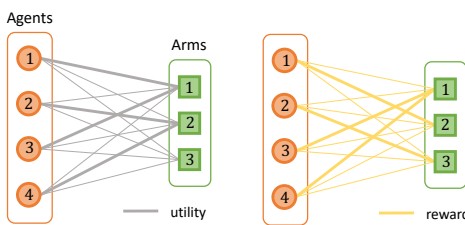 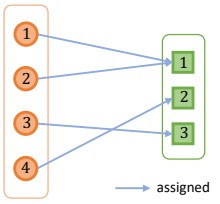 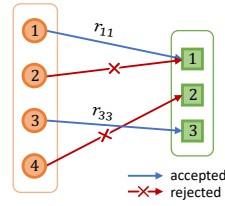

(a) There are multiple agents and arms, each pair associated with an unknown utility reflecting their preference, along with different rewards across agent-arm pairs.

(b) Each agent is assigned to an arm by a policy.

(c) Each arm stochastically accepts at most one from assignments and receives the reward.

Figure 1: Illustration of our stochastic matching process with 4 agents ($N = 4$) and 3 arms ($K = 3$).

an epoch-based algorithm, followed by subsequent contributions from Agrawal et al. (2017b); Chen et al. (2023); Oh & Iyengar (2019; 2021); Lee & Oh (2024). However, unlike selecting an assortment at each time step, our novel framework for the stochastic matching market mandates choosing at most $K$ distinct assortments to assign agents to each arm. Consequently, handling $K$-multiple MNLs simultaneously results in exponential computational complexity. More recently, Kim & Oh (2024) studied MNL-based preferences in matching bandits; however, their focus was on system stability under binary $(0/1)$ rewards, rather than revenue maximization. Additionally, their work did not address the computational intractability of exact combinatorial optimization in this context.

**Batch learning in Bandits.** Batch learning in bandit problems has been explored in the context of multi-armed bandits (MAB) (Perchet et al., 2015; Gao et al., 2019) and later extended to (generalized) linear bandit models (Ruan et al., 2021; Hanna et al., 2023; Han et al., 2020; Ren & Zhou, 2024; Sawarni et al., 2024; Ren et al., 2024). Also, a concurrent work of Midigeshi et al. (2025) study the multinomial logistic model with batched updates, but their setting is fundamentally different from other relevant works in the MNL bandit literature (Oh & Iyengar, 2019; 2021; Agrawal et al., 2017a;b). In their framework, the agent selects a single item (i.e., one arm), so that the learner does not selects a combinatorial set of arms.

To the best of our knowledge, batch-limited updates have not yet been explored in the context of matching bandits with a combinatorial set of arms.

## 3 PROBLEM STATEMENT

We study stochastic matching bandits (SMB) with $N$ agents and $K$ arms. For better intuition, the overall setup is illustrated in Figure 1. The detailed formulation is as follows: For each agent $n \in [N]$, feature information is known as $x_n \in \mathbb{R}^d$, and each arm $k \in [K]$ is characterized by latent vector $\theta_k \in \mathbb{R}^d$. We define the set of features as $X = [x_1, \ldots, x_N] \in \mathbb{R}^{d \times N}$ and the rank of $X$ as $rank(X) = r(\leq d)$. At each time $t \in [T]$, every agent $n$ may be assigned to an arm $k_{n,t} \in [K]$. Let assortment $S_{k,t} = \{n \in [N] : k_{n,t} = k\}$, which is the set of agents that are assigned to an arm $k$ at time $t$. Then given an assortment to each arm $k$ at time $t$, $S_{k,t}$, each arm $k$ randomly accepts an agent $n \in S_{k,t}$ and receives reward $r_{n,k} \in [0, 1]$ according to the arm's preference specified as follows. The probability for arm $k$ to accept agent $n \in S_{k,t}$ follows Multi-nomial Logit (MNL) model (Agrawal et al., 2017a;b; Oh & Iyengar, 2019; 2021; Chen et al., 2023) given by

$$p(n|S_{k,t}, \theta_k) = \frac{\exp(x_n^\top \theta_k)}{1 + \sum_{m \in S_{k,t}} \exp(x_m^\top \theta_k)}.$$

We denote $x_n^\top \theta_k$ as the latent preference utility of arm $k$ for agent $n$. Following prior work on MNL bandits (Oh & Iyengar, 2019; 2021; Agrawal et al., 2019), we consider that the candidate set size is bounded by $|S_{k,t}| \leq L$ for all arms $k$ and rounds $t$, and that the reward $r_{n,k}$ is known to the arms in advance. This reflects practical constraints in real-world platforms such as ride-hailing, where only a limited number of riders can be suggested to a driver—due to screen limitations or cognitive load—and the reward (e.g., fare or price) is known prior to each assignment.

However, the expected rewards remain unknown, as they depend jointly on both the latent preference utilities and the associated rewards. At each time step $t$, the agents receive stochastic feedback

based on the assortments $\{S_{k,t}\}_{k\in[K]}$. Specifically, for each agent $n \in S_{k,t}$ and arm $k \in [K]$, the feedback is denoted by $y_{n,t} \in 0,1$, where $y_{n,t} = 1$ if arm $k$ accepts agent $n$ (i.e., a successful match occurs), and $y_{n,t} = 0$ otherwise. Following the standard MNL model, each arm $k$ may also choose an outside option $n_0$ (i.e., reject all assigned agents) with probability $p(n_0|S_{k,t}, \theta_k) = 1/(1 + \sum_{m\in S_{k,t}} \exp(x_m^\top \theta_k))$. Then, given assortments to every arm $k$, $\{S_k\}_{k\in[K]}$, the expected reward (revenue) for the assortments at time $t$ is defined as

$$\sum_{k\in[K]} R_k(S_k) := \sum_{k\in[K]} \sum_{n\in S_k} r_{n,k} p(n|S_k, \theta_k) = \sum_{k\in[K]} \sum_{n\in S_k} \frac{r_{n,k} \exp(x_n^\top \theta_k)}{1 + \sum_{m\in S_k} \exp(x_m^\top \theta_k)}.$$

The goal of the problem is to maximize the cumulative expected reward over a time horizon $T$ by learning the unknown parameters $\{\theta_k\}_{k\in[K]}$. We define the oracle strategy as the optimal assortment selection when the preference parameters $\theta_k$ are known. Let the set of all feasible assignments be: $\mathcal{M} = \{\{S_k\}_{k\in[K]} : S_k \subset [N], |S_k| \leq L \forall k \in [K], S_k \cap S_l = \emptyset \forall k \neq l\}$. Then the oracle assortment is given by: $\{S_k^*\}_{k\in[K]} = \operatorname{argmax}_{\{S_k\}_{k\in[K]}\in\mathcal{M}} \sum_{k\in[K]} R_k(S_k)$. Given $\{S_{k,t}\}_{k\in[K]} \in \mathcal{M}$ for all $t \in [T]$, the expected cumulative regret is defined as $\mathcal{R}(T) = \mathbb{E}\big[\sum_{t\in[T]} \sum_{k\in[K]} R_k(S_k^*) - R_k(S_{k,t})\big]$. The objective is to design a policy that minimizes this regret over the time horizon $T$.

Similar to previous work for logistic and MNL bandit (Oh & Iyengar, 2019; 2021; Lee & Oh, 2024; Goyal & Perivier, 2021; Faury et al., 2020; Abeille et al., 2021), we consider the following regularity condition and non-linearity quantity.

**Assumption 3.1.** $\|x_n\|_2 \leq 1$ for all $n \in [N]$ and $\|\theta_k\|_2 \leq 1$ for all $k \in [K]$.

Then we define a problem-dependent quantity regarding non-linearity of the MNL structure as follows:

$$\kappa := \inf_{\theta\in\mathbb{R}^d:\|\theta\|_2\leq 2;n\in S\subseteq[N]:|S|\leq L} p(n|S, \theta)p(n_0|S, \theta).$$

# 4 OPTIMIZATION IN STOCHASTIC MATCHING BANDITS: THE CURSE OF COMPLEXITY

In this work, we develop algorithms for the Stochastic Matching Bandit (SMB) problem with preference feedback. SMB can be viewed as a generalization of the standard Multinomial Logit (MNL) bandit model with a single assortment (Oh & Iyengar, 2021; Lee & Oh, 2024) to a setting with $K$ simultaneous assortments—one for each arm. Applying existing MNL-based methods to this setting requires dynamically selecting $K$ assortments at each round while simultaneously learning arm preferences in an online fashion. This extension introduces significant computational challenges: the resulting combinatorial optimization problem is NP-hard. In contrast, the standard MNL bandit problem with a single assortment is known to be solvable in polynomial time (Oh & Iyengar, 2021). Thus, the SMB framework poses a substantially more complex optimization problem, highlighting the need for efficient algorithmic solutions.

Naively extending MNL bandits (e.g. Oh & Iyengar (2021); Lee & Oh (2024)) to SMB requires defining the UCB index for the expected reward of an assortment $S_k$ for all $k \in [K]$ as $R_{k,t}^{UCB}(S_k) = \sum_{n\in S_k} \frac{r_{n,k} \exp(h_{n,k,t})}{1+\sum_{m\in S_k} \exp(h_{m,k,t})}$, where $h_{n,k,t}$ is an UCB index for the utility value between $n$ and $k$ at each time $t$. Then at each time, the algorithm determines assortments by following the UCB strategy:

$$\{S_{k,t}\}_{k\in[K]} = \operatorname*{argmax}_{\{S_k\}_{k\in[K]}\in\mathcal{M}} \sum_{k\in[K]} R_{k,t}^{UCB}(S_k). \tag{1}$$

While this method can achieve a regret bound of $\tilde{O}(Kr\sqrt{T})$, it suffers from severe computational limitations. Specifically, solving the combinatorial optimization in (1) incurs a worst-case computational cost of $O(K^N)$ *per round*, particularly when the candidate set size $L \geq N$, rendering the approach impractical for large-scale settings. Further details of the algorithm and regret analysis are provided in Appendix A.2.

To overcome the computational burden, we propose a *batched learning* approach that substantially reduces per-round computational cost on average (i.e., the amortized cost). Our method is inspired by the batched bandit literature (Perchet et al., 2015; Gao et al., 2019; Hanna et al., 2023; Dong et al., 2020; Han et al., 2020; Ren & Zhou, 2024; Sawarni et al., 2024), and the full details are presented in the following sections.

**Remark 4.1.** *For combinatorial optimization, approximation oracles (Kakade et al., 2007; Chen et al., 2013) are often used to address computational challenges. However, this approach inevitably targets approximation regret rather than exact regret that we aim to minimize. In this work, we tackle the computational challenges while targeting exact regret by employing batch updates. Note that even under approximation optimization, our proposed batch updates can also be beneficial in further reducing the computational cost. We will discuss this in more detail in Section 5.*

## 5 BATCH LEARNING FOR STOCHASTIC MATCHING BANDITS

---

**Algorithm 1** Batched Stochastic Matching Bandit (`B-SMB`)

---

**Input:** $\kappa$, $M \geq 1$

**Init:** $t \leftarrow 1, T_1 \leftarrow \eta_T$

1 Compute SVD of $X = U\Sigma V^\top$ and obtain $U_r = [u_1, \ldots, u_r]$; Construct $z_n \leftarrow U_r^\top x_n$ for $n \in [N]$

2 **for** $\tau = 1, 2\ldots$ **do**

3     **for** $k \in [K]$ **do**

4        $\widehat{\theta}_{k,\tau} \leftarrow \operatorname{argmin}_{\theta \in \mathbb{R}^r} l_{k,\tau}(\theta)$ with (2) where $\mathcal{T}_{k,\tau-1} = \mathcal{T}^{(1)}_{k,\tau-1} \cup \mathcal{T}^{(2)}_{k,\tau-1}$ and $\mathcal{T}^{(2)}_{k,\tau-1} = \bigcup_{n \in \mathcal{N}_{k,\tau-1}} \mathcal{T}^{(2)}_{n,k,\tau-1}$

       `// Assortments Construction`

5        $\{S^{(n,k)}_{l,\tau}\}_{l \in [K]} \leftarrow \operatorname*{argmax}_{\{S_l\}_{l \in [K]} \in \mathcal{M}_{\tau-1}: n \in S_k} \sum_{l \in [K]} R^{UCB}_{l,\tau}(S_l)$ for all $n \in \mathcal{N}_{k,\tau-1}$ with (3)

       `// Elimination`

6        $\mathcal{N}_{k,\tau} \leftarrow \{n \in \mathcal{N}_{k,\tau-1} : \max_{\{S_l\}_{l \in [K]} \in \mathcal{M}_{\tau-1}} \sum_{l \in [K]} R^{LCB}_{l,\tau}(S_l) \leq \sum_{l \in [K]} R^{UCB}_{l,\tau}(S^{(n,k)}_{l,\tau})\}$ with (3)

       `// G-Optimal Design`

7        $\pi_{k,\tau} \leftarrow \operatorname{argmin}_{\pi \in \mathcal{P}(\mathcal{N}_{k,\tau})} \max_{n \in \mathcal{N}_{k,\tau}} \|z_n\|^2_{(\sum_{n \in \mathcal{N}_{k,\tau}} \pi_{k,\tau}(n) z_n z_n^\top + (1/rT_\tau)I_r)^{-1}}$

       `// Exploration`

8        Run `Warm-up` (Algorithm 4) over time steps in $\mathcal{T}^{(1)}_{k,\tau}$ (defined in Algorithm 4)

9        **for** $n \in \mathcal{N}_{k,\tau}$ **do**

10          $t_{n,k} \leftarrow t, \mathcal{T}^{(2)}_{n,k,\tau} \leftarrow [t_{n,k}, t_{n,k} + \lceil r\pi_{k,\tau}(n)T_\tau \rceil - 1]$

11          **while** $t \in \mathcal{T}^{(2)}_{n,k,\tau}$ **do**

12            Offer $\{S_{l,t}\}_{l \in [K]} = \{S^{(n,k)}_{l,\tau}\}_{l \in [K]}$ and observe feedback $y_{m,t} \in \{0,1\}$ for all $m \in S_{l,t}$ and $l \in [K]$

13            $t \leftarrow t + 1$

14     $\mathcal{M}_\tau \leftarrow \{\{S_k\}_{k \in [K]} : S_k \subset \mathcal{N}_{k,\tau}, |S_k| \leq L \ \forall k \in [K], S_k \cap S_l = \emptyset \ \forall k \neq l\}; T_{\tau+1} \leftarrow \eta_T \sqrt{T_\tau}$

---

For batch learning to reduce the computational cost, we adopt the elimination-based bandit algorithm (Lattimore & Szepesvári, 2020). This approach presents several key challenges in the framework of SMB, including efficiently handling the large number of possible matchings between agents and arms for elimination, designing an appropriate estimator for the elimination process, and minimizing the total number of updates to reduce computational overhead. The details of our algorithm (Algorithm 1) is described as follows.

Before advancing on the rounds, the algorithm computes Singular Value Decomposition (SVD) for feature matrix $X = U\Sigma V^\top \in \mathbb{R}^{d \times N}$. From $U = [u_1, \ldots, u_d] \in \mathbb{R}^{d \times d}$ and $rank(X) = r$, we can construct $U_r = [u_1, \ldots, u_r] \in \mathbb{R}^{d \times r}$ by extracting the left singular vectors from $U$ that correspond to non-zero singular values. We note that the algorithm does not necessitate prior knowledge of $r$ because the value can be obtained from SVD. The algorithm, then, operates within the full-rank $r$-dimensional feature space with $z_n = U_r^\top x_n \in \mathbb{R}^r$ for $n \in [N]$. Let $\theta^*_k = U_r^\top \theta_k$. Then we can reformulate the MNL model using $r$-dimensional feature $z_n \in \mathbb{R}^r$ and latent $\theta^*_k \in \mathbb{R}^r$. The detailed description for the insight behind this approach is deferred to Appendix A.3.

In what follows, we describe the process for constructing assortments at each time step. The algorithm consists of several epochs. For each $k \in [K]$, from observed feedback $y_{n,t} \in \{0,1\}$ for

$n \in S_{k,t}$, $t \in \mathcal{T}_{k,\tau-1}$, where $\mathcal{T}_{k,\tau-1}$ is a set of the exploration time steps regarding arm $k$ in the $\tau - 1$-th epoch, we first define the negative log-likelihood loss as

$$l_{k,\tau}(\theta) = -\sum_{t \in \mathcal{T}_{k,\tau-1}} \sum_{n \in S_{k,t} \cup \{n_0\}} y_{n,t} \log p(n|S_{k,t}, \theta) + \tfrac{1}{2}\|\theta\|_2^2, \tag{2}$$

where, with a slight abuse of notation, $p(n|S_{k,t}, \theta) := \exp(z_n^\top \theta)/(1 + \sum_{m \in S_{k,t}} \exp(z_m^\top \theta))$. Then at the beginning of each epoch $\tau$, the algorithm estimates $\widehat{\theta}_{k,\tau}$ from the method of Maximum Likelihood Estimation (MLE).

From the estimator, we define upper and lower confidence bounds for expected reward of assortment $S_k$ as

$$R_{k,\tau}^{UCB}(S_k) := \sum_{n \in S_k} [r_{n,k} p(n|S_k, \widehat{\theta}_{k,\tau})] + 2\beta_T \max_{n \in S_k} \|z_n\|_{V_{k,\tau}^{-1}},$$

$$R_{k,\tau}^{LCB}(S_k) := \sum_{n \in S_k} [r_{n,k} p(n|S_k, \widehat{\theta}_{k,\tau})] - 2\beta_T \max_{n \in S_k} \|z_n\|_{V_{k,\tau}^{-1}}, \tag{3}$$

where confidence term $\beta_T = \frac{C_1}{\kappa}\sqrt{\log(TNK)}$ for some constant $C_1 > 0$ and $V_{k,\tau} = \sum_{t \in \mathcal{T}_{k,\tau-1}} \sum_{n \in S_{k,t}} z_n z_n^\top + I_r$. It is important to note that, unlike prior MNL bandit literature (Oh & Iyengar, 2021; Lee & Oh, 2024), which constructs confidence intervals on each latent utility within the MNL function, our approach places the confidence term outside the MNL structure, as shown in (3). This modification is essential due to the need to incorporate both UCB and LCB indices in conjunction with the reward terms $r_{n,k}$. In particular, while our LCB formulation provides a valid lower bound on the expected reward, applying LCBs directly to the latent utility values does not guarantee a lower bound on the reward. This distinction is crucial for ensuring theoretical guarantees in our learning algorithm.

For batch updates, we utilize elimination for suboptimal matches. However, exploring all possible matchings naïvely for the elimination is statistically expensive. Therefore, we utilized a statistically efficient exploration strategy by assessing the eligibility of each assignment $(n, k)$ for $n \in \mathcal{N}_{k,\tau-1}$ and $k \in [K]$ as a potential optimal assortment, where $\mathcal{N}_{k,\tau-1}$ is the active set of agents regarding arm $k$ at epoch $\tau$. To evaluate the assignment $(n, k)$, it constructs a representative assortment of $\{S_{l,\tau}^{(n,k)}\}_{l \in [K]}$ from an optimistic view (Line 5). Then based on the representative assortments, it obtains $\mathcal{N}_{k,\tau}$ by eliminating $n \in \mathcal{N}_{k,\tau-1}$ which satisfies an elimination condition (Line 6). From the obtained $\mathcal{N}_{k,\tau}$ for all $k \in [K]$, it constructs an active set of assortments $\mathcal{M}_\tau$ (Line 14), which is likely to contain the optimal assortments as $\{S_k^*\}_{k \in [K]} \in \mathcal{M}_\tau$.

Following the elimination process outlined above, here we describe the policy of assigning assortment $\{S_{k,t}\}_{k \in [K]}$ at each time $t$ corresponding to Lines 7-13 in Algorithm 1. The algorithm initiates the warm-up stage (Algorithm 4 in Appendix A.4) to apply regularization to the estimators, by uniform exploration across all agents $n \in [N]$ for each arm $k \in [K]$. Then for each $k \in [K]$, the algorithm utilizes the G-optimal design problem (Lattimore & Szepesvári, 2020) to obtain proportion $\pi_{k,\tau} \in \mathcal{P}(\mathcal{N}_{k,\tau})$ for learning $\theta_k^*$ efficiently by exploring agents in $\mathcal{N}_{k,\tau}$, where $\mathcal{P}(\mathcal{N}_{k,\tau})$ is the probability simplex with vertex set $\mathcal{N}_{k,\tau}$. Notably, the G-optimal design problem can be solved by the Frank-Wolfe algorithm (Damla Ahipasaoglu et al., 2008). Then, for all $n \in \mathcal{N}_{k,\tau}$, it explores $\{S_{l,\tau}^{(n,k)}\}_{l \in [K]}$ several times using $\pi_{k,\tau}(n)$ which is the corresponding value of $n$ in $\pi_{k,\tau}$.

The algorithm repeats those processes over epochs $\tau$ until it reaches the time horizon $T$. We schedule $T_\tau$ rounds for each epoch by updating $T_\tau = \eta_T \sqrt{T_{\tau-1}}$. Then, the algorithm requires a limited number of updates for assortment assignments, which is crucial to reduce the amortized computational cost. Let $\eta_T = (T/rK)^{1/2(1-2^{-M})}$ with a parameter for batch update budget $M \geq 1$. Let $\tau_T$ be the last epoch over $T$, which indicates the number of batch updates. We next observe that the scheduling parameter $M$ serves as a budget for the number of batch updates, as formalized in the following proposition. This parameter plays a key role in the amortized efficiency of our algorithm, which we discuss shortly. (The proof of the proposition is provided in Appendix A.5.)

**Proposition 5.1** (Number of Batch Updates). $\tau_T \leq M$.

We establish the following regret bound for our algorithm, with the proof provided in Appendix A.6.

**Theorem 5.2.** *Algorithm 1 with $M = O(\log T)$ achieves:*

$$\mathcal{R}(T) = \tilde{\mathcal{O}}\left( \frac{1}{\kappa} K^{\frac{3}{2}} \sqrt{rT} \left( \frac{T}{rK} \right)^{\frac{1}{2(2^M - 1)}} \right).$$

**Corollary 5.3.** *For $M = \Theta(\log \log(T/rK))$, Algorithm 1 achieves:*

$$\mathcal{R}(T) = \tilde{\mathcal{O}}\left( \frac{1}{\kappa} K^{3/2} \sqrt{rT} \right).$$

**Remark 5.4** (Amortized Efficiency). *As mentioned in Corollary 5.3, our algorithm only requires combinatorial optimization at most $M = \Theta(\log \log(T/rK))$ times over $T$, while achieving $\tilde{O}(\sqrt{T})$ regret bound. This implies that the amortized computation cost is $O(1)$ for large enough $T$, since the average cost per round for combinatorial optimization becomes negligible as $\frac{NK^{N+1} \log \log(T/rK)}{T} = O(1)$ for $T = \Omega(NK^{N+1} \log \log(T/rK))$. Furthermore, in Algorithm 1, the optimization is not performed over the full matching spaces, but the active set $\mathcal{M}_\tau$, whose size typically shrinks quickly due to elimination—an effect that we later confirm empirically. This is significantly lower than the computational cost of the naive approach discussed in Section 4 (e.g. Algorithm 3 in Appendix A.2), which is $O(K^N)$ per round.*

**Discussion on the Tightness of the Regret Bound.** We begin by comparing our results to those from previous batch bandit studies under a (generalized) linear structure. Our regret bound, given as $\tilde{O}(T^{1/2 + 1/2(2^M - 1)}) = \tilde{O}(T^{1/2(1 - 2^{-M})})$ for a general $M = O(\log(T))$, matches the results from Han et al. (2020); Ren & Zhou (2024); Sawarni et al. (2024). Notably, this bound also aligns with the lower bound for the linear structure, $\Omega(T^{1/2(1 - 2^{-M})})$ (Han et al., 2020). For the case of $M = \Theta(\log \log(T/rK))$, our bound of $\tilde{O}(\sqrt{T})$ corresponds to the findings for linear bandits in Ruan et al. (2021); Hanna et al. (2023), where only such values of $M$ were considered. Additionally, with respect to the parameter $r$, we achieve a tight bound of $O(\sqrt{r})$ for $M = \Theta(\log \log(T/rK))$, which matches the lower bound for linear bandits established by Lattimore & Szepesvári (2020). To the best of our knowledge, this is the first work to address batch updates in matching bandits.

Given that our problem generalizes the single-assortment MNL setting to $K$-multiple assortments, we can obtain the regret lower bound of $\Omega(K\sqrt{T})$ with respect to $K$ and $T$ for the contextual setting, by simply extending the result of Theorem 3 in Lee & Oh (2024) for single-assortment to $K$-multiple assortments. In comparison, our analysis indicates a regret dependence of $K^{3/2}$ when $M = \Theta(\log \log(T/(rK)))$, which is worse by a factor of $\sqrt{K}$ relative to the lower bound. This gap arises from the need to explore all potential matches during the epoch-based elimination procedure in batch updates.

Our batch updates can also be applied to approximation oracles, introduced in Kakade et al. (2007); Chen et al. (2013) to mitigate computational challenges in combinatorial optimization. The approximation oracle approach focuses on obtaining an approximate solution to the optimization problem rather than identifying the exact optimal assortment, with the trade-off of incurring a guarantee for a relaxed regret measure ($\gamma$-regret). Further details are provided in Appendix A.8.

Although Algorithm 1 is amortized efficient in computation, achieving regret of $\tilde{O}(\sqrt{T})$, the regret bound relies on problem-specific knowledge of $\kappa$ and, importantly, requires this parameter to be known in advance for setting $\beta_T$. The regret bound scales linearly with $1/\kappa$, which can be as large as $O(L^2)$ in the worst-case scenario. In the following section, we propose an algorithm improving the dependence on $\kappa$ without using the knowledge of $\kappa$.

## 6 IMPROVING DEPENDENCE ON $\kappa$ WITHOUT PRIOR KNOWLEDGE

Here we provide details of our proposed algorithm (Algorithm 2 in Appendix A.1), focusing on the difference from the algorithm in the previous section. While we follow the framework of Algorithm 1, for the improvement on $\kappa$ without knowledge of it, we utilize the local curvature information for the gram matrix as

$$H_{k,\tau}(\widehat{\theta}_{k,\tau}) = \sum_{t \in \mathcal{T}_{k,\tau-1}} \left[ \sum_{n \in S_{k,t}} p(n|S_{k,t}, \widehat{\theta}_{k,\tau}) z_n z_n^\top - \sum_{n,m \in S_{k,t}} p(n|S_{k,t}, \widehat{\theta}_{k,\tau}) p(m|S_{k,t}, \widehat{\theta}_{k,\tau}) z_n z_m^\top \right] + \lambda I_r, \quad (4)$$

where $\lambda = C_2 r \log(K)$ for some constant $C_2 > 0$ and we denote $H_{k,\tau}(\widehat{\theta}_{k,\tau})$ by $H_{k,\tau}$ when there is no confusion. We define $\tilde{z}_{n,k,\tau}(S_{k,t}) = z_n - \sum_{m \in S_{k,t}} p(m|S_{k,t}, \widehat{\theta}_\tau) z_m$ and we use $\tilde{z}_{n,k,\tau}$ for it, when there is no confusion. For the confidence bound, we define

$$B_\tau(S_{k,t}) := \frac{13}{2} \zeta_\tau^2 \max_{n \in S_{k,t}} \|z_n\|_{H_{k,\tau}^{-1}}^2 + 2\zeta_\tau^2 \max_{n \in S_{k,t}} \|\tilde{z}_{n,k,\tau}\|_{H_{k,\tau}^{-1}}^2 + \zeta_\tau \sum_{n \in S_{k,t}} p(n|S_{k,t}, \widehat{\theta}_{k,\tau-1}) \|\tilde{z}_{n,k,\tau}\|_{H_{k,\tau}^{-1}},$$

where $\zeta_\tau = \frac{1}{2}\sqrt{\lambda} + \frac{2r}{\sqrt{\lambda}} \log(4KT(1 + \frac{2(t_\tau - 1)L}{r\lambda}))$ with the start time of $\tau$-th episode $t_\tau$. We note that the first term arises from the second-order term in the Taylor expansion for the error from estimator, while the second and last terms originate from the first-order term. Notably, our confidence bounds for $\tau$-th episode utilize not only the current estimator $\widehat{\theta}_{k,\tau}$ but the previous one $\widehat{\theta}_{k,\tau-1}$ (in the last term) because the historical data in $H_{k,\tau}$ is obtained from the G/D-optimal policy which is optimized under $\widehat{\theta}_{k,\tau-1}$. Then we define upper and lower confidence bounds as

$$R_{k,\tau}^{UCB}(S_{k,t}) := \sum_{n \in S_{k,t}} r_{n,k} p(n|S_{k,t}, \widehat{\theta}_{k,\tau}) + B_\tau(S_{k,t}),$$

$$R_{k,\tau}^{LCB}(S_{k,t}) := \sum_{n \in S_{k,t}} r_{n,k} p(n|S_{k,t}, \widehat{\theta}_{k,\tau}) - B_\tau(S_{k,t}). \tag{5}$$

For the G/D-optimal design aimed at exploring the space of arms, the algorithm must account for both $V_{k,\tau}$ and $H_{k,\tau}(\widehat{\theta}_{k,\tau})$ to achieve a tight regret bound that avoids dependence on $1/\kappa$. This marks a key distinction from Algorithm 1. From this, the algorithm requires two different types of procedures regarding assortment construction, elimination, and exploration. Let $\mathcal{J}(A)$ be the set of all combinations of subset of $A$ with cardinality bound $L$ as $\mathcal{J}(A) = \{B \subseteq A \mid |B| \leq L\}$, and let $\mathcal{K}(A)$ be the set of all combinations of subset $A$ (with cardinality bound $L$) and its element as $\mathcal{K}(A) = \{(b, B) \mid b \in B \subseteq A, |B| \leq L\}$. The G/D-optimal design seeks to minimize the ellipsoidal volume under $V_{k,\tau}$, based on arm selection probabilities within the active set of arms $\mathcal{N}_{k,\tau}$. Additionally, since the action space in $H_{k,\tau}(\widehat{\theta}_{k,\tau})$ depends not only on the selection of actions but also on the selection of assortments, the G/D-optimal design incorporates assortment selection probabilities for $\mathcal{J}(\mathcal{N}_{k,\tau})$ and $\mathcal{K}(\mathcal{N}_{k,\tau})$. Following this policy, the algorithm includes two separate exploration procedures regarding the selection of arms and assortments.

**Remark 6.1.** *It is worth noting that our localized Gram matrix in* (4) *offers advantages over the localized Gram matrices proposed in the MNL bandit literature (Goyal & Perivier, 2021; Lee & Oh, 2024). In Goyal & Perivier (2021), the localized term introduces a dependency on non-convex optimization to achieve optimism, whereas our approach utilizes $\widehat{\theta}_{k,\tau}$ without requiring such complex optimization. Meanwhile, Lee & Oh (2024) incorporate all historical information of the estimator into the Gram matrix, which is not well-suited for the G/D-optimal design. In contrast, our method leverages the most current estimator, enabling alignment with the rescaled feature for the G/D-optimal design.*

**Remark 6.2.** *Our G/D-optimal design for the localized Gram matrix differs from those employed in linear bandits (Lattimore & Szepesvári, 2020) and generalized linear bandits (Sawarni et al., 2024). Unlike these settings, where the probability depends on a single action, our approach accounts for the dependence on assortments (combinatorial actions). As a result, it requires exploring a rescaled feature space that considers the assortment space rather than focusing solely on individual actions.*

We set $\eta_T = (T/rK)^{1/(2(1-2^{-M}))}$ with a parameter for batch update budget $M \geq 1$. Then, by following the same proof of Proposition 5.1, we have the following bound for the number of epochs.

**Proposition 6.3** (Number of Batch Updates)**.** $\tau_T \leq M$.

Then, we have the following regret bounds (the proof is provided in Appendix A.1).

**Theorem 6.4.** *Algorithm 2 with $M = O(\log(T))$ achieves: $\mathcal{R}(T) = \tilde{\mathcal{O}}\big(rK^{\frac{3}{2}}\sqrt{T}\big(\frac{T}{rK}\big)^{\frac{1}{2(2^M-1)}}\big)$.*

**Corollary 6.5.** *For $M = \Theta(\log\log(T/rK))$, Algorithm 2 achieves: $\mathcal{R}(T) = \tilde{\mathcal{O}}\big(rK^{\frac{3}{2}}\sqrt{T}\big)$.*

**Remark 6.6** (Improvement on $\kappa$)**.** *This algorithm does not require prior knowledge of $\kappa$, which enhances its practicality in real-world applications. Moreover, in terms of dependence on $\kappa$, the regret bound improves over that of Algorithm 1 (Theorem 5.2) by eliminating the $1/\kappa = O(L^2)$ dependency from the leading term. This improvement comes at the cost of an additional multiplicative factor of $\sqrt{r}$ in the regret.*

**Remark 6.7** (Amortized-Efficiency). *Like Algorithm 1, this advanced algorithm requires only $\Theta(\log\log(T/rK))$ updates to achieve a $\tilde{O}(\sqrt{T})$ regret bound. This implies that the amortized computational cost is $O(1)$ for sufficiently large $T$, since the average cost for combinatorial optimization becomes negligible as $\frac{LK^{1+N}N^L \log\log(T/Kr)}{T} = O(1)$ for $T = \Omega(LK^{1+N}N^L \log\log(T/Kr))$.*

## 7 EXPERIMENTS

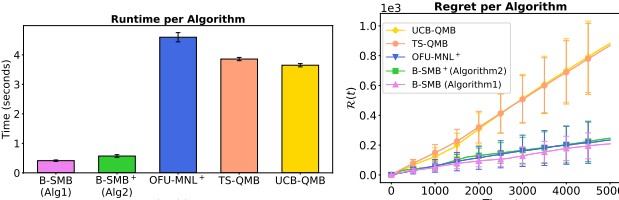

Figure 2: Experimental results with $N = 3$, $K = 2$, for (left) runtime cost and (right) regret

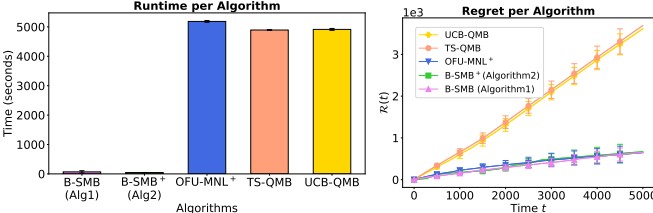

Figure 3: Experimental results with $N = 7$, $K = 4$, for (left) runtime cost and (right) regret

In our experiments, we compare the proposed algorithms with existing methods for MNL bandits and matching bandits under the MNL model. Specifically, the feature vectors $x_n$ and the latent parameters $\theta_k$ are independently sampled from the uniform distribution over $[-1, 1]^d$ and then normalized. Also, the reward $r_{n,k}$ is generated from uniform distribution over $[0, 1]$. We use the settings $N = 3$, $K = 2$, $r = 2$, and $T = 5000$ for Figure 2, and increase the problem size to $N = 7$, $K = 4$ for Figure 3. Additional experiments, including larger problem instances and results illustrating the reduction of the active set, are provided in Appendix A.13.

We first evaluate the computational efficiency of our proposed algorithms, B-SMB (Algorithm 1) and B-SMB$^+$ (Algorithm 2), by comparing them with an adapted version of the MNL bandit algorithm OFU-MNL$^+$ (Lee & Oh, 2024) and existing matching bandit algorithms for the stable MNL model, UCB-QMB and TS-QMB (Kim & Oh, 2024). The details of how OFU-MNL$^+$ is adapted to our setting are provided in Appendix A.2. As discussed in Section 4, although the extension of OFU-MNL$^+$ achieves sublinear regret, it suffers from significant computational overhead due to the need to solve a combinatorial optimization problem at every round. In Figure 2 (left), we observe that our batched algorithms are faster than OFU-MNL$^+$, UCB-QMB, and TS-QMB. This efficiency gap becomes more evident as $N$ and $K$ increase, as shown in Figure 3 (left). Notably, while the computational cost of the benchmark algorithms grows rapidly with larger $N$ and $K$, our batched algorithms maintain their efficiency, demonstrating scalability to larger problem instances.

On the regret side, as shown in Figures 2 and 3 (right), our algorithms achieve sublinear regret comparable to that of OFU-MNL$^+$, in line with our theoretical guarantees, while outperforming UCB-QMB and TS-QMB across both problem sizes.

## 8 CONCLUSION

In this work, we propose a novel and practical framework for stochastic matching bandits, where a naive approach incurs a prohibitive computational cost of $O(K^N)$ per round due to the combinatorial optimization. To address this challenge, we propose an elimination-based algorithm that achieves a regret of $\tilde{O}\left(\frac{1}{\kappa}K^{\frac{3}{2}}\sqrt{rT}\right)$ with $M = \Theta(\log\log(T/rK))$ batch updates under known $\kappa$. Additionally, we present an algorithm without knowledge of $\kappa$, achieving a regret of $\tilde{O}\left(rK^{\frac{3}{2}}\sqrt{T}\right)$ under the same number of batch updates. Leveraging the batch approach, our algorithms significantly reduce the computational overhead, achieving an amortized cost of $O(1)$ per round.

## REPRODUCIBILITY STATEMENT

All theoretical results are derived under clearly stated assumptions, with complete proofs provided in the appendix. The proposed algorithms (`B-SMB` and `B-SMB`$^+$) are described in detail in the main text and appendix, including pseudocode and explanations of the elimination and exploration procedures. To facilitate replication of our experiments, we provide code as supplementary material. The experimental setup is described in the main and Appendix A.13.

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

# A APPENDIX

## A.1 ALGORITHM WITHOUT PRIOR KNOWLEDGE OF $\kappa$ (ALGORITHM 2)

## A.2 NAIVE APPROACH BY EXTENDING MNL BANDIT

For our framework, we can utilize MNL bandit Lee & Oh (2024) by extending it to $K$-mutliple MNLs (Algorithm 3) as follows. Let the negative log-likelihood $l_{k,t}(\theta) = -\sum_{n \in S_{k,s} \cup \{0\}} y_{n,t} \log p(n|S_{k,t}, \theta)$ where $y_{n,t} \in \{0, 1\}$ is observed preference feedback (1 denotes a choice, and 0 otherwise). Then we define the gradient of the likelihood as

$$g_{k,t}(\theta) := \nabla_\theta l_{k,t}(\theta) = \sum_{n \in S_t} (p(n|S_{k,t}, \theta) - y_{n,t}) x_n. \tag{6}$$

We also define gram matrices from $\nabla_\theta^2 l_{k,t}(\theta)$ as follows:

$$G_{k,t}(\theta) := \sum_{n \in S_{k,t}} p(n|S_{k,t}, \theta) z_n z_n^\top - \sum_{n,m \in S_{k,t}} p(n|S_{k,t}, \theta) p(m|S_{k,t}, \theta) z_n z_m^\top. \tag{7}$$

We define the UCB index for assortment $S_k$ as

$$R_{k,t}^{UCB}(S_k) = \sum_{n \in S_k} \frac{\exp(h_{n,k,t})}{1 + \sum_{m \in S_k} \exp(h_{m,k,t})}, \tag{8}$$

where $h_{n,k,t} = z_n^\top \widehat{\theta}_{k,t} + \gamma_t \|z_n\|_{G_{k,t}^{-1}}$ with $\gamma_t = C_4 \log(L) \sqrt{d \log(t) \log(KT)}$ for some $C_4 > 0$. We set $\lambda = C_5 d \log(K)$ and $\eta = C_6 \log(K)$ for some $C_5 > 0$ and $C_6 > 0$.

**Proposition A.1.** *Algorithm 3 achieves a regret bound of $\mathcal{R}(T) = \tilde{O}(rK\sqrt{T})$ and the computational cost per round is $O(K^N)$.*

*Proof.* The proof is provided in Appendix A.10. □

---

**Algorithm 3** Extension of `OFU-MNL+` Lee & Oh (2024)

---

Compute SVD of $X = U\Sigma V^\top$ and obtain $U_r = [u_1, \ldots, u_r]$; Construct $z_n \leftarrow U_r^\top x_n$ for $n \in [N]$
**for** $t = 1, \ldots, T$ **do**
    **for** $k \in [K]$ **do**
        $\tilde{\mathcal{G}}_{k,t} \leftarrow \lambda I_d + \sum_{s=1}^{t-2} G_{k,s}(\widehat{\theta}_{k,s}) + \eta G_{k,t-1}(\widehat{\theta}_{k,t-1})$ with (7)
        $\mathcal{G}_{k,t} \leftarrow \lambda I_d + \sum_{s=1}^{t-1} G_{k,s}(\widehat{\theta}_{k,s})$ with (7)
        $\widehat{\theta}_{k,t} \leftarrow \arg\min_{\theta \in \Theta} g_{k,t-1}(\widehat{\theta}_{k,t-1})^\top \theta + \frac{1}{2\eta} \|\theta - \widehat{\theta}_{k,t-1}\|_{\tilde{\mathcal{G}}_{k,t}^{-1}}^2$ with (6)
    $\{S_{k,t}\}_{k \in [K]} \leftarrow \arg\max_{\{S_k\}_{k \in [K]} \in \mathcal{M}} \sum_{k \in [K]} R_{k,t}^{UCB}(S_k)$ with (8)
    Offer $\{S_{k,t}\}_{k \in [K]}$ and observe $y_{n,t}$ for all $n \in S_{k,t}, k \in [K]$

---

## A.3 DETAILS REGARDING PROJECTION IN FEATURE SPACE

Since $x_n$ for $n \in [N]$ lies in the subspace $U_r$, we observe that $x_n = U_r b_n$ for some $b_n \in \mathbb{R}^r$. Let $\theta_k^* = U_r^\top \theta_k$. Then we have $x_n^\top \theta_k = z_n^\top \theta_k^*$ by following $x_n^\top \theta_k = b_n^\top U_r^\top \theta_k = b_n^\top (U_r^\top U_r) U_r^\top \theta_k = x_n^\top U_r U_r^\top \theta_k = z_n^\top \theta_k^*$ using $U_r^\top U_r = I_d$. Therefore, we can reformulate the MNL model using $r$-dimensional feature $z_n \in \mathbb{R}^r$ and latent $\theta_k^* \in \mathbb{R}^r$ in place of $d$-dimensional $x_n \in \mathbb{R}^d$ and $\theta_k \in \mathbb{R}^d$, respectively, for $n \in [N]$ and $k \in [K]$. We note that this procedure is beneficial not only for reducing feature dimension but also for introducing appropriate regularization for estimators without imposing any assumption about feature distributions considered in Oh & Iyengar (2021).

## A.4 WARM-UP STAGE FOR ALGORITHM 1

Let $\lambda_{\min}(A)$ denote the minimum eigenvalue of matrix $A$. Then we provide the warm-up stage for Algorithm 1 in Algorithm 4.

---

**Algorithm 2** Batched Stochastic Matching Bandit$^+$ (B$-$SMB$^+$)

---

**Input:** $M \geq 1$; **Init:** $t \leftarrow 1, T_1 \leftarrow C_3 \log(T) \log^2(TKL)$ for some constant $C_3 > 0$

15: Compute SVD of $X = U\Sigma V^\top$ and obtain $U_r = [u_1, \ldots, u_r]$; Construct $z_n \leftarrow U_r^\top x_n$ for $n \in [N]$

16: **for** $\tau = 1, 2\ldots$ **do**

17:   **for** $k \in [K]$ **do**

18:     $\widehat{\theta}_{k,\tau} \leftarrow \operatorname{argmin}_{\theta \in \mathbb{R}^r : \|\theta\|_2 \leq 1} l_{k,\tau}(\theta)$ with (2) where $\mathcal{T}_{k,\tau-1} = \bigcup_{n \in \mathcal{N}_{k,\tau-1}} \mathcal{T}_{n,k,\tau-1} \bigcup_{J \in \mathcal{J}(\mathcal{N}_{k,\tau-1})} \mathcal{T}_{J,k,\tau-1}$

    // Assortments Construction

19:     $\{S_{l,\tau}^{(n,k)}\}_{l \in [K]} \leftarrow \operatorname{argmax}_{\{S_l\}_{l \in [K]} \in \mathcal{M}_{\tau-1} : n \in S_k} \sum_{l \in [K]} R_{l,\tau}^{UCB}(S_l)$ for all $n \in \mathcal{N}_{k,\tau-1}$ with (5)

20:     $\{S_{l,\tau}^{(J,k)}\}_{l \in [K]} \leftarrow \operatorname{argmax}_{\{S_l\}_{l \in [K]} \in \mathcal{M}_{\tau-1} : S_k = J} \sum_{l \in [K]} R_{l,\tau}^{UCB}(S_l)$ for all $J \in \mathcal{J}(\mathcal{N}_{k,\tau-1})$ with (5)

    // Elimination

21:     $\mathcal{N}'_{k,\tau} \leftarrow \{n \in \mathcal{N}_{k,\tau-1} : \max_{\{S_l\}_{l \in [K]} \in \mathcal{M}_{\tau-1}} \sum_{l \in [K]} R_{l,\tau}^{LCB}(S_l) \leq \sum_{l \in [K]} R_{l,\tau}^{UCB}(S_{l,\tau}^{(n,k)})\}$ with (5)

22:     $\mathcal{N}_{k,\tau} \leftarrow \{n \in J : J \in \mathcal{J}(\mathcal{N}'_{k,\tau}), \max_{\{S_l\}_{l \in [K]} \in \mathcal{M}_{\tau-1}} \sum_{l \in [K]} R_{l,\tau}^{LCB}(S_l) \leq \sum_{l \in [K]} R_{l,\tau}^{UCB}(S_{l,\tau}^{(J,k)})\}$ with (5)

    // G-Optimal Design

23:     $\pi_{k,\tau} \leftarrow \operatorname{argmin}_{\pi \in \mathcal{P}(\mathcal{N}_{k,\tau})} \max_{n \in \mathcal{N}_{k,\tau}} \|z_n\|^2_{(\sum_{n \in \mathcal{N}_{k,\tau}} \pi(n) z_n z_n^\top + (\lambda/rT_\tau)I_r)^{-1}}$

24:     $\tilde{\pi}_{k,\tau} \leftarrow \operatorname*{argmin}_{\pi \in \mathcal{P}(\mathcal{J}(\mathcal{N}_{k,\tau}))} \max_{J \in \mathcal{J}(\mathcal{N}_{k,\tau})} \left\|\sum_{n \in J} \tilde{z}'_{n,k,\tau}(J)\right\|^2_{(\sum_{J \in \mathcal{J}(\mathcal{N}_{k,\tau})} \pi(J) \sum_{n \in J} \tilde{z}'_{n,k,\tau}(J) \tilde{z}'_{n,k,\tau}(J)^\top + (\lambda/T_\tau r)I_r)^{-1}}$

    where $\tilde{z}'_{n,k,\tau}(J) = \sqrt{p(n|J, \widehat{\theta}_{k,\tau})} \tilde{z}_{n,k,\tau}(J)$

25:     $\bar{\pi}_{k,\tau} \leftarrow \operatorname*{argmin}_{\pi \in \mathcal{P}(\mathcal{K}(\mathcal{N}_{k,\tau}))} \max_{(n,J) \in \mathcal{K}(\mathcal{N}_{k,\tau})} \|\tilde{z}_{n,k,\tau}(J)\|^2_{(\sum_{(n,J) \in \mathcal{K}(\mathcal{N}_{k,\tau})} \pi(n,J) \tilde{z}_{n,k,\tau}(J) \tilde{z}_{n,k,\tau}(J)^\top + (\lambda/T_\tau r)I_r)^{-1}}$

    // Exploration

26:     **for** $n \in \mathcal{N}_{k,\tau}$ **do**

27:       $t_{n,k} \leftarrow t, \mathcal{T}_{n,k,\tau} \leftarrow [t_{n,k}, t_{n,k} + \lceil r\pi_{k,\tau}(n)T_\tau \rceil - 1]$

28:       **while** $t \in \mathcal{T}_{n,k,\tau}$ **do**

29:         Offer $\{S_{l,t}\}_{l \in [K]} = \{S_{l,\tau}^{(n,k)}\}_{l \in [K]}$ and observe feedback $y_{m,t} \in \{0,1\}$ for all $m \in S_{l,t}$ and $l \in [K]$

30:         $t \leftarrow t + 1$

31:     **for** $J \in \mathcal{J}(\mathcal{N}_{k,\tau})$ **do**

32:       $t_{J,k} \leftarrow t, \mathcal{T}_{J,k,\tau} \leftarrow [t_{J,k}, t_{J,k} + \lceil r\tilde{\pi}_{k,\tau}(J)T_\tau \rceil - 1]$

33:       **while** $t \in \mathcal{T}_{J,k,\tau}$ **do**

34:         Offer $\{S_{l,t}\}_{l \in [K]} = \{S_{l,\tau}^{(J,k)}\}_{l \in [K]}$ and observe feedback $y_{m,t} \in \{0,1\}$ for all $m \in S_{l,t}$ and $l \in [K]$

35:         $t \leftarrow t + 1$

36:     **for** $(n, J) \in \mathcal{K}(\mathcal{N}_{k,\tau})$ **do**

37:       $t_{n,J,k} \leftarrow t, \mathcal{T}_{n,J,k,\tau} \leftarrow [t_{n,J,k}, t_{n,J,k} + \lceil r\bar{\pi}_{k,\tau}(n,J)T_\tau \rceil - 1]$

38:       **while** $t \in \mathcal{T}_{J,k,\tau}$ **do**

39:         Offer $\{S_{l,t}\}_{l \in [K]} = \{S_{l,\tau}^{(J,k)}\}_{l \in [K]}$ and observe feedback $y_{m,t} \in \{0,1\}$ for all $m \in S_{l,t}$ and $l \in [K]$

40:         $t \leftarrow t + 1$

41:   $\mathcal{M}_\tau \leftarrow \{\{S_k\}_{k \in [K]} : S_k \subseteq \mathcal{N}_{k,\tau}, |S_k| \leq L \, \forall k \in [K], S_k \cap S_l = \emptyset \, \forall k \neq l\}$

42:   $T_{\tau+1} \leftarrow \eta_T \sqrt{T_\tau}$

---

---

**Algorithm 4** `Round-robin Warm-up`

---

$\lambda_{\min} \leftarrow \lambda_{\min}(\sum_{n \in [N]} z_n z_n^\top)$
$t_k \leftarrow t, i \leftarrow \min\{L, N\}$
$T_k' \leftarrow (C_3 N / i\kappa^2 \lambda_{\min} \log(TK))(r + \log(TK))^2$
$\mathcal{T}_{k,\tau}^{(1)} \leftarrow [t_k, t_k + T_k' - 1]$
**for** $t \in \mathcal{T}_{k,\tau}^{(1)}$ **do**
    $a \leftarrow (i(t-1) + 1 \mod N), b \leftarrow (it \mod N)$
    **if** $a \leq b$ **then**
        $S_{k,t} \leftarrow [a, b]$
    **else**
        $S_{k,t} \leftarrow [1, b] \cup [a, N]$
    Construct any $S_{l,t}$ for $l \in [K]/\{k\}$ satisfying $\{S_{k,t}\}_{k \in [K]} \in \mathcal{M}_0$
    Offer $\{S_{k,t}\}_{k \in [K]}$ and observe feedback $y_{n,t} \in \{0,1\}$ for all $n \in S_{k,t}, k \in [K]$

---

## A.5 PROOF OF PROPOSITION 5.1

Here we utilize the proof techniques in Sawarni et al. (2024). Recall that $\tau_T$ to be the smallest $\tau \in [T]$ such that

$$\sum_{\tau' \in [\tau]} \sum_{k \in [K]} |\mathcal{T}_{k,\tau'}^{(1)}| + |\mathcal{T}_{k,\tau'}^{(2)}| \geq T.$$

In other words, $\sum_{\tau' \in [\tau_T - 1]} \sum_{k \in [K]} |\mathcal{T}_{k,\tau'}^{(1)}| + |\mathcal{T}_{k,\tau'}^{(2)}| < T$. Then we can show that $\tau_T \leq M$ by contradiction as follows. Suppose $\tau_T > M$. Then, we have

$$T_{\tau_T - 1} \geq (\eta_T)^{\sum_{k=1}^{\tau_T - 1} (\frac{1}{2})^{k-1}} \geq (\eta_T)^{2(1 - (\frac{1}{2})^{\tau_T - 1})} = (T/rK)^{\frac{1 - 2^{1-\tau_T}}{1 - 2^{-M}}} \geq T/rK,$$

where the last inequality comes from $M + 1 \leq \tau_T$. This implies that $\sum_{\tau' \in [\tau_T - 1]} \sum_{k \in [K]} |\mathcal{T}_{k,\tau'}^{(1)}| + |\mathcal{T}_{k,\tau'}^{(2)}| \geq KrT_{\tau_T - 1} \geq T$, which is contradiction. Thus, we can conclude that $\tau_T \leq M$.

## A.6 PROOF OF THEOREM 5.2

In the following proof, with a slight abuse of notation, we use $p(n|S,\theta) = \exp(z_n^\top \theta)/(1 + \sum_{m \in S} \exp(z_m^\top \theta))$ with $z_n \in \mathbb{R}^r$ instead of $x_n \in \mathbb{R}^d$. We provide a lemma for a confidence bound.

**Lemma A.2.** *For any $\tau \in [T]$, $k \in [K]$, and $n \in [N]$, with probability at least $1 - \delta$, for some constant $C > 0$, we have*

$$|z_n^\top(\widehat{\theta}_{k,\tau} - \theta_k^*)| \leq \frac{C}{\kappa} \sqrt{\|z_n\|_{V_{k,\tau}^{-1}}^2 \log(TKN/\delta)}.$$

*Proof.* We define the gradient of the likelihood as

$$g_{k,\tau}(\theta) := \sum_{t \in \mathcal{T}_{k,\tau}} \nabla_\theta l_{k,t}(\theta) = \sum_{t \in \mathcal{T}_{k,\tau}} \sum_{n \in S_{k,t}} (p(n|S_{k,t}, \theta) - y_{n,t}) z_n + \theta.$$

Then we first provide a bound in the following lemma.

**Lemma A.3.** *For any $n \in [N]$, $k \in [K]$, and $\tau \in [T]$, with probability at least $1 - \delta$, we have*

$$|z_n^\top(\widehat{\theta}_{k,\tau} - \theta_k^*)| \leq \frac{3\sqrt{\log(TKN/\delta)}}{\kappa} \|z_n\|_{V_{k,\tau}^{-1}} + \frac{6}{\kappa^2} \|\widehat{\theta}_{k,\tau} - \theta_k^*\|_2 \|g_{k,\tau}(\widehat{\theta}_{k,\tau}) - g_{k,\tau}(\theta_k^*)\|_{V_{k,\tau}^{-1}} \|z_n\|_{V_{k,\tau}^{-1}}.$$

*Proof.* The proof is deferred to Appendix A.9.1 □

Then we define

$$E_1 = \left\{ |z_n^\top(\widehat{\theta}_{k,\tau} - \theta_k^*)| \leq \frac{3\sqrt{\log(TKN/\delta)}}{\kappa} \|z_n\|_{V_{k,\tau}^{-1}} \right.$$

$$\left. + \frac{6}{\kappa^2} \|\widehat{\theta}_{k,\tau} - \theta_k^*\|_2 \|g_{k,\tau}(\widehat{\theta}_{k,\tau}) - g_{k,\tau}(\theta_k^*)\|_{V_{k,\tau}^{-1}} \|z_n\|_{V_{k,\tau}^{-1}} \, \forall n \in [N], k \in [K], \tau \in [T] \right\},$$

which holds at least $1-\delta$. Now we provide bounds for $\|\widehat{\theta}_{k,\tau} - \theta_k^*\|_2$ and $\|g_{k,\tau}(\widehat{\theta}_{k,\tau}) - g_{k,\tau}(\theta_k^*)\|_{V_{k,\tau}^{-1}}$.

**Lemma A.4** (Lemma 7 in Li et al. (2017))**.** *For all $k \in [K]$, $\tau \in [T]$, with probability at least $1 - \delta$ for $\delta > 0$, we have*

$$\|g_{k,\tau}(\widehat{\theta}_{k,\tau}) - g_{k,\tau-1}(\theta_k^*)\|_{V_{k,\tau}^{-1}} \leq 4\sqrt{2r + \log(KTN/\delta)}.$$

We define $V_{k,\tau}^0 = \sum_{t \in \mathcal{T}_{k,\tau-1}^{(1)}} \sum_{n \in S_{k,t}} z_n z_n^\top$. Then we have the following lemma.

**Lemma A.5.** *For all $k \in [K]$ and $\tau \geq 2$, we have $\lambda_{\min}(V_{k,\tau}^0) \geq (C_0/\kappa^2 \log(TKN/\delta))(r^2 + \log^2(TKN/\delta) + 2r\log(TKN/\delta))$.*

*Proof.* Let $\lambda' = (C_0/\kappa^2 \lambda_{\min} \log(TK/\delta))(r^2 + \log^2(TKN/\delta) + 2r\log(TKN/\delta))$ and recall $\lambda_{\min} = \lambda_{\min}(\sum_{n \in [N]} z_n z_n^\top)$. From the phase in the warm-up stage (Algorithm 4), we can observe that $V_{k,\tau}^0$ contains $z_n z_n^\top$ for each $n \in [N]$ at least $\lambda'$. Since $\sum_{n \in [N]} z_n z_n^\top = \sum_{s \in [r]} \lambda_s u_s u_s^\top$, we have $V_{k,\tau}^0 = \sum_{t \in \mathcal{T}_{k,\tau-1}^{(1)}} \sum_{n \in S_{k,t}} z_n z_n^\top = \sum_{s \in [r]} \lambda_s' u_s u_s^\top$ where $\lambda_s' \geq \lambda' \lambda_s$. Then from $\lambda_{\min} = \lambda_r$, we can conclude $\lambda_{\min}(V_k^0) \geq \lambda' \lambda_{\min}$. $\square$

**Lemma A.6** (Lemma 9 in Kveton et al. (2020))**.** *Suppose $\lambda_{\min}(V_{k,\tau}^0) \geq \max\{(1/4\kappa^2)(r\log(T/r) + 2\log(KTN/\delta)), 1\}$ for all $k \in [K]$. Then, for all $\tau \in [T]$ and $k \in [K]$, we have*

$$\mathbb{P}(\|\widehat{\theta}_{k,\tau} - \theta_k^*\|_2 \geq 1) \leq 1/\delta.$$

We define $E_2 = \{\|\widehat{\theta}_{k,\tau} - \theta_k^*\|_2 \leq 1 \, \forall k \in [K], \tau \in [T]\}$. Then from Lemmas A.5, A.6, we have $\mathbb{P}(E_1) \geq 1 - \delta$.

We also denote by $E_3$ the event of $\{\|g_{k,\tau}(\widehat{\theta}_{k,\tau}) - g_{k,\tau-1}(\theta_k^*)\|_{V_{k,\tau}^{-1}} \leq 4\sqrt{2r + \log(KTN/\delta)} \, \forall \tau \in [T], k \in [K]\}$, which hold with probability at least $1 - \delta$ from Lemma A.4.

**Lemma A.7.** *Under $E_2$ and $E_3$, for any $\tau \in [T]$, $k \in [K]$, we have*

$$\|\widehat{\theta}_{k,\tau} - \theta_k^*\|_2 \leq \frac{2}{\kappa} \sqrt{\frac{2r + \log(TNK/\delta)}{\lambda_{\min}(V_k^0)}}.$$

*Proof.* The proof is deferred to Appendix A.9.2

$\square$

Finally, under $E_1 \cup E_2 \cup E_3$ which holds with probability at least $1 - 3\delta$, we have

$$|z_n^\top(\widehat{\theta}_{k,\tau} - \theta_k^*)|$$

$$\leq \frac{2\sqrt{\log(TKN/\delta)}}{\kappa} \|z_n\|_{V_{k,\tau}^{-1}} + (6/\kappa^2) \|z_n\|_{V_{k,\tau}^{-1}} \|\widehat{\theta}_{k,\tau} - \theta_k^*\|_2 \|(g_{k,\tau}(\widehat{\theta}_{k,\tau}) - g_{k,\tau}(\theta_k^*))\|_{V_{k,\tau}^{-1}}$$

$$\leq \frac{2\sqrt{\log(TKN/\delta)}}{\kappa} \|z_n\|_{V_{k,\tau}^{-1}} + \frac{48(2r + \log(KTN/\delta))}{\kappa^2 \sqrt{\lambda_{\min}(V_{k,\tau}^0)}} \|z_n\|_{V_{k,\tau}^{-1}}$$

$$\leq \frac{3\sqrt{\log(TKN/\delta)}}{\kappa} \|z_n\|_{V_{k,\tau}^{-1}}$$

$$= (3/\kappa) \sqrt{\|z_n\|_{V_{\tau,k}^{-1}}^2 \log(TKN/\delta)} := \beta(\delta) \|z_n\|_{V_{\tau,k}^{-1}},$$

which concludes the proof. □

Then we define event $E = \{|z_n^\top(\widehat{\theta}_{k,\tau} - \theta_k^*)| \leq \beta_T \|z_n\|_{V_{k,\tau}^{-1}} \ \forall \tau \in [T], k \in [K], n \in [N]\}$ for some $c_1 > 0$, which holds at least $1 - 1/T$ with Lemma A.2 and $\delta = 1/T$.

**Lemma A.8.** *Under $E$, for all $\tau \in [T]$, $k \in [K]$, and $S \subseteq \mathcal{N}_{k,\tau-1}$, we have*

$$0 \leq R_{k,\tau}^{UCB}(S) - R_k(S) \leq 4\beta_T \max_{n \in S} \|z_n\|_{V_{k,\tau}^{-1}} \ \text{ and } \ -4\beta_T \max_{n \in S} \|z_n\|_{V_{k,\tau}^{-1}} \leq R_{k,\tau}^{LCB}(S) - R_k(S) \leq 0$$

*Proof.* Let $u_{n,k} = z_n^\top \theta_k^*$, $\widehat{u}_{n,k} = z_n^\top \widehat{\theta}_{k,\tau}$, and $\widehat{R}_{k,\tau}(S) = \frac{\sum_{n \in S} r_{n,k} \exp(\widehat{u}_{n,k})}{1 + \sum_{m \in S} \exp(\widehat{u}_{m,k})}$. Then by the mean value theorem, there exists $\bar{u}_{n,k} = (1-c)\widehat{u}_{n,k} + c u_{n,k}$ for some $c \in (0,1)$ satisfying, for any $S \subset \mathcal{N}_{k,\tau-1}$

$$\left|\widehat{R}_{k,\tau}(S) - R_k(S)\right| = \left|\frac{\sum_{n \in S} r_{n,k} \exp(\widehat{u}_{n,k})}{1 + \sum_{m \in S} \exp(\widehat{u}_{n,k})} - \frac{\sum_{n \in S} r_{n,k} \exp(u_{n,k})}{1 + \sum_{m \in S} \exp(u_{m,k})}\right|$$

$$= \left|\sum_{n \in S} \nabla_{v_n} \left(\frac{\sum_{m \in S} r_{m,k} \exp(v_m)}{1 + \sum_{m \in S} \exp(v_m)}\right)\Big|_{v_n = \bar{u}_{n,k}} (\widehat{u}_{n,k} - u_{n,k})\right|$$

$$\leq \left|\frac{(1 + \sum_{n \in S} \exp(\bar{u}_{n,k}))(\sum_{n \in S} r_{n,k} \exp(\bar{u}_{n,k})(\widehat{u}_{n,k} - u_{n,k}))}{(1 + \sum_{n \in S} \exp(\bar{u}_{n,k}))^2}\right|$$

$$+ \left|\frac{(\sum_{n \in S} \exp(\bar{u}_{n,k}))(\sum_{n \in S} r_{n,k} \exp(\bar{u}_{n,k})(\widehat{u}_{n,k} - u_{n,k}))}{(1 + \sum_{n \in S} \exp(\bar{u}_{n,k}))^2}\right|$$

$$\leq 2 \sum_{n \in S} \frac{\exp(\bar{u}_{n,k})}{1 + \sum_{m \in S} \exp(\bar{u}_{m,k})}|\widehat{u}_{n,k} - u_{n,k}|$$

$$\leq 2 \max_{n \in S} |\widehat{u}_{n,k} - u_{n,k}|$$

$$\leq 2\beta_T \max_{n \in S} \|z_n\|_{V_{k,\tau}^{-1}},$$

where the last inequality is obtained from, under $E$, $|z_n^\top \theta_k^* - z_n^\top \widehat{\theta}_{k,\tau}| \leq \beta_T \|z_n\|_{V_{k,\tau}^{-1}}$. Then, from the definition of $R_{k,\tau}^{UCB}(S)$ and $R_{k,\tau}^{LCB}(S)$, we can conclude the proof. □

In the following, by adopting the proof technique in Chen et al. (2023), we provide a lemma for showing that $\mathcal{M}_\tau$ is likely to contain the optimal assortment.

**Lemma A.9.** *Under $E$, $(S_1^*, \ldots, S_K^*) \in \mathcal{M}_{\tau-1}$ for all $\tau \in [T]$.*

*Proof.* Here we use induction for the proof. Suppose that for fixed $\tau$, we have $(S_1^*, \ldots, S_K^*) \in \mathcal{M}_\tau$ for all $k \in [K]$. Recall that $\beta_T = (C_1/\kappa)\sqrt{\log(TKN)}$. From Lemma A.8, we have $R_{k,\tau+1}^{UCB}(S) \geq R_k(S)$ and $R_{k,\tau+1}^{LCB}(S) \leq R_k(S)$ for any $S \subset [N]$. Then for $k \in [K]$, $n \in S_k^*$, and any $(S_1, .., S_K) \in \mathcal{M}_\tau$, we have

$$\sum_{l \in [K]} R_{l,\tau+1}^{UCB}(S_{l,\tau+1}^{(n,k)}) \geq \sum_{l \in [K]} R_{l,\tau+1}^{UCB}(S_l^*)$$

$$\geq \sum_{l \in [K]} R_l(S_l^*)$$

$$\geq \sum_{l \in [K]} R_l(S_l)$$

$$\geq \sum_{l \in [K]} R_{l,\tau+1}^{LCB}(S_l), \tag{9}$$

where the first inequality comes from the elimination condition in the algorithm and $(S_1^*, \ldots S_K^*) \in \mathcal{M}_\tau$, and the third inequality comes from the optimality of $(S_1^*, \ldots, S_K^*)$. This implies that $n \in \mathcal{N}_{k,\tau+1}$ from the algorithm. Then by following the same statement of (9) for all $n \in S_k^*$ and

$k \in [K]$, we have $S_k^* \subset \mathcal{N}_{k,\tau+1}$ for all $k \in [K]$, which implies $(S_1^*, \ldots, S_K^*) \in \mathcal{M}_{\tau+1}$. Therefore, with $(S_1^*, \ldots, S_K^*) \in \mathcal{M}_1$, we can conclude the proof from the induction. $\qquad\square$

From the above Lemmas A.8 and A.9, under $E$, we have

$$\sum_{l \in [K]} R_l(S_l^*) - \sum_{l \in [K]} R_l(S_{l,\tau}^{(n,k)}) \leq \sum_{l \in [K]} R_{l,\tau}^{LCB}(S_l^*) + 4\beta_T \max_{m \in S_l^*} \|z_m\|_{V_{l,\tau}^{-1}}$$

$$- \sum_{l \in [K]} R_{l,\tau-1}^{UCB}(S_{l,\tau}^{(n,k)}) + 4\beta_T \max_{m \in S_{l,\tau}^{(n,k)}} \|z_m\|_{V_{l,\tau-1}^{-1}}$$

$$\leq 4\beta_T \sum_{l \in [K]} (\max_{m \in S_l^*} \|z_m\|_{V_{l,\tau-1}^{-1}} + \max_{m \in S_{l,\tau}^{(n,k)}} \|z_m\|_{V_{l,\tau-1}^{-1}}), \quad (10)$$

where the last inequality comes from the fact that $(S_1^*, \ldots, S_K^*) \in \mathcal{M}_{\tau-1}$ and $\max_{(S_1,\ldots,S_K) \in \mathcal{M}_{\tau-1}} \sum_{l \in [K]} R_{l,\tau}^{LCB}(S_l) \leq \sum_{l \in [K]} R_{l,\tau}^{UCB}(S_{l,\tau}^{(n,k)})$ from the algorithm.

We define $V(\pi_{k,\tau}) = \sum_{n \in \mathcal{N}_{k,\tau}} \pi_{k,\tau}(n)z_n z_n^\top$ and $supp(\pi_{k,\tau}) = \{n \in \mathcal{N}_{k,\tau} : \pi_{k,\tau}(n) \neq 0\}$. Then we have the following lemma from the G/D-optimal design problem.

**Lemma A.10** (Theorem 21.1 (Kiefer-Wolfowitz) in Lattimore & Szepesvári (2020)). *For all $\tau \in [T]$ and $k \in [K]$, we have*

$$\max_{n \in \mathcal{N}_{k,\tau}} \|z_n\|^2_{(V(\pi_{k,\tau})+(1/rT_\tau)I_r)^{-1}} \leq r \text{ and } |supp(\pi_{k,\tau})| \leq r(r+1)/2.$$

*Proof.* For completeness, we provide a proof in Appendix A.11. $\qquad\square$

From the definition of $V_{k,\tau}$ and $T_\tau$, we have

$$V_{k,\tau} \succeq \sum_{n \in \mathcal{N}_{k,\tau-1}} r\pi_{k,\tau-1}(n)T_{\tau-1}z_n z_n^\top + I_r$$

$$= T_{\tau-1}r(V(\pi_{k,\tau-1}) + (1/T_{\tau-1}r)I_r). \quad (11)$$

Then from Lemma A.10 and (11), for any $n \in \mathcal{N}_{k,\tau}$ we have

$$\beta_T \|z_n\|_{V_{k,\tau}^{-1}} = (1/\kappa)\sqrt{\|z_n\|^2_{V_{k,\tau}^{-1}} \log(KNT)}$$

$$= \tilde{\mathcal{O}}\left((1/\kappa)\sqrt{1/T_{\tau-1}}\sqrt{\|z_n\|^2_{(V(\pi_{k,\tau-1})+(1/T_{\tau-1}r)I_r)^{-1}}/r}\right)$$

$$= \tilde{\mathcal{O}}((1/\kappa)\sqrt{1/T_{\tau-1}}). \quad (12)$$

Therefore under $E$, from (10) and (12), for $\tau > 1$, we have

$$\sum_{l \in [K]} (R_l(S_l^*) - R_l(S_{l,\tau}^{(n,k)})) = \tilde{\mathcal{O}}((1/\kappa)K\sqrt{1/T_{\tau-1}}).$$

We have

$$\mathcal{R}(T) = \mathbb{E}\left[\sum_{t \in [T]} \sum_{k \in [K]} R_k(S_k^*) - R_k(S_{k,t})\right]$$

$$\leq \mathbb{E}\left[\sum_{\tau \in [\tau_T]} \sum_{l \in [K]} \sum_{t \in \mathcal{T}_{l,\tau}^{(1)} \bigcap \mathcal{T}_{l,\tau}^{(2)}} \sum_{k \in [K]} R_k(S_k^*) - R_k(S_{k,t})\right],$$

$$(13)$$

which consists of regret from the stage of warming up and main. We first analyze the regret from the warming-up as follows:

$$\mathbb{E}\left[\sum_{\tau \in [\tau_T]} \sum_{l \in [K]} \sum_{t \in \mathcal{T}_{l,\tau}^{(1)}} \sum_{k \in [K]} R_k(S_k^*) - R_k(S_{k,t})\right] \leq \mathbb{E}\left[\sum_{\tau \in [\tau_T]} \sum_{l \in [K]} K\left|\mathcal{T}_{l,\tau}^{(1)}\right|\right]$$

$$= \tilde{\mathcal{O}}(r^2 K^2 N/(\min\{L, N\}\kappa^2\lambda_{\min})), \quad (14)$$

where the first equality comes from $\tau_T \leq M = O(\log(\log(T/rK)))$ from Proposition 5.1.

For the regret bound from the main part of the algorithm, with large enough $T$, we have

$$\mathbb{E}\left[\sum_{\tau\in[\tau_T]}\sum_{l\in[K]}\sum_{t\in\mathcal{T}_{l,\tau_T}^{(2)}}\sum_{k\in[K]} R_k(S_k^*) - R_t(S_{k,t})\right]$$

$$\leq \mathbb{E}\left[\sum_{\tau\in[\tau_T]}\sum_{l\in[K]}\sum_{t\in\mathcal{T}_{l,\tau}^{(2)}}\sum_{k\in[K]} (R_k(S_k^*) - R_k(S_{k,t}))\,\mathbf{1}(E)\right]$$

$$+ \mathbb{E}\left[\sum_{\tau\in[\tau_T]}\sum_{l\in[K]}\sum_{t\in\mathcal{T}_{l,\tau}^{(2)}}\sum_{k\in[K]} (R_k(S_k^*) - R_k(S_{k,t}))\,\mathbf{1}(E^c)\right]$$

$$= \tilde{\mathcal{O}}\left((K/\kappa)\sum_{\tau=2}^{\tau_T}\sum_{l\in[K]}\sum_{n\in\mathcal{N}_{l,\tau}} |\mathcal{T}_{n,l,\tau}^{(2)}|\sqrt{1/T_{\tau-1}}\right) + \mathcal{O}(rK\eta_T) + \mathcal{O}(K)$$

$$= \tilde{\mathcal{O}}\left((K/\kappa)\sum_{\tau=2}^{\tau_T}\sum_{l\in[K]}\sum_{n\in\mathcal{N}_{l,\tau}} |\mathcal{T}_{n,l,\tau}^{(2)}|\sqrt{1/T_{\tau-1}}\right) + \mathcal{O}(rK\eta_T)$$

$$= \tilde{\mathcal{O}}\left((K/\kappa)\sum_{\tau=2}^{\tau_T}\sum_{l\in[K]} (rT_\tau + |Supp(\pi_{l,\tau})|)\sqrt{1/T_{\tau-1}}\right) + \mathcal{O}(rK\eta_T)$$

$$= \tilde{\mathcal{O}}\left((K^2/\kappa)\sum_{\tau=2}^{\tau_T} (r\eta_T + r^2\sqrt{1/T_{\tau-1}})\right)$$

$$= \tilde{\mathcal{O}}\left((K^2/\kappa)(r\eta_T + r^2)\right)$$

$$= \tilde{\mathcal{O}}\left(\tfrac{1}{\kappa}rK^2(T/rK)^{\frac{1}{2(1-2^{-M})}}\right), \tag{15}$$

where the third last equality comes from Lemma A.10 and the second last equality comes from $\tau_T \leq M = O(\log(\log(T/rK)))$ from Proposition 5.1. From (13), (14), (15), for $T \geq r^3KN^2/\min\{L,N\}^2\kappa^2\lambda_{\min}^2$, we can conclude the proof.

### A.7 PROOF OF THEOREM 6.4

Let $g_{k,\tau}(\theta) = \sum_{t\in\mathcal{T}_{\tau-1}}\sum_{n\in S_{k,t}} p(n|S_{k,t},\theta)z_n + \lambda\theta$ and $\zeta_\tau(\delta) = \frac{1}{2}\sqrt{\lambda} + \frac{2r}{\sqrt{\lambda}}\log\left(\frac{4K}{\delta}\left(1 + \frac{2(t_\tau-1)L}{r\lambda}\right)\right)$.

**Lemma A.11** (Proposition 2 in Goyal & Perivier (2021))**.** *With probability at least $1 - \delta$, for all $\tau \geq 1$ and $k \in [K]$, we have*

$$\|g_{k,\tau}(\widehat{\theta}_{k,\tau}) - g_{k,\tau}(\theta_k^*)\|_{H_{k,\tau}^{-1}(\theta_k^*)} \leq \zeta_\tau(\delta).$$

From the above lemma, we define event $E = \{\|g_{k,\tau}(\widehat{\theta}_{k,\tau}) - g_{k,\tau}(\theta_k^*)\|_{H_{k,\tau}^{-1}(\theta_k^*)} \leq \zeta_\tau(\delta), \forall \tau \geq 1, k \in [K]\}$. Then we have the following lemma.

**Lemma A.12.** *Under $E$, for any $\tau \geq 1$ and $k \in [K]$, we have*

$$\|\widehat{\theta}_{k,\tau} - \theta_k^*\|_{H_{k,\tau}(\widehat{\theta}_{k,\tau})} \leq (1 + 3\sqrt{2})\zeta_\tau(\delta).$$

*Proof.* Here we utilize the proof techniques in Goyal & Perivier (2021). Let $G_{k,\tau}(\theta_1, \theta_2) = \int_{v=0}^{1} \nabla g_{k,\tau}(\theta_1 + v(\theta_2 - \theta_1))dv$. By the multivariate mean value theorem, we have

$$g_{k,\tau}(\theta_1) - g_{k,\tau}(\theta_2) = \int_{v=0}^{1} \nabla g_{k,\tau}(\theta_1 + v(\theta_2 - \theta_1))dv(\theta_1 - \theta_2) = G_{k,\tau}(\theta_1, \theta_2)(\theta_1 - \theta_2), \quad (16)$$

which implies

$$\|g_{k,\tau}(\theta_1) - g_{k,\tau}(\theta_2)\|_{G_{k,\tau}^{-1}(\theta_1,\theta_2)} = \|\theta_1 - \theta_2\|_{G_{k,\tau}(\theta_1,\theta_2)}.$$

By following the proof steps of Proposition 3 in Goyal & Perivier (2021) with Proposition C.1 in Lee & Oh (2024), we can show that

$$G_{k,\tau}(\theta_1, \theta_2) \succeq \frac{1}{1 + 3\sqrt{2}} H_{k,\tau}(\theta_1) \text{ and } G_{k,\tau}(\theta_1, \theta_2) \succeq \frac{1}{1 + 3\sqrt{2}} H_{k,\tau}(\theta_2).$$

Finally, we have

$$\|\theta_1 - \theta_2\|_{H_{k,\tau}(\theta_1)} \leq (1 + 3\sqrt{2})^{1/2} \|\theta_1 - \theta_2\|_{G_{k,\tau}(\theta_1,\theta_2)}$$
$$= (1 + 3\sqrt{2})^{1/2} \|g_{k,\tau}(\theta_1) - g_{k,\tau}(\theta_2)\|_{G_{k,\tau}^{-1}(\theta_1,\theta_2)}$$
$$\leq (1 + 3\sqrt{2}) \|g_{k,\tau}(\theta_1) - g_{k,\tau}(\theta_2)\|_{H_{k,\tau}^{-1}(\theta_2)},$$

which concludes the proof with $E$.

$\square$

From the above lemma and $E$ with $\delta = 1/T$, with probability at least $1 - (1/T)$, for all $\tau \geq 1$ and $k \in [K]$, we have

$$|z_n^\top(\widehat{\theta}_{k,\tau} - \theta_k^*)| \leq \|z_n\|_{H_{k,\tau}^{-1}(\widehat{\theta}_{k,\tau})} \|\widehat{\theta}_{k,\tau} - \theta_k^*\|_{H_{k,\tau}(\widehat{\theta}_{k,\tau})} \leq \zeta_\tau \|z_n\|_{H_{k,\tau}^{-1}(\widehat{\theta}_{k,\tau})}.$$

In the following proof, with a slight abuse of notation, we define $E = \{|z_n^\top(\widehat{\theta}_{k,\tau} - \theta_k^*)| \leq \zeta_\tau \|z_n\|_{H_{k,\tau}^{-1}(\widehat{\theta}_{k,\tau})} \forall \tau \geq 1, k \in [K], n \in [N]\}$, which holds at least $1 - (1/T)$. We also use $p(n|S, \theta) = \exp(z_n^\top\theta)/(1 + \sum_{m \in S} \exp(z_m^\top\theta))$ with $z_n$ instead of $x_n$.

**Lemma A.13.** *Under $E$, for all $k \in [K]$ and $\tau \in [T]$, for any $S \subset \mathcal{N}_{k,\tau-1}$, we have*

$$0 \leq R_{k,\tau}^{UCB}(S) - R_k(S)$$
$$\leq 13\zeta_\tau^2 \max_{n \in S} \|z_n\|_{H_{k,\tau}^{-1}(\widehat{\theta}_{k,\tau})}^2 + 4\zeta_\tau^2 \max_{n \in S} \|\tilde{z}_{n,k,\tau}\|_{H_{k,\tau}^{-1}(\widehat{\theta}_{k,\tau})}^2 + 2\zeta_\tau \sum_{n \in S} p(n|S, \widehat{\theta}_{k,\tau-1}) \|\tilde{z}_{n,k,\tau}\|_{H_{k,\tau}^{-1}(\widehat{\theta}_{k,\tau})},$$

$$0 \leq R_k(S) - R_{k,\tau}^{LCB}(S)$$
$$\leq 13\zeta_\tau^2 \max_{n \in S} \|z_n\|_{H_{k,\tau}^{-1}(\widehat{\theta}_{k,\tau})}^2 + 4\zeta_\tau^2 \max_{n \in S} \|\tilde{z}_{n,k,\tau}\|_{H_{k,\tau}^{-1}(\widehat{\theta}_{k,\tau})} + 2\zeta_\tau \sum_{n \in S} p(n|S, \widehat{\theta}_{k,\tau-1}) \|\tilde{z}_{n,k,\tau}\|_{H_{k,\tau}^{-1}(\widehat{\theta}_{k,\tau})}.$$

*Proof.* Let $u_{n,k} = z_n^\top\theta_k^*$, $\widehat{u}_{n,k} = z_n^\top\widehat{\theta}_{k,\tau}$, and $\widehat{R}_{k,\tau}(S) = \frac{\sum_{n \in S} r_{n,k} \exp(\widehat{u}_{n,k})}{1 + \sum_{m \in S} \exp(\widehat{u}_{m,k})}$. We also define $u_{n,k} = z_n^\top\theta_k^*$, $\mathbf{u}_k = (u_{n,k} : n \in S)$, $\widehat{\mathbf{u}}_{k,\tau} = (\widehat{u}_{n,k,\tau} : n \in S)$, and $Q(\mathbf{v}) = \sum_{n \in S} \frac{r_{n,k} \exp(v_n)}{1 + \sum_{m \in S} \exp(v_m)}$. Then by a second-order Taylor expansion, we have

$$\left|\widehat{R}_{k,\tau}(S) - R_k(S)\right| = |Q(\widehat{\mathbf{u}}_{k,\tau}) - Q(\mathbf{u}_k)|$$
$$= \left|\nabla Q(\mathbf{u}_k)^\top(\widehat{\mathbf{u}}_{k,\tau} - \mathbf{u}_k)\right| + \left|\frac{1}{2}(\widehat{\mathbf{u}}_{k,\tau} - \mathbf{u}_k)^\top \nabla^2 Q(\bar{\mathbf{u}}_k)(\widehat{\mathbf{u}}_{k,\tau} - \mathbf{u}_k)\right|, (17)$$

where $\bar{\mathbf{u}}_k$ is the convex combination of $\widehat{\mathbf{u}}_{k,\tau}$ and $\mathbf{u}_k$. Let $e_{n,k,\tau} = \widehat{u}_{n,k,\tau} - u_{n,k}$, $e_{n_0,k,\tau} = 0$, $\bar{e}_{n,k,\tau} = e_{n,k,\tau} - \sum_{m \in S \cup \{n_0\}} p(m|S, \theta_k^*)e_{m,k,\tau} = e_{n,k,\tau} - \mathbb{E}_{\theta_k^*}[e_{m,k,\tau}]$, and $\tilde{e}_{n,k,\tau} = e_{n,k,\tau} - $

$\sum_{m \in S \cup \{n_0\}} p(m|S, \widehat{\theta}_{k,\tau}) e_{m,k,\tau} = e_{n,k,\tau} - \mathbb{E}_{\widehat{\theta}_{k,\tau}}[e_{m,k,\tau}]$. Then the first-order term in the above is bounded as

$$\left| \nabla Q(\mathbf{u}_k)^\top (\widehat{\mathbf{u}}_{k,\tau} - \mathbf{u}_k) \right|$$

$$= \left| \frac{\sum_{n \in S} r_{n,k} \exp(u_{n,k})(\widehat{u}_{n,k,\tau} - u_{n,k})}{1 + \sum_{n \in S} \exp(u_{n,k})} - \frac{(\sum_{n \in S} r_{n,k} \exp(u_{n,k}))(\sum_{n \in S} \exp(u_{n,k})(\widehat{u}_{n,k,\tau} - u_{n,k}))}{(1 + \sum_{n \in S} \exp(u_{n,k}))^2} \right|$$

$$= \left| \sum_{n \in S} r_{n,k} p(n|S, \theta_k^*)(\widehat{u}_{n,k,\tau} - u_{n,k}) - \sum_{n,m \in S} r_{m,k} p(n|S, \theta_k^*) p(m|S, \theta_k^*)(\widehat{u}_{n,k,\tau} - u_{n,k}) \right|$$

$$= \left| \sum_{n \in S} r_{n,k} p(n|S, \theta_k^*) \left( (\widehat{u}_{n,k,\tau} - u_{n,k}) - \sum_{m \in S} p(m|S, \theta_k^*)(\widehat{u}_{m,k,\tau} - u_{m,k}) \right) \right|$$

$$\leq \sum_{n \in S} r_{n,k} p(n|S, \theta_k^*) \left| e_{n,k,\tau} - \mathbb{E}_{\theta_k^*}[e_{m,k,\tau}] \right|$$

$$\leq \sum_{n \in S} p(n|S, \theta_k^*) \left| e_{n,k,\tau} - \mathbb{E}_{\theta_k^*}[e_{m,k,\tau}] \right|$$

$$= \sum_{n \in S} p(n|S, \theta_k^*) \left| \bar{e}_{n,k,\tau} \right|$$

$$\leq \sum_{n \in S} p(n|S, \theta_k^*) \left| \bar{e}_{n,k,\tau} - \tilde{e}_{n,k,\tau} \right| + \sum_{n \in S} p(n|S, \theta_k^*) \left| \tilde{e}_{n,k,\tau} \right|$$

For the first term above, we have

$$\sum_{n \in S} p(n|S, \theta_k^*) \left| \bar{e}_{n,k,\tau} - \tilde{e}_{n,k,\tau} \right|$$

$$= \sum_{n \in S} p(n|S, \theta_k^*) \left| \mathbb{E}_{\theta_k^*}[e_{m,k,\tau}] - \mathbb{E}_{\widehat{\theta}_{k,\tau}}[e_{m,k,\tau}] \right|$$

$$= \sum_{n \in S} p(n|S, \theta_k^*) \left| \sum_{m \in S} (p(m|S, \theta_k^*) - p(m|S, \widehat{\theta}_{k,\tau})) e_{m,k,\tau} \right|$$

$$\leq 2\zeta_\tau^2 \sum_{n \in S} p(n|S, \theta_k^*) \|z_n\|_{H_{k,\tau}^{-1}}^2$$

$$\leq 2\zeta_\tau^2 \max_{n \in S} \|z_n\|_{H_{k,\tau}^{-1}}^2,$$

where the first inequality is obtained by using the mean value theorem. Then for the second term, we have

$$\sum_{n \in S} p(n|S, \theta_k^*) |\tilde{e}_{n,k,\tau}| \leq \sum_{n \in S} (p(n|S, \theta_k^*) - p(n|S, \widehat{\theta}_{k,\tau-1})) |\tilde{e}_{n,k,\tau}| + \sum_{n \in S} p(n|S, \widehat{\theta}_{k,\tau-1}) |\tilde{e}_{n,k,\tau}|$$

$$\leq 2\zeta_\tau \max_{n \in S} \|z_n\|_{H_{k,\tau}^{-1}} |(\widehat{\theta}_{k,\tau} - \theta_k^*)^\top (z_n - \mathbb{E}_{\widehat{\theta}_{k,\tau}}[z_n])|$$

$$+ \sum_{n \in S} p(n|S, \widehat{\theta}_{k,\tau-1}) |(\widehat{\theta}_{k,\tau} - \theta_k^*)^\top (z_n - \mathbb{E}_{\widehat{\theta}_{k,\tau}}[z_n])|$$

$$\leq 2\zeta_\tau^2 (\max_{n \in S} \|z_n\|_{H_{k,\tau}^{-1}}^2 + \max_{n \in S} \|\tilde{z}_{n,k,\tau}\|_{H_{k,\tau}^{-1}}^2) + \zeta_\tau \sum_{n \in S} p(n|S, \widehat{\theta}_{k,\tau-1}) \|\tilde{z}_{n,k,\tau}\|_{H_{k,\tau}^{-1}}.$$

From the above inequalities, we have

$$\left| \nabla Q(\mathbf{u}_k)^\top (\widehat{\mathbf{u}}_{k,\tau} - \mathbf{u}_k) \right| \leq 4\zeta_\tau^2 \max_{n \in S} \|z_n\|_{H_{k,\tau}^{-1}}^2 + 2\zeta_\tau^2 \max_{n \in S} \|\tilde{z}_{n,k,\tau}\|_{H_{k,\tau}^{-1}}^2 + \zeta_\tau \sum_{n \in S} p(n|S, \widehat{\theta}_{k,\tau-1}) \|\tilde{z}_{n,k,\tau}\|_{H_{k,\tau}^{-1}}.$$

$$(18)$$

Now we focus on the second-order term which is bounded as

$$
\left| \frac{1}{2} (\widehat{\mathbf{u}}_{k,\tau} - \mathbf{u}_k)^\top \nabla^2 Q(\bar{\mathbf{u}}_k)(\widehat{\mathbf{u}}_{k,\tau} - \mathbf{u}_k) \right|
$$

$$
= \left| \frac{1}{2} \sum_{n,m\in S} (\widehat{u}_{n,k,\tau} - u_{n,k}) \frac{\partial^2 Q(\bar{\mathbf{u}}_k)}{\partial_n \partial_m} (\widehat{u}_{m,k,\tau} - u_{m,k}) \right|
$$

$$
= \left| \frac{1}{2} \sum_{n,m\in S} (\widehat{u}_{n,k,\tau} - u_{n,k}) \frac{\partial^2 Q(\bar{\mathbf{u}}_k)}{\partial_n \partial_m} (\widehat{u}_{m,k,\tau} - u_{m,k}) + \frac{1}{2} \sum_{n,m\in S} (\widehat{u}_{n,k,\tau} - u_{n,k}) \frac{\partial^2 Q(\bar{\mathbf{u}}_k)}{\partial_n \partial_m} (\widehat{u}_{m,k,\tau} - u_{m,k}) \right|
$$

$$
\leq \sum_{n,m\in S} |\widehat{u}_{n,k,\tau} - u_{n,k}| \frac{\exp(\bar{u}_{n,k})}{1 + \sum_{l\in S} \exp(\bar{u}_{l,k})} \frac{\exp(\bar{u}_{m,k})}{1 + \sum_{l\in S} \exp(\bar{u}_{l,k})} |\widehat{u}_{m,k,\tau} - u_{m,k}|
$$

$$
+ \frac{3}{2} \sum_{n\in S} (\widehat{u}_{n,k,\tau} - u_{n,k})^2 \frac{\exp(\bar{u}_{n,k})}{1 + \sum_{l\in S} \exp(\bar{u}_{l,k})}
$$

$$
\leq \frac{5}{2} \sum_{n\in S} (\widehat{u}_{n,k,\tau} - u_{n,k})^2 \frac{\exp(\bar{u}_{n,k})}{1 + \sum_{l\in S} \exp(\bar{u}_{l,k})}
$$

$$
\leq \frac{5}{2} \zeta_\tau^2 \max_{n\in S} \|z_n\|^2_{H_{k,\tau}^{-1}(\widehat{\theta}_{k,\tau})}, \tag{19}
$$

where the first inequality is obtained from Lemma A.22 and the second inequality is obtained from AM-GM inequality. Then from (17), (18), (19), and with the definition of $R_{k,\tau}^{UCB}(S)$ and $R_{k,\tau}^{LCB}(S)$, we can conclude the proof. $\qquad\square$

In the following, similar to Lemma A.9, we provide a lemma for showing that $\mathcal{M}_\tau$ is likely to contain the optimal assortment.

**Lemma A.14.** *Under $E$, $(S_1^*, \ldots, S_K^*) \in \mathcal{M}_{\tau-1}$ for all $\tau \in [T]$.*

*Proof.* Here we use induction for the proof. Suppose that for fixed $\tau$, we have $(S_1^*, \ldots, S_K^*) \in \mathcal{M}_\tau$ for all $k \in [K]$. From $E$, we have $R_{k,\tau+1}^{UCB}(S) \geq R_k(S)$ and $R_{k,\tau+1}^{LCB}(S) \leq R_k(S)$ for any $S \subset [N]$. Then for $k \in [K]$, $n \in S_k^*$, and any $(S_1, .., S_K) \in \mathcal{M}_\tau$, we have

$$
\sum_{l\in[K]} R_{l,\tau+1}^{UCB}(S_{l,\tau+1}^{(n,k)}) \geq \sum_{l\in[K]} R_{l,\tau+1}^{UCB}(S_l^*)
$$

$$
\geq \sum_{l\in[K]} R_l(S_l^*)
$$

$$
\geq \sum_{l\in[K]} R_l(S_l)
$$

$$
\geq \sum_{l\in[K]} R_{l,\tau+1}^{LCB}(S_l), \tag{20}
$$

where the first inequality comes from the elimination condition in the algorithm and $(S_1^*, \ldots S_K^*) \in \mathcal{M}_\tau$, and the third inequality comes from the optimality of $(S_1^*, \ldots, S_K^*)$. This implies that $n \in \mathcal{N}_{k,\tau+1}'$ from the algorithm. Then by following the same statement of (20) for all $n \in S_k^*$ and $k \in [K]$, we have $S_k^* \subseteq \mathcal{N}_{k,\tau+1}'$ for all $k \in [K]$.

Then for $k \in [K]$, $J = S_k^*$, and any $(S_1, .., S_K) \in \mathcal{M}_\tau$, we have

$$
\begin{aligned}
\sum_{l \in [K]} R_{l,\tau+1}^{UCB}(S_{l,\tau+1}^J) &\geq \sum_{l \in [K]} R_{l,\tau+1}^{UCB}(S_l^*) \\
&\geq \sum_{l \in [K]} R_l(S_l^*) \\
&\geq \sum_{l \in [K]} R_l(S_l) \\
&\geq \sum_{l \in [K]} R_{l,\tau+1}^{LCB}(S_l),
\end{aligned}
\tag{21}
$$

where the first inequality comes from the elimination condition in the algorithm and $(S_1^*, \ldots S_K^*) \in \mathcal{M}_\tau$, and the third inequality comes from the optimality of $(S_1^*, \ldots, S_K^*)$. This implies that $J (= S_k^*) \in \mathcal{J}(\mathcal{N}'_{k,\tau+1})$ from the algorithm. Then by following the same statement of (21) for all $k \in [K]$, we have $S_k^* \subseteq \mathcal{N}_{k,\tau+1}$ for all $k \in [K]$, which implies $(S_1^*, \ldots, S_K^*) \in \mathcal{M}_{\tau+1}$. Therefore, with $(S_1^*, \ldots, S_K^*) \in \mathcal{M}_1$, we can conclude the proof from the induction.

$\square$

We define $\bar{V}(\bar{\pi}_{k,\tau}) = \sum_{n \in J \in \mathcal{J}_{k,\tau}} \bar{\pi}_{k,\tau}(n, J) \tilde{z}_{n,k,\tau}(J) \tilde{z}_{n,k,\tau}(J)^\top$ and $\tilde{V}(\tilde{\pi}_{k,\tau}) = \sum_{J \in \mathcal{J}_{k,\tau}} \tilde{\pi}_{k,\tau}(J) \sum_{n \in J} p(n|J, \widehat{\theta}_{k,\tau}) \tilde{z}_{n,k,\tau}(J) \tilde{z}_{n,k,\tau}(J)^\top$. Then we have the following lemma from the G/D-optimal design problem.

**Lemma A.15** (Kiefer-Wolfowitz). *For all $\tau \in [T]$ and $k \in [K]$, we have*

$$
\max_{n \in J \in \mathcal{J}(\mathcal{N}_{k,\tau})} \|\tilde{z}_{n,k,\tau}(J)\|_{(\bar{V}(\bar{\pi}_{k,\tau}) + (\lambda/T_\tau r)I_r)^{-1}}^2 \leq r \text{ and } |supp(\bar{\pi}_{k,\tau})| \leq r(r+1)/2,
$$

$$
\max_{J \in \mathcal{J}(\mathcal{N}_{k,\tau})} \sum_{n \in J} p(n|J, \widehat{\theta}_{k,\tau}) \|\tilde{z}_{n,k,\tau}(J)\|_{(\tilde{V}(\tilde{\pi}_{k,\tau}) + (\lambda/T_\tau r)I_r)^{-1}}^2 \leq r \text{ and } |supp(\tilde{\pi}_{k,\tau})| \leq r(r+1)/2.
$$

*Proof.* This lemma follows by adapting the proof steps of Lemma A.10. To establish the result, we utilize the following:

$$
\begin{aligned}
&\sum_{n \in J \in \mathcal{J}} \bar{\pi}_{k,\tau}(n, J) \|\tilde{z}_{n,k,\tau}(J)\|_{(\bar{V}(\bar{\pi}_{k,\tau}) + (\lambda/T_\tau r)I_r)^{-1}}^2 \\
&= trace(\sum_{n \in J \in \mathcal{J}} \bar{\pi}(n, J) \tilde{z}_{n,k,\tau}(J) \tilde{z}_{n,k,\tau}(J)^\top (\bar{V}(\bar{\pi}_{k,\tau}) + (\lambda/T_\tau r)I_r)^{-1}) \\
&= trace(I_r) - (\lambda/T_r r) trace((\bar{V}(\bar{\pi}_{k,\tau}) + (\lambda/T_\tau r)I_r)^{-1}) \leq r.
\end{aligned}
$$

Similarly, we have:

$$
\begin{aligned}
&\sum_{J \in \mathcal{J}(\mathcal{N}_{k,\tau})} \tilde{\pi}_{k,\tau}(J) \sum_{n \in J} p(n|J, \widehat{\theta}_{k,\tau}) \|\tilde{z}_{n,k,\tau}(J)\|_{(\tilde{V}(\tilde{\pi}_{k,\tau}) + (\lambda/T_\tau r)I_r)^{-1}}^2 \\
&= trace(\sum_J \tilde{\pi}_{k,\tau}(J) \sum_n p(n|J, \widehat{\theta}_{k,\tau}) \tilde{z}_{n,k,\tau}(J) \tilde{z}_{n,k,\tau}(J)^\top (\tilde{V}(\tilde{\pi}_{k,\tau}) + (\lambda/T_\tau r)I_r)^{-1}) \\
&= trace(I_r) - (\lambda/T_\tau r) trace((\tilde{V}(\tilde{\pi}_{k,\tau}) + (\lambda/T_\tau r)I_r)^{-1}) \leq r.
\end{aligned}
$$

The remaining steps are identical to the proof of Lemma A.10. $\square$

From the above Lemmas A.14 and A.8, under $E$, we have

$$
\sum_{l\in[K]} R_l(S_l^*) - \sum_{l\in[K]} R_l(S_{l,\tau}^{(n,k)})
$$

$$
\leq \sum_{l\in[K]} \left[ R_{l,\tau}^{LCB}(S_l^*) + 13\zeta_\tau^2 \max_{m\in S_l^*} \|z_m\|_{H_{l,\tau}^{-1}(\widehat\theta_{l,\tau})}^2 + 4\zeta_\tau^2 \max_{m\in S_l^*} \|\tilde z_{m,l,\tau}\|_{H_{l,\tau}^{-1}(\widehat\theta_{l,\tau})}^2 \right.
$$

$$
\left. + 2\zeta_\tau \sum_{m\in S_l^*} p(m|S_l^*,\widehat\theta_{l,\tau-1}) \|\tilde z_{m,l,\tau}\|_{H_{l,\tau}^{-1}(\widehat\theta_{l,\tau})} \right]
$$

$$
- \sum_{l\in[K]} \left[ R_{l,\tau}^{UCB}(S_{l,\tau}^{(n,k)}) - 13\zeta_\tau^2 \max_{m\in S_{l,\tau}^{(n,k)}} \|z_m\|_{H_{l,\tau}^{-1}(\widehat\theta_{l,\tau})}^2 - 4\zeta_\tau^2 \max_{m\in S_{l,\tau}^{(n,k)}} \|z_m\|_{H_{l,\tau-1}^{-1}(\widehat\theta_{l,\tau-1})}^2 \right.
$$

$$
\left. - 2\zeta_\tau \sum_{m\in S_{l,\tau}^{(n,k)}} p(m|S_{l,\tau}^{(n,k)},\widehat\theta_{l,\tau-1}) \|\tilde z_{m,l,\tau}\|_{H_{l,\tau}^{-1}(\widehat\theta_{l,\tau})} \right]
$$

$$
\lesssim \sum_{l\in[K]} \left[ \zeta_\tau^2 \max_{m\in S_l^*} \|z_m\|_{H_{l,\tau}^{-1}(\widehat\theta_{l,\tau})}^2 + \zeta_\tau^2 \max_{m\in S_l^*} \|\tilde z_{m,l,\tau}\|_{H_{l,\tau}^{-1}(\widehat\theta_{l,\tau})}^2 + \zeta_\tau \sum_{m\in S_l^*} p(m|S_l^*,\widehat\theta_{l,\tau-1}) \|\tilde z_{m,l,\tau}\|_{H_{l,\tau}^{-1}(\widehat\theta_{l,\tau})} \right.
$$

$$
+ \zeta_\tau^2 \max_{m\in S_{l,\tau}^{(n,k)}} \|z_m\|_{H_{l,\tau}^{-1}(\widehat\theta_{l,\tau})}^2 + \zeta_\tau^2 \max_{m\in S_{l,\tau}^{(n,k)}} \|\tilde z_{m,l,\tau}\|_{H_{l,\tau}^{-1}(\widehat\theta_{l,\tau})}^2
$$

$$
\left. + \zeta_\tau \sum_{m\in S_{l,\tau}^{(n,k)}} p(m|S_{l,\tau}^{(n,k)},\widehat\theta_{l,\tau-1}) \|\tilde z_{m,l,\tau}\|_{H_{l,\tau}^{-1}(\widehat\theta_{l,\tau})} \right]
$$

$$
\leq \sum_{l\in[K]} \left[ \zeta_\tau^2 \max_{m\in S_l^*} \|z_m\|_{H_{l,\tau}^{-1}(\widehat\theta_{l,\tau})}^2 + \zeta_\tau^2 \max_{m\in S_l^*} \|\tilde z_{m,l,\tau}\|_{H_{l,\tau}^{-1}(\widehat\theta_{l,\tau})} + \zeta_\tau^2 \max_{m\in S_{l,\tau}^{(n,k)}} \|z_m\|_{H_{l,\tau}^{-1}(\widehat\theta_{l,\tau})}^2 \right.
$$

$$
+ \zeta_\tau^2 \max_{m\in S_{l,\tau}^{(n,k)}} \|\tilde z_{m,l,\tau}\|_{H_{l,\tau}^{-1}(\widehat\theta_{l,\tau})} + \zeta_\tau \sqrt{\sum_{m\in S_l^*} p(m|S_l^*,\widehat\theta_{l,\tau-1})} \sqrt{\sum_{m\in S_l^*} p(m|S_l^*,\widehat\theta_{l,\tau-1}) \|\tilde z_{m,l,\tau}\|_{H_{l,\tau}^{-1}(\widehat\theta_{l,\tau})}^2}
$$

$$
\left. + \zeta_\tau \sqrt{\sum_{m\in S_{l,\tau}^{(n,k)}} p(m|S_{l,\tau}^{(n,k)},\widehat\theta_{l,\tau-1})} \sqrt{\sum_{m\in S_{l,\tau}^{(n,k)}} p(m|S_{l,\tau}^{(n,k)},\widehat\theta_{l,\tau-1}) \|\tilde z_{m,l,\tau}\|_{H_{l,\tau}^{-1}(\widehat\theta_{l,\tau})}^2} \right],
$$

$$
\tag{22}
$$

where the second inequality comes from the fact that $(S_1^*,\ldots,S_K^*) \in \mathcal{M}_{\tau-1}$ and $\max_{(S_1,\ldots,S_K)\in\mathcal{M}_{\tau-1}} \sum_{l\in[K]} R_{l,\tau}^{LCB}(S_l) \leq \sum_{l\in[K]} R_{l,\tau}^{UCB}(S_{l,\tau}^{(n,k)})$ from the algorithm.

Likewise, we also have

$$
\sum_{l\in[K]} R_l(S_l^*) - \sum_{l\in[K]} R_l(S_{l,\tau}^{(J,k)})
$$

$$
\lesssim \sum_{l\in[K]} \Bigg[ \zeta_\tau^2 \max_{m\in S_l^*} \|z_m\|_{H_{l,\tau}^{-1}(\widehat{\theta}_{l,\tau})}^2 + \zeta_\tau^2 \max_{m\in S_l^*} \|\tilde{z}_{m,l,\tau}\|_{H_{l,\tau}^{-1}(\widehat{\theta}_{l,\tau})}^2 + \zeta_\tau^2 \max_{m\in S_{l,\tau}^{(J,k)}} \|z_m\|_{H_{l,\tau}^{-1}(\widehat{\theta}_{l,\tau})}^2
$$

$$
+ \zeta_\tau^2 \max_{m\in S_{l,\tau}^{(J,k)}} \|\tilde{z}_{m,l,\tau}\|_{H_{l,\tau}^{-1}(\widehat{\theta}_{l,\tau})}^2 + \zeta_\tau \sqrt{\sum_{m\in S_l^*} p(m|S_l^*, \widehat{\theta}_{l,\tau-1})} \sqrt{\sum_{m\in S_l^*} p(m|S_l^*, \widehat{\theta}_{l,\tau-1}) \|\tilde{z}_{m,l,\tau}\|_{H_{l,\tau}^{-1}(\widehat{\theta}_{l,\tau})}^2}
$$

$$
+ \zeta_\tau \sqrt{\sum_{m\in S_{l,\tau}^{(n,k)}} p(m|S_{l,\tau}^{(J,k)}, \widehat{\theta}_{l,\tau-1})} \sqrt{\sum_{m\in S_{l,\tau}^{(J,k)}} p(m|S_{l,\tau}^{(J,k)}, \widehat{\theta}_{l,\tau-1}) \|\tilde{z}_{m,l,\tau}\|_{H_{l,\tau}^{-1}(\widehat{\theta}_{l,\tau})}^2} \Bigg]. \tag{23}
$$

We can show that

$$
H_{k,\tau}(\widehat{\theta}_{k,\tau})
$$

$$
= \lambda I_r + \sum_{t\in\mathcal{T}_{k,\tau-1}} \sum_{n\in S_{k,t}} p(n|S_{k,t}, \widehat{\theta}_{k,\tau}) z_n z_n^\top - \sum_{t\in\mathcal{T}_{k,\tau-1}} \sum_{n\in S_{k,t}} \sum_{m\in S_{k,t}} p(n|S_{k,t}, \widehat{\theta}_{k,\tau}) p(m|S_{k,t}, \widehat{\theta}_{k,\tau}) z_n z_m^\top
$$

$$
= \lambda I_r + \sum_{t\in\mathcal{T}_{k,\tau-1}} \sum_{n\in S_{k,t}} p(n|S_{k,t}, \widehat{\theta}_{k,\tau}) z_n z_n^\top - \frac{1}{2} \sum_{t\in\mathcal{T}_{k,\tau-1}} \sum_{n\in S_{k,t}} \sum_{m\in S_{k,t}} p(n|S_{k,t}, \widehat{\theta}_{k,\tau}) p(m|S_{k,t}, \widehat{\theta}_{k,\tau}) (z_n z_m^\top + z_n z_m^\top)
$$

$$
\succeq \lambda I_r + \sum_{t\in\mathcal{T}_{k,\tau-1}} \sum_{n\in S_{k,t}} p(n|S_{k,t}, \widehat{\theta}_{k,\tau}) z_n z_n^\top - \frac{1}{2} \sum_{t\in\mathcal{T}_{k,\tau-1}} \sum_{n\in S_{k,t}} \sum_{m\in S_{k,t}} p(n|S_{k,t}, \widehat{\theta}_{k,\tau}) p(m|S_{k,t}, \widehat{\theta}_{k,\tau}) (z_n z_n^\top + z_m z_m^\top)
$$

$$
= \lambda I_r + \sum_{t\in\mathcal{T}_{k,\tau-1}} \sum_{n\in S_{k,t}} p(n|S_{k,t}, \widehat{\theta}_{k,\tau}) z_n z_n^\top - \sum_{t\in\mathcal{T}_{k,\tau-1}} \sum_{n\in S_{k,t}} \sum_{m\in S_{k,t}} p(n|S_{k,t}, \widehat{\theta}_{k,\tau}) p(m|S_{k,t}, \widehat{\theta}_{k,\tau}) z_n z_n^\top
$$

$$
= \lambda I_r + \sum_{t\in\mathcal{T}_{k,\tau-1}} \sum_{n\in S_{k,t}} p(n|S_{k,t}, \widehat{\theta}_{k,\tau}) \left( 1 - \sum_{m\in S_{k,t}} p(m|S_{k,t}, \widehat{\theta}_{k,\tau}) \right) z_n z_n^\top
$$

$$
= \lambda I_r + \sum_{t\in\mathcal{T}_{k,\tau-1}} \sum_{n\in S_{k,t}} p(n|S_{k,s}, \widehat{\theta}_{k,s}) p(n_0|S_{k,s}, \widehat{\theta}_{k,}) z_n z_n^\top \succeq \lambda I_r + \sum_{t\in\mathcal{T}_{k,\tau-1}} \sum_{n\in S_{k,t}} \kappa z_n z_n^\top
$$

$$
\succeq \lambda I_r + \sum_{n\in\mathcal{N}_{k,\tau-1}} \kappa r \pi_{k,\tau-1}(n) T_{\tau-1} z_n z_n^\top = \kappa T_{\tau-1} r(V(\pi_{k,\tau-1}) + (\lambda/\kappa r T_{\tau-1}) I_r)
$$

$$
\succeq \kappa T_{\tau-1} r(V(\pi_{k,\tau-1}) + (\lambda/r T_{\tau-1}) I_r). \tag{24}
$$

From Lemma A.10 and (24), we also have, for any $n \in \mathcal{N}_{k,\tau}$

$$
\|z_n\|_{H_{k,\tau}^{-1}(\widehat{\theta}_{k,\tau})}^2 = \mathcal{O}\left( \frac{\|z_n\|_{(V(\pi_{k,\tau-1}) + (\lambda/r T_{\tau-1}) I_r)^{-1}}^2}{\kappa r T_{\tau-1}} \right)
$$

$$
= \mathcal{O}\left( \frac{1}{\kappa T_{\tau-1}} \right). \tag{25}
$$

We have

$$H_{k,\tau}(\widehat{\theta}_{k,\tau})$$

$$= \lambda I_r + \sum_{t \in \mathcal{T}_{k,\tau-1}} \sum_{n \in S_{k,t}} p(n|S_{k,t}, \widehat{\theta}_{k,\tau}) z_n z_n^\top - \sum_{t \in \mathcal{T}_{k,\tau-1}} \sum_{n \in S_{k,t}} \sum_{m \in S_{k,t}} p(n|S_{k,t}, \widehat{\theta}_{k,\tau}) p(m|S_{k,t}, \widehat{\theta}_{k,\tau}) z_n z_m^\top$$

$$= \lambda I_r + \sum_{t \in \mathcal{T}_{k,\tau-1}} \mathbb{E}_{\widehat{\theta}_{k,\tau}}[z_n z_n^\top] - \mathbb{E}_{\widehat{\theta}_{k,\tau}}[z_n] \mathbb{E}_{\widehat{\theta}_{k,\tau}}[z_n]^\top$$

$$= \lambda I_r + \sum_{t \in \mathcal{T}_{k,\tau-1}} \mathbb{E}_{\widehat{\theta}_{k,\tau}}[\tilde{z}_{n,k,\tau} \tilde{z}_{n,k,\tau}^\top]$$

$$= \lambda I_r + \sum_{t \in \mathcal{T}_{k,\tau-1}} \sum_{n \in S_{k,t}} p(n|S_{k,t}, \widehat{\theta}_{k,\tau}) \tilde{z}_{n,k,\tau} \tilde{z}_{n,k,\tau}^\top$$

$$\succeq \lambda I_r + \sum_{J \in \mathcal{J}(\mathcal{N}_{k,\tau-1})} \sum_{t \in \mathcal{T}_{J,k,\tau-1}} \sum_{n \in J} p(n|J, \widehat{\theta}_{k,\tau}) \tilde{z}_{n,k,\tau} \tilde{z}_{n,k,\tau}^\top$$

$$\succeq \lambda I_r + \sum_{J \in \mathcal{J}(\mathcal{N}_{k,\tau-1})} r \bar{\pi}_{k,\tau-1}(J) T_{\tau-1} \sum_{n \in J} \kappa \tilde{z}_{n,k,\tau} \tilde{z}_{n,k,\tau}^\top$$

$$\succeq \kappa T_{\tau-1} r \left( \bar{V}(\bar{\pi}_{k,\tau-1}) + (\lambda/T_{\tau-1} r) I_r \right). \tag{26}$$

From Lemma A.15 and (26) with $\mathcal{N}_{k,\tau} \subseteq \mathcal{N}_{k,\tau-1}$, we also have, for any $n \in J \in \mathcal{J}(\mathcal{N}_{k,\tau})$

$$\|\tilde{z}_{n,k,\tau}(J)\|^2_{H_{k,\tau}^{-1}(\widehat{\theta}_{k,\tau})} = \mathcal{O}\left( \frac{\|\tilde{z}_{n,k,\tau}(J))\|^2_{(\bar{V}(\bar{\pi}_{k,\tau-1}) + (\lambda/r T_{\tau-1}) I_r)^{-1}}}{\kappa r T_{\tau-1}} \right)$$

$$= \mathcal{O}\left( \frac{1}{\kappa T_{\tau-1}} \right). \tag{27}$$

We have

$$H_{k,\tau}(\widehat{\theta}_{k,\tau})$$

$$= \lambda I_r + \sum_{t \in \mathcal{T}_{k,\tau-1}} \sum_{n \in S_{k,t}} p(n|S_{k,t}, \widehat{\theta}_{k,\tau}) z_n z_n^\top - \sum_{t \in \mathcal{T}_{k,\tau-1}} \sum_{n \in S_{k,t}} \sum_{m \in S_{k,t}} p(n|S_{k,t}, \widehat{\theta}_{k,\tau}) p(m|S_{k,t}, \widehat{\theta}_{k,\tau}) z_n z_m^\top$$

$$= \lambda I_r + \sum_{t \in \mathcal{T}_{k,\tau-1}} \mathbb{E}_{\widehat{\theta}_{k,\tau}}[z_n z_n^\top] - \mathbb{E}_{\widehat{\theta}_{k,\tau}}[z_n] \mathbb{E}_{\widehat{\theta}_{k,\tau}}[z_n]^\top$$

$$= \lambda I_r + \sum_{t \in \mathcal{T}_{k,\tau-1}} \mathbb{E}_{\widehat{\theta}_{k,\tau}}[\tilde{z}_{n,k,\tau} \tilde{z}_{n,k,\tau}^\top]$$

$$= \lambda I_r + \sum_{t \in \mathcal{T}_{k,\tau-1}} \sum_{n \in S_{k,t}} p(n|S_{k,t}, \widehat{\theta}_{k,\tau}) \tilde{z}_{n,k,\tau} \tilde{z}_{n,k,\tau}^\top$$

$$\succeq \lambda I_r + \sum_{J \in \mathcal{J}(\mathcal{N}_{k,\tau-1})} \sum_{t \in \mathcal{T}_{J,k,\tau-1}} \sum_{n \in J} p(n|J, \widehat{\theta}_{k,\tau}) \tilde{z}_{n,k,\tau} \tilde{z}_{n,k,\tau}^\top$$

$$\succeq \lambda I_r + \sum_{J \in \mathcal{J}(\mathcal{N}_{k,\tau-1})} r \tilde{\pi}_{k,\tau-1}(J) T_{\tau-1} \sum_{n \in J} p(n|J, \widehat{\theta}_{k,\tau}) \tilde{z}_{n,k,\tau} \tilde{z}_{n,k,\tau}^\top$$

$$\succeq \lambda I_r + \sum_{J \in \mathcal{J}(\mathcal{N}_{k,\tau-1})} r \tilde{\pi}_{k,\tau-1}(J) T_{\tau-1} \sum_{n \in J} p(n|J, \widehat{\theta}_{k,\tau-1}) \tilde{z}_{n,k,\tau} \tilde{z}_{n,k,\tau}^\top$$

$$\qquad - 2\zeta_\tau \sum_{J \in \mathcal{J}(\mathcal{N}_{k,\tau-1})} r \tilde{\pi}_{k,\tau-1}(J) T_{\tau-1} \max_{n \in J}(\|z_n\|_{H_{k,\tau}^{-1}(\widehat{\theta}_{k,\tau})} + \|z_n\|_{H_{k,\tau-1}^{-1}(\widehat{\theta}_{k,\tau-1})}) \tilde{z}_{n,k,\tau} \tilde{z}_{n,k,\tau}^\top$$

$$= T_{\tau-1} r \left( \tilde{V}(\tilde{\pi}_{k,\tau-1}) + (\lambda/T_{\tau-1} r) I_r \right.$$

$$\left. - 2\zeta_\tau \sum_{J \in \mathcal{J}(\mathcal{N}_{k,\tau-1})} \tilde{\pi}_{k,\tau-1}(J) \max_{n \in J}(\|z_n\|_{H_{k,\tau}^{-1}(\widehat{\theta}_{k,\tau})} + \|z_n\|_{H_{k,\tau-1}^{-1}(\widehat{\theta}_{k,\tau-1})}) \tilde{z}_{n,k,\tau} \tilde{z}_{n,k,\tau}^\top \right),$$

$$\tag{28}$$

where the last inequality is obtained from, using the mean value theorem,

$$
\sum_{n \in J} (p(n|J, \widehat{\theta}_{k,\tau}) - p(n|J, \widehat{\theta}_{k,\tau-1}) \tilde{z}_{n,k,\tau} \tilde{z}_{n,k,\tau}^\top
$$

$$
= \sum_{n \in J} (p(n|J, \widehat{\theta}_{k,\tau}) - p(n|J, \theta_k^*) + p(n|J, \theta_k^*) - p(n|J, \widehat{\theta}_{k,\tau-1})) \tilde{z}_{n,k,\tau} \tilde{z}_{n,k,\tau}^\top
$$

$$
\succeq -2\zeta_\tau (\max_{n \in J} \|z_n\|_{H_{k,\tau}^{-1}(\widehat{\theta}_{k,\tau})} + \max_{n \in J} \|z_n\|_{H_{k,\tau-1}^{-1}(\widehat{\theta}_{k,\tau-1})}) \tilde{z}_{n,k,\tau} \tilde{z}_{n,k,\tau}^\top. \tag{29}
$$

Let $B = 2\zeta_\tau \sum_{J \in \mathcal{J}(\mathcal{N}_{k,\tau-1})} \tilde{\pi}_{k,\tau-1}(J) \max_{n \in J} (\|z_n\|_{H_{k,\tau}^{-1}(\widehat{\theta}_{k,\tau})} + \|z_n\|_{H_{k,\tau-1}^{-1}(\widehat{\theta}_{k,\tau-1})}) \tilde{z}_{n,k,\tau} \tilde{z}_{n,k,\tau}^\top$
and we have $B \preceq 4\zeta_\tau \sqrt{\frac{1}{\kappa T_{\tau-2}}} \sum_{J \in \mathcal{J}(\mathcal{N}_{k,\tau-1})} \tilde{\pi}_{k,\tau-1}(J) \max_{n \in J} \tilde{z}_{n,k,\tau} \tilde{z}_{n,k,\tau}^\top$ from (25). Then for $\tau \geq 3$, we have

$$
\tilde{V}(\tilde{\pi}_{k,\tau-1}) - B
$$

$$
\succeq \frac{1}{2} \tilde{V}(\tilde{\pi}_{k,\tau-1}) + \frac{1}{2} \tilde{V}(\tilde{\pi}_{k,\tau-1}) - B
$$

$$
\succeq \frac{1}{2} \tilde{V}(\tilde{\pi}_{k,\tau-1}) + \frac{1}{2} \sum_{J \in \mathcal{J}(\mathcal{N}_{k,\tau})} \tilde{\pi}_{k,\tau}(J) \sum_{n \in J} \kappa \tilde{z}_{n,k,\tau} \tilde{z}_{n,k,\tau}^\top - 4\zeta_\tau \sqrt{\frac{1}{\kappa T_{\tau-2}}} \sum_{J \in \mathcal{J}(\mathcal{N}_{k,\tau-1})} \tilde{\pi}_{k,\tau-1}(J) \max_{n \in J} \tilde{z}_{n,k,\tau} \tilde{z}_{n,k,\tau}^\top
$$

$$
\succeq \frac{1}{2} \tilde{V}(\tilde{\pi}_{k,\tau-1}), \tag{30}
$$

where the last inequality is obtained from $\frac{1}{2}\kappa \geq 4\zeta_\tau \sqrt{\frac{1}{\kappa T_{\tau-2}}}$ because $T_{\tau-2} \geq \min\{T_1, \eta_T\}$ with large enough $T$ such that $T \geq \max\{\frac{r^3 K}{\kappa^6} \log^4(KTL), \exp(\frac{r}{\kappa^3})\}$.

Then, we have

$$
\|\tilde{z}_{n,k,\tau}\|_{H_{k,\tau}^{-1}(\widehat{\theta}_{k,\tau})}^2 \leq rT_{\tau-1} \|\tilde{z}_{n,k,\tau}\|_{(\tilde{V}(\tilde{\pi}_{k,\tau-1}) + (\lambda/T_{\tau-1}r)I_r - B)^{-1}}^2
$$

$$
\leq rT_{\tau-1} \|\tilde{z}_{n,k,\tau}\|_{(\frac{1}{2}\tilde{V}(\tilde{\pi}_{k,\tau-1}) + \frac{1}{2}(\lambda/T_{\tau-1}r)I_r)^{-1}}^2
$$

$$
\leq 2rT_{\tau-1} \|\tilde{z}_{n,k,\tau}\|_{(\tilde{V}(\tilde{\pi}_{k,\tau-1}) + (\lambda/T_{\tau-1}r)I_r)^{-1}}^2.
$$

Then from the above, Lemma A.15, and (28) with $\mathcal{N}_{k,\tau} \subseteq \mathcal{N}_{k,\tau-1}$, we have, for any $J \in \mathcal{J}(\mathcal{N}_{k,\tau})$

$$
\sum_{n \in J} p(n|J, \widehat{\theta}_{k,\tau-1}) \|\tilde{z}_{n,k,\tau}\|_{H_{k,\tau}^{-1}(\widehat{\theta}_{k,\tau})}^2
$$

$$
= \mathcal{O}\left( \frac{\sum_{n \in J} p(n|J, \widehat{\theta}_{k,\tau-1}) \|\tilde{z}_{n,k,\tau}\|_{(\tilde{V}(\tilde{\pi}_{k,\tau-1}) + (\lambda/T_{\tau-1}r)I_r)^{-1}}^2}{rT_{\tau-1}} \right)
$$

$$
= \mathcal{O}\left( \frac{1}{T_{\tau-1}} \right). \tag{31}
$$

Therefore under $E$, from (22), (23), (25),(27), and (31), we have the following.

For $t \in \bigcup_{n \in \mathcal{N}_{k,\tau}, k \in [K]} \mathcal{T}_{n,k,\tau} \bigcup_{J \in \mathcal{J}(\mathcal{N}_{k,\tau}), k \in [K]} \mathcal{T}_{J,k,\tau} \bigcup_{n \in J \in \mathcal{J}(\mathcal{N}_{k,\tau}), k \in [K]} \mathcal{T}_{n,J,k,\tau}$,

$$
\sum_{k \in [K]} (R_k(S_k^*) - R_k(S_{k,t})) = \mathcal{O}\left( K\left( \sqrt{\frac{r}{T_{\tau-1}}} + \frac{r}{T_{\tau-1}\kappa} \right) \right).
$$

For the regret bound, we have

$$\mathbb{E}\left[\sum_{t\in[T]}\sum_{k\in[K]} R_k(S_k^*) - R_t(S_{k,t})\right]$$

$$\leq \mathbb{E}\left[\sum_{t\in[T]}\sum_{k\in[K]}\left(R_k(S_k^*) - R_k(S_{k,t})\right)\mathbf{1}(E)\right] + \mathbb{E}\left[\sum_{t\in[T]}\sum_{k\in[K]}\left(R_k(S_k^*) - R_k(S_{k,t})\right)\mathbf{1}(E^c)\right]$$

$$= \tilde{\mathcal{O}}\left(K\sum_{\tau=3}^{\tau_T}\sum_{k\in[K]}\left(\sum_{J\in\mathcal{J}(\mathcal{N}_{k,\tau})}|\mathcal{T}_{J,k,\tau}| + \sum_{n\in\mathcal{N}_{k,\tau}}|\mathcal{T}_{n,k,\tau}|\right)\left(\sqrt{\frac{r}{T_{\tau-1}}} + \frac{r}{T_{\tau-1}\kappa}\right)\right) + \tilde{\mathcal{O}}(rK\eta_T) + \mathcal{O}(K)$$

$$= \tilde{\mathcal{O}}\left(K\sum_{\tau=3}^{\tau_T}\sum_{k\in[K]}\left(\sum_{J\in\mathcal{J}(\mathcal{N}_{k,\tau})}|\mathcal{T}_{J,k,\tau}| + \sum_{n\in\mathcal{N}_{k,\tau}}|\mathcal{T}_{n,k,\tau}|\right)\left(\sqrt{\frac{r}{T_{\tau-1}}} + \frac{r}{T_{\tau-1}\kappa}\right)\right) + \tilde{\mathcal{O}}(rK\eta_T)$$

$$= \tilde{\mathcal{O}}\left(K\sum_{\tau=3}^{\tau_T}\sum_{k\in[K]}(rT_\tau + |Supp(\pi_{k,\tau})| + |Supp(\tilde{\pi}_{k,\tau})|)\left(\sqrt{\frac{r}{T_{\tau-1}}} + \frac{r}{T_{\tau-1}\kappa}\right)\right) + \tilde{\mathcal{O}}(rK\eta_T)$$

$$= \tilde{\mathcal{O}}\left(K^2\sum_{\tau=3}^{\tau_T}\left(r^{3/2}\eta_T + r^2\frac{1}{\kappa\sqrt{T_{\tau-1}}}\eta_T\right)\right)$$

$$= \tilde{\mathcal{O}}\left(K^2 r^{3/2}\eta_T\right)$$

$$= \tilde{\mathcal{O}}\left(r^{3/2}K^2(T/rK)^{\frac{1}{2(1-2^{-M})}}\right), \tag{32}$$

where the third last equality comes from Lemma A.10 and the second last equality comes from $\tau_T \leq M = \tilde{O}(1)$ and $T_{\tau-1} \geq \eta_T$ for $\tau \geq 3$.

### A.8 APPROXIMATION ORACLE

Here we discuss the combinatorial optimization in our algorithm. We can utilize an $\alpha$-approximation oracle with $0 \leq \alpha \leq 1$, first introduced in Kakade et al. (2007). Instead of obtaining the exact optimal assortment solution, the $\alpha$-approximation oracle, denoted by $\mathbb{O}^\alpha$, outputs $\{S_k^\alpha\}_{k\in[K]}$ satisfying $\sum_{k\in[K]} f_k(S_k^\alpha) \geq \max_{\{S_k\}_{k\in[K]}\in\mathcal{M}}\sum_{k\in[K]}\alpha f_k(S_k)$.

We introduce an algorithm (Algorithm 5 in Appendix A.8) by modifying Algorithm 1 to incorporate $\alpha$-approximation oracles for the optimization. Due to the redundancy, we explain only the distinct parts of the algorithm here. (Approximation oracles can also be applied to Algorithm 2 similarly, but we omit it in this discussion.) For testing the assignment $(n, k)$, the algorithm constructs assortment $\{S_{l,\tau}^{\alpha,(n,k)}\}_{l\in[K]}$ (where $n \in S_{k,\tau}^{\alpha,(n,k)}$) in an optimistic view with an $\alpha$-approximation oracle to resolve computation issue as follows. We define an approximation oracle $\mathbb{O}_{UCB}^{\alpha,(n,k)}$ which outputs $\{S_{l,\tau}^{\alpha,(n,k)}\}_{l\in[K]}$ satisfying

$$\max_{\{S_l\}_{l\in[K]}\in\mathcal{M}_{\tau-1}:n\in S_k}\sum_{l\in[K]}\alpha R_{l,\tau}^{UCB}(S_l) \leq \sum_{l\in[K]} R_{l,\tau}^{UCB}(S_{l,\tau}^{\alpha,(n,k)}), \tag{33}$$

which replaces Line 5 in Algorithm 1. For the elimination procedure, we define another $\beta$-approximation oracle, denoted by $\mathbb{O}_{LCB}^\beta$, which outputs $\{S_{l,\tau}^\beta\}_{l\in[K]}$ satisfying

$$\max_{\{S_l\}_{l\in[K]}\in\mathcal{M}_{\tau-1}}\sum_{l\in[K]}\beta R_{l,\tau}^{LCB}(S_l) \leq \sum_{l\in[K]} R_{l,\tau}^{LCB}(S_{l,\tau}^\beta). \tag{34}$$

Then it updates $\mathcal{N}_{k,\tau}$ by eliminating $n \in \mathcal{N}_{k,\tau-1}$ which satisfies the elimination condition of

$$\sum_{l\in[K]}\alpha R_{l,\tau}^{LCB}(S_{l,\tau}^\beta) > \sum_{l\in[K]} R_{l,\tau}^{UCB}(S_{l,\tau}^{\alpha,(n,k)}), \tag{35}$$

which replaces Line 6 in Algorithm 1. We note that the algorithm utilizes the two different types of approximation oracles, $\mathbb{O}_{UCB}^{\alpha,(n,k)}$ and $\mathbb{O}_{LCB}^{\beta}$. Then the algorithm achieves a regret bound for $\gamma$-regret defined as $\mathcal{R}^{\gamma}(T) = \mathbb{E}[\sum_{t\in[T]}\sum_{k\in[K]}\gamma R_k(S_k^*) - R_k(S_{k,t})]$ in the following theorem.

**Theorem A.16.** *Algorithm 5 with $M = O(\log(T))$ achieves a regret bound with $\gamma = \alpha\beta$ as*

$$\mathcal{R}^{\gamma}(T) = \tilde{\mathcal{O}}\left(\frac{1}{\kappa}K^{\frac{3}{2}}\sqrt{rT}\left(\frac{T}{rK}\right)^{\frac{1}{2(2^{M}-1)}}\right).$$

*Proof.* The proof is provided in Appendix A.8.2. □

### A.8.1 $\alpha$-APPROXIMATED ALGORITHM (ALGORITHM 5)

---
**Algorithm 5** Batched Stochastic Matching Bandit with $\beta$-Approximation Oracle

---
**Input:** $\beta, \kappa, M \geq 1$; **Init:** $t \leftarrow 1, T_1 \leftarrow \eta_T$
43 Compute SVD of $X = U\Sigma V^{\top}$ and obtain $U_r = [u_1, \ldots, u_r]$; Construct $z_n \leftarrow U_r^{\top}x_n$ for $n \in [N]$
44 **for** $\tau = 1, 2\ldots$ **do**
45     **for** $k \in [K]$ **do**
        // Estimation
46         $\widehat{\theta}_{k,\tau} \leftarrow \mathrm{argmin}_{\theta\in\mathbb{R}^r} l_{k,\tau}(\theta)$ with (2) where $\mathcal{T}_{k,\tau-1} = \mathcal{T}_{k,\tau-1}^{(1)} \cup \mathcal{T}_{k,\tau-1}^{(2)}$ and $\mathcal{T}_{k,\tau-1}^{(2)} = \bigcup_{n\in\mathcal{N}_{k,\tau-1}}\mathcal{T}_{n,k,\tau-1}^{(2)}$
        // Assortments Construction
47         $\{S_{l,\tau}^{\alpha,(n,k)}\}_{l\in[K]} \leftarrow \mathbb{O}_{UCB}^{\alpha,(n,k)}$ from (33) for all $n \in \mathcal{N}_{k,\tau-1}$ with (3)
        // Elimination
48         $\{S_{l,\tau}^{\beta}\}_{l\in[K]} \leftarrow \mathbb{O}_{LCB}^{\beta}$ from (34)
49         $\mathcal{N}_{k,\tau} \leftarrow \{n \in \mathcal{N}_{k,\tau} : \sum_{l\in[K]}\alpha R_{l,\tau}^{LCB}(S_{l,\tau}^{\beta}) \leq \sum_{l\in[K]}R_{l,\tau}^{UCB}(S_{l,\tau}^{\alpha,(n,k)})\}$ for $k \in [K]$
        // G/D-optimal design
50         $\pi_{k,\tau} \leftarrow \mathrm{argmax}_{\pi\in\mathcal{P}(\mathcal{N}_{k,\tau})}\log\det(\sum_{n\in\mathcal{N}_{k,\tau}}\pi_{k,\tau}(n)z_nz_n^{\top} + (1/rT_{\tau})I_r)$
        // Exploration
51         Run Warm-up (Algorithm 4) over time steps in $\mathcal{T}_{k,\tau}^{(1)}$ (defined in Algorithm 4)
52         **for** $n \in \mathcal{N}_{k,\tau}$ **do**
53             $t_{n,k} \leftarrow t, \mathcal{T}_{n,k,\tau}^{(2)} \leftarrow [t_{n,k}, t_{n,k} + \lceil r\pi_{k,\tau}(n)T_{\tau}\rceil - 1]$
54             **while** $t \in \mathcal{T}_{n,k,\tau}^{(2)}$ **do**
55                 Offer $\{S_{l,t}\}_{l\in[K]} = \{S_{l,\tau}^{(n,k)}\}_{l\in[K]}$ and observe feedback $y_{m,t} \in \{0,1\}$ for all $m \in S_{l,t}$ and $l \in [K]$
56                 $t \leftarrow t + 1$
57     $\mathcal{M}_{\tau} \leftarrow \{\{S_k\}_{k\in[K]} : S_k \subset \mathcal{N}_{k,\tau}, |S_k| \leq L \,\forall k \in [K], S_k \cap S_l = \emptyset \,\forall k \neq l\}; T_{\tau+1} \leftarrow \eta_T\sqrt{T_{\tau}}$

---

### A.8.2 PROOF OF THEOREM A.16

In this proof, we provide only the parts that are different from the proof of Theorem 5.2.

**Lemma A.17.** *Under $E$, $(S_1^*, \ldots, S_K^*) \in \mathcal{M}_{\tau-1}$ for all $\tau \in [T]$.*

*Proof.* Here we use induction for the proof. Suppose that for fixed $\tau$, we have $(S_1^*, \ldots, S_K^*) \in \mathcal{M}_{\tau}$ for all $k \in [K]$. Recall that $\beta_T = (C_1/\kappa)\sqrt{\log(TKN)}$. From Lemma A.8, we have $R_{k,\tau+1}^{UCB}(S) \geq R_k(S)$ and $R_{k,\tau+1}^{LCB}(S) \leq R_k(S)$ for any $S \subset [N]$. Then for $k \in [K]$, $n \in S_k^*$, and any $(S_1, .., S_K) \in$

$\mathcal{M}_\tau$, we have

$$
\begin{aligned}
\sum_{l\in[K]} R^{UCB}_{l,\tau+1}(S^{\alpha,(n,k)}_{l,\tau+1}) &\geq \max_{\{S_k\}_{k\in[K]}\in\mathcal{M}_\tau : n\in S_k} \sum_{l\in[K]} \alpha R^{UCB}_{l,\tau+1}(S_l) \\
&\geq \sum_{l\in[K]} \alpha R^{UCB}_{l,\tau+1}(S^*_l) \\
&\geq \sum_{l\in[K]} \alpha R_l(S^*_l) \\
&\geq \sum_{l\in[K]} \alpha R_l(S^\beta_{l,\tau+1}) \\
&\geq \sum_{l\in[K]} \alpha R^{LCB}_{l,\tau+1}(S^\beta_{l,\tau+1}),
\end{aligned}
\tag{36}
$$

where the first inequality comes from (33), the second one comes from $(S^*_1,\dots S^*_K)\in\mathcal{M}_\tau$, and the firth one comes from the optimality of $(S^*_1,\dots,S^*_K)$. This implies that $n\in\mathcal{N}_{k,\tau+1}$ from the algorithm. Then by following the same statement of (36) for all $n\in S^*_k$ and $k\in[K]$, we have $S^*_k\subset\mathcal{N}_{k,\tau+1}$ for all $k\in[K]$, which implies $(S^*_1,\dots,S^*_K)\in\mathcal{M}_{\tau+1}$. Therefore, with $(S^*_1,\dots,S^*_K)\in\mathcal{M}_1$, we can conclude the proof from the induction. $\qquad\square$

From Lemmas A.17 and A.8, under $E$, we have

$$
\begin{aligned}
\sum_{l\in[K]} \alpha\beta R_l(S^*_l) - \sum_{l\in[K]} R_l(S^{\alpha,(n,k)}_{l,\tau}) &\leq \sum_{l\in[K]} \alpha\beta R^{LCB}_{l,\tau}(S^*_l) + 4\beta_T \max_{m\in S^*_l} \|z_m\|_{V^{-1}_{l,\tau}} \\
&\quad - \sum_{l\in[K]} R^{UCB}_{l,\tau}(S^{\alpha,(n,k)}_{l,\tau}) + 4\beta_T \max_{m\in S^{(n,k)}_{l,\tau}} \|z_m\|_{V^{-1}_{l,\tau}} \\
&\leq \sum_{l\in[K]} \alpha R^{LCB}_{l,\tau}(S^\beta_{l,\tau}) + 4\beta_T \max_{m\in S^*_l} \|z_m\|_{V^{-1}_{l,\tau}} \\
&\quad - \sum_{l\in[K]} R^{UCB}_{l,\tau}(S^{\alpha,(n,k)}_{l,\tau}) + 4\beta_T \max_{m\in S^{(n,k)}_{l,\tau}} \|z_m\|_{V^{-1}_{l,\tau}} \\
&\leq 4\beta_T \sum_{l\in[K]} \big( \max_{m\in S^*_l} \|z_m\|_{V^{-1}_{l,\tau}} + \max_{m\in S^{(n,k)}_{l,\tau}} \|z_m\|_{V^{-1}_{l,\tau-1}} \big),
\end{aligned}
$$

$$\tag{37}$$

where the second inequality comes from (34) and last inequality comes from the fact that $(S^*_1,\dots,S^*_K)\in\mathcal{M}_{\tau-1}$ and $\sum_{l\in[K]} \alpha R^{LCB}_{l,\tau}(S^\beta_{l,\tau}) \leq \sum_{l\in[K]} R^{UCB}_{l,\tau}(S^{\alpha,(n,k)}_{l,\tau})$ from the algorithm. Then, by following the proof in Theorem 1, we can conclude the proof.

## A.9 PROOF OF LEMMAS

### A.9.1 PROOF OF LEMMA A.3

For the poof, we follow the proof steps in (Bounding the Prediction Error) Oh & Iyengar (2021). We define

$$
H_{k,\tau}(\theta) = \sum_{t\in\mathcal{T}_{k,\tau}} \left( \sum_{n\in S_{k,t}} p(n|S_{k,t},\theta) z_n z_n^\top - \sum_{n\in S_{k,t}} \sum_{m\in S_{k,t}} p(n|S_{k,t},\theta) p(m|S_{k,t},\theta) z_n z_m^\top \right) + I_r.
$$

We note that $g_{k,\tau}(\theta_1) - g_{k,\tau}(\theta_2) = \sum_{t \in \mathcal{T}_{k,\tau}} \sum_{n \in S_{k,t}} (p(n, |S_{k,t}, \theta_1) - p(n, |S_{k,t}, \theta_2))z_n + (\theta_1 - \theta_2)$. Then from the mean value theorem, there exists $\bar{\theta} = c\theta_1 + (1-c)\theta_2$ with some $c \in (0, 1)$ such that

$$g_{k,\tau}(\theta_1) - g_{k,\tau}(\theta_2)$$

$$= \nabla_\theta g_{k,\tau}(\theta)\big|_{\theta = \bar{\theta}}(\theta_1 - \theta_2)$$

$$= \left( \sum_{t \in \mathcal{T}_{k,\tau}} \left( \sum_{n \in S_{k,t}} p(n|S_{k,t}, \bar{\theta})z_n z_n^\top - \sum_{n \in S_{k,t}} \sum_{m \in S_{k,t}} p(n|S_{k,t}, \bar{\theta})p(m|S_{k,t}, \bar{\theta})z_n z_m^\top \right) + I_r \right) (\theta_1 - \theta_2)$$

$$= H_{k,\tau}(\bar{\theta})(\theta_1 - \theta_2) \tag{38}$$

We define $L_{k,\tau} = H_{k,\tau}(\theta_k^*)$ and $E_{k,\tau} = H_{k,\tau}(\bar{\theta}_k) - H_{k,\tau}(\theta_k^*)$ where $\bar{\theta}_k = c\theta_k^* + (1-c)\widehat{\theta}_{k,\tau}$ for some constant $c \in (0, 1)$.

From (38), we have $g_{k,\tau}(\widehat{\theta}_{k,\tau}) - g_{k,\tau}(\theta_k^*) = (L_{k,\tau} + E_{k,\tau})(\widehat{\theta}_{k,\tau} - \theta_k^*)$. Then, for any $z \in \mathbb{R}^r$, we have

$$z^\top(\widehat{\theta}_{k,\tau} - \theta_k^*) = z^\top(L_{k,\tau} + E_{k,\tau})^{-1}(g_{k,\tau}(\widehat{\theta}_{k,\tau}) - g_{k,\tau}(\theta_k^*))$$

$$= z^\top L_{k,\tau}^{-1}(g_{k,\tau}(\widehat{\theta}_{k,\tau}) - g_{k,\tau}(\theta_k^*)) - z^\top L_{k,\tau}^{-1} E_{k,\tau}(L_{k,\tau} + E_{k,\tau})^{-1}(g_{k,\tau}(\widehat{\theta}_{k,\tau}) - g_{k,\tau}(\theta_k^*)). \tag{39}$$

For obtaining a bound for $|z^\top(\widehat{\theta}_{k,\tau} - \theta_k^*)|$, we analyze the two terms in (39). We first provide a bound for $|z^\top L_{k,\tau}^{-1}(g_{k,\tau}(\widehat{\theta}_{k,\tau}) - g_{k,\tau}(\theta_k^*))|$. Let $\epsilon_{n,t} = y_{n,t} - p(n|S_{k,t}, \theta_k^*)$ for $n \in S_{k,t}$. Since $\widehat{\theta}_{k,\tau}$ is the solution from MLE such that $\sum_{t \in \mathcal{T}_{k,\tau}} \sum_{n \in S_{k,t}} (p(n|S_{k,t}, \widehat{\theta}_{k,\tau}) - y_{n,k,\tau})z_n = 0$, we have

$$g_{k,\tau}(\widehat{\theta}_{k,\tau}) - g_{k,\tau}(\theta_k^*)$$

$$= \sum_{t \in \mathcal{T}_{k,\tau}} \sum_{n \in S_{k,t}} \left( p(n|S_{k,t}, \widehat{\theta}_{k,\tau}) - p(n|S_{k,t}, \theta_k^*) \right) z_n + (\widehat{\theta}_{k,\tau} - \theta_k^*)$$

$$= \sum_{t \in \mathcal{T}_{k,\tau}} \sum_{n \in S_{k,t}} \left( p(n|S_{k,t}, \widehat{\theta}_{k,\tau}) - y_{n,k,t} \right) z_n + \widehat{\theta}_{k,\tau} + \sum_{t \in \mathcal{T}_{k,\tau}} \sum_{n \in S_{k,t}} \left( y_{n,k,\tau} - p(n|S_{k,t}, \theta_k^*) \right) z_n - \theta_k^*$$

$$= 0 + \sum_{t \in \mathcal{T}_{k,\tau}} \sum_{n \in S_{k,t}} \epsilon_{n,t} z_n - \theta_k^* \tag{40}$$

We define

$$Z_{k,t} = [z_n : n \in S_{k,t}]^\top \in \mathbb{R}^{|S_{k,t}| \times r} \text{ for } t \in \mathcal{T}_{k,\tau},$$

$$D_{k,\tau} = [Z_{k,t} : t \in \mathcal{T}_{k,\tau}]^\top \in \mathbb{R}^{(\sum_{t \in \mathcal{T}_{k,\tau}} |S_{k,t}|) \times r},$$

$$\mathcal{E}_{k,t} = [\epsilon_{n,t} : n \in S_{k,t}]^\top \in \mathbb{R}^{|S_{k,t}|}.$$

Then using Hoeffding inequality, we have

$$\mathbb{P}(|z^\top L_{k,\tau}^{-1}(g_{k,\tau}(\widehat{\theta}_{k,\tau}) - g_{k,\tau}(\theta_k^*))| \geq \nu) \leq \mathbb{P}\left( \left| \sum_{t \in \mathcal{T}_{k,\tau}} z^\top L_{k,\tau}^{-1} Z_{k,t}^\top \mathcal{E}_{k,t} \right| \geq \nu - |z^\top L_{k,\tau}^{-1} \theta_k^*| \right)$$

$$\leq \mathbb{P}\left( \left| \sum_{t \in \mathcal{T}_{k,\tau}} z^\top L_{k,\tau}^{-1} Z_{k,t}^\top \mathcal{E}_{k,t} \right| \geq \nu - 1 \right)$$

$$\leq 2 \exp\left( -\frac{2(\nu - 1)^2}{\sum_{t \in \mathcal{T}_{k,\tau}} (2\sqrt{2}\|z^\top L_{k,\tau}^{-1} Z_{k,t}^\top\|_2)^2} \right)$$

$$= 2 \exp\left( -\frac{(\nu - 1)^2}{4\|z^\top L_{k,\tau}^{-1} D_{k,\tau}^\top\|_2^2} \right)$$

$$\leq 2 \exp\left( -\frac{\kappa^2(\nu - 1)^2}{4\|z\|_{V_{k,\tau}^{-1}}^2} \right),$$

$$\tag{41}$$

where the last inequality is obtained from the fact that

$$L_{k,\tau} = \sum_{t \in \mathcal{T}_{k,\tau}} \left( \sum_{n \in S_{k,t}} p(n|S_{k,t}, \theta_k^*) z_n z_n^\top - \sum_{n \in S_{k,t}} \sum_{m \in S_{k,t}} p(n|S_{k,t}, \theta_k^*) p(m|S_{k,t}, \theta_k^*) z_n z_m^\top \right)$$

$$= \sum_{t \in \mathcal{T}_{k,\tau}} \left( \sum_{n \in S_{k,t}} p(n|S_{k,t}, \theta_k^*) z_n z_n^\top - \frac{1}{2} \sum_{n \in S_{k,t}} \sum_{m \in S_{k,t}} p(n|S_{k,t}, \theta_k^*) p(m|S_{k,t}, \theta_k^*) (z_n z_m^\top + z_m z_n^\top) \right)$$

$$\succeq \sum_{t \in \mathcal{T}_{k,\tau}} \left( \sum_{n \in S_{k,t}} p(n|S_{k,t}, \theta_k^*) z_n z_n^\top - \frac{1}{2} \sum_{n \in S_{k,t}} \sum_{m \in S_{k,t}} p(n|S_{k,t}, \theta_k^*) p(m|S_{k,t}, \theta_k^*) (z_n z_n^\top + z_m z_m^\top) \right)$$

$$= \sum_{t \in \mathcal{T}_{k,\tau}} \left( \sum_{n \in S_{k,t}} p(n|S_{k,t}, \theta_k^*) z_n z_n^\top - \sum_{n \in S_{k,t}} \sum_{m \in S_{k,t}} p(n|S_{k,t}, \theta_k^*) p(m|S_{k,t}, \theta_k^*) z_n z_n^\top \right)$$

$$= \sum_{t \in \mathcal{T}_{k,\tau}} \left( \sum_{n \in S_{k,t}} p(n|S_{k,t}, \theta_k^*) p(n_0|S_{k,t}, \theta_k^*) z_n z_n^\top \right)$$

$$\succeq \kappa D_\tau^\top D_\tau (= \kappa V_{k,\tau}),$$

where the first inequality is obtained from $(z_n - z_m)(z_n - z_m)^\top = z_n z_n^\top + z_m z_m^\top - z_n z_m^\top - z_m z_n^\top \succeq 0$.

Then from (41) using $\nu = (2/\kappa)\sqrt{\log(2TKN/\delta)}\|z\|_{V_{k,\tau}^{-1}} + 1$ and the union bound, with probability at least $1 - \delta$, for all $\tau \in [T], k \in [K]$, we have

$$|z^\top L_{k,\tau}^{-1}(g_{k,\tau}(\widehat{\theta}_{k,\tau}) - g_{k,\tau}(\theta_k^*))| \leq \frac{3\sqrt{\log(TKN/\delta)}}{\kappa} \|z\|_{V_{k,\tau}^{-1}}. \tag{42}$$

Now we provide a bound for the second term in (39) of $|z^\top L_{k,\tau}^{-1} E_{k,\tau}(L_{k,\tau} + E_{k,\tau})^{-1}(g_{k,\tau}(\widehat{\theta}_{k,\tau}) - g_{k,\tau}(\theta_k^*))|$. We have

$$|z^\top L_{k,\tau}^{-1} E_{k,\tau}(L_{k,\tau} + E_{k,\tau})^{-1}(g_{k,\tau}(\widehat{\theta}_{k,\tau}) - g_{k,\tau}(\theta_k^*))|$$

$$\leq \|z\|_{L_{k,\tau}^{-1}} \|L_{k,\tau}^{-1/2} E_{k,\tau}(L_{k,\tau} + E_{k,\tau})^{-1} L^{1/2}\|_2 \|g_{k,\tau}(\widehat{\theta}_{k,\tau}) - g_{k,\tau}(\theta_k^*)\|_{L_{k,\tau}^{-1}}$$

$$\leq (1/\kappa)\|z\|_{V_{k,\tau}^{-1}} \|L_{k,\tau}^{-1/2} E_{k,\tau}(L_{k,\tau} + E_{k,\tau})^{-1} L^{1/2}\|_2 \|g_{k,\tau}(\widehat{\theta}_{k,\tau}) - g_{k,\tau}(\theta_k^*)\|_{V_{k,\tau}^{-1}}. \tag{43}$$

Then it follows that

$$\|L_{k,\tau}^{-1/2} E_{k,\tau}(L_{k,\tau} + E_{k,\tau})^{-1} L^{1/2}\|_2$$

$$= \|L_{k,\tau}^{-1/2} E_{k,\tau}(L_{k,\tau}^{-1} - L_{k,\tau}^{-1} E_{k,\tau}(L_{k,\tau} + E_{k,\tau})^{-1}) L^{1/2}\|_2$$

$$\leq \|L_{k,\tau}^{-1/2} E_{k,\tau} L_{k,\tau}^{-1/2}\|_2 + \|L_{k,\tau}^{-1/2} E_{k,\tau} L_{k,\tau}^{-1/2}\|_2 \|L_{k,\tau}^{-1/2} E_{k,\tau}(L_{k,\tau} + E_{k,\tau})^{-1} L_{k,\tau}^{1/2}\|_2,$$

which implies

$$\|L_{k,\tau}^{-1/2} E_{k,\tau}(L_{k,\tau} + E_{k,\tau})^{-1} L_{k,\tau}^{1/2}\|_2 \leq \frac{\|L_{k,\tau}^{-1/2} E_{k,\tau} L_{k,\tau}^{-1/2}\|_2}{1 - \|L_{k,\tau}^{-1/2} E_{k,\tau} L_{k,\tau}^{-1/2}\|_2}$$

$$\leq 2\|L_{k,\tau}^{-1/2} E_{k,\tau} L_{k,\tau}^{-1/2}\|_2$$

$$\leq \frac{6}{\kappa}\|\widehat{\theta}_{k,\tau} - \theta_k^*\|_2, \tag{44}$$

where the last inequality is obtained from (17) and (18) in Oh & Iyengar (2021). Then from (43), (44), we have

$$|z^\top L_{k,\tau}^{-1} E_{k,\tau}(L_{k,\tau} + E_{k,\tau})^{-1}(g_{k,\tau}(\widehat{\theta}_{k,\tau}) - g_{k,\tau}(\theta_k^*))|$$

$$\leq \frac{6}{\kappa^2}\|\widehat{\theta}_{k,\tau} - \theta_k^*\|_2 \|g_{k,\tau}(\widehat{\theta}_{k,\tau}) - g_{k,\tau}(\theta_k^*)\|_{V_{k,\tau}^{-1}} \|z\|_{V_{k,\tau}^{-1}}. \tag{45}$$

We can conclude the proof from (42) and (45).

### A.9.2 PROOF OF LEMMA A.7

We note that $g_{k,\tau}(\theta_1) - g_{k,\tau}(\theta_2) = \sum_{t\in\mathcal{T}_{k,\tau}} \sum_{n\in S_{k,t}} (p(n,|S_{k,t},\theta_1) - p(n,|S_{k,t},\theta_2))z_n + (\theta_1 - \theta_2)$.

Define $H_{k,\tau}(\theta) = \sum_{t\in\mathcal{T}_{k,\tau}} \left( \sum_{n\in S_{k,t}} p(n|S_{k,t},\theta)z_n z_n^\top - \sum_{n\in S_{k,t}} \sum_{m\in S_{k,t}} p(n|S_{k,t},\theta)p(m|S_{k,t},\theta)z_n z_m^\top \right) + I_r$. Then we can show that there exists $\bar{\theta} = c\theta_1 + (1-c)\theta_2$ with some $c\in(0,1)$ such that

$$g_{k,\tau}(\theta_1) - g_{k,\tau}(\theta_2)$$

$$= \nabla_\theta g_{k,\tau}(\theta)\big|_{\theta=\bar{\theta}}(\theta_1 - \theta_2)$$

$$= \left( \sum_{t\in\mathcal{T}_{k,\tau}} \left( \sum_{n\in S_{k,t}} p(n|S_{k,t},\bar{\theta})z_n z_n^\top - \sum_{n\in S_{k,t}} \sum_{m\in S_{k,t}} p(n|S_{k,t},\bar{\theta})p(m|S_{k,t},\bar{\theta})z_n z_m^\top \right) + I_r \right)(\theta_1 - \theta_2)$$

$$= H_{k,\tau}(\bar{\theta})(\theta_1 - \theta_2). \tag{46}$$

Define $\bar{H}_{k,\tau}(\bar{\theta}) = \sum_{t\in\mathcal{T}_{k,\tau}} \sum_{n\in S_{k,t}} p(n|S_{k,t},\bar{\theta})p(n_0|S_{k,t},\bar{\theta})z_n z_n^\top + I_r$. Then we have $H_{k,\tau}(\bar{\theta}) \succeq \bar{H}_{k,\tau}(\bar{\theta})$ from the following.

$$\sum_{t\in\mathcal{T}_{k,\tau}} \left( \sum_{n\in S_{k,t}} p(n|S_{k,t},\bar{\theta})z_n z_n^\top - \sum_{n\in S_{k,t}} \sum_{m\in S_{k,t}} p(n|S_{k,t},\bar{\theta})p(m|S_{k,t},\bar{\theta})z_n z_m^\top \right)$$

$$= \sum_{t\in\mathcal{T}_{k,\tau}} \left( \sum_{n\in S_{k,t}} p(n|S_{k,t},\bar{\theta})z_n z_n^\top - \sum_{n\in S_{k,t}} \sum_{m\in S_{k,t}} p(n|S_{k,t},\bar{\theta})p(m|S_{k,t},\bar{\theta})z_n z_m^\top \right)$$

$$= \sum_{t\in\mathcal{T}_{k,\tau}} \left( \sum_{n\in S_{k,t}} p(n|S_{k,t},\bar{\theta})z_n z_n^\top - \frac{1}{2} \sum_{n\in S_{k,t}} \sum_{m\in S_{k,t}} p(n|S_{k,t},\bar{\theta})p(m|S_{k,t},\bar{\theta})(z_n z_m^\top + z_m z_n^\top) \right)$$

$$\succeq \sum_{t\in\mathcal{T}_{k,\tau}} \left( \sum_{n\in S_{k,t}} p(n|S_{k,t},\bar{\theta})z_n z_n^\top - \frac{1}{2} \sum_{n\in S_{k,t}} \sum_{m\in S_{k,t}} p(n|S_{k,t},\bar{\theta})p(m|S_{k,t},\bar{\theta})(z_n z_n^\top + z_m z_m^\top) \right)$$

$$= \sum_{t\in\mathcal{T}_{k,\tau}} \left( \sum_{n\in S_{k,t}} p(n|S_{k,t},\bar{\theta})z_n z_n^\top - \sum_{n\in S_{k,t}} \sum_{m\in S_{k,t}} p(n|S_{k,t},\bar{\theta})p(m|S_{k,t},\bar{\theta})z_n z_n^\top \right)$$

$$= \sum_{t\in\mathcal{T}_{k,\tau}} \left( \sum_{n\in S_{k,t}} p(n|S_{k,t},\bar{\theta})p(n_0|S_{k,t},\bar{\theta})z_n z_n^\top \right), \tag{47}$$

where the inequality is obtained from $(z_n - z_m)(z_n - z_m)^\top \succeq 0$. Under $E_1$, we have $\|\hat{\theta}_{k,\tau}\|_2 - \|\theta_k^*\|_2 \leq 1$ implying $\|\hat{\theta}_{k,\tau}\|_2 \leq 1 + \|\theta_k^*\|_2 = 1 + \|U_r^\top \theta_k\|_2 \leq 2$. Then for $\bar{\theta} = c\hat{\theta}_{k,\tau} + (1-c)\theta_k^*$ for some $c\in(0,1)$, we have $\|U_r\bar{\theta}\|_2 \leq 2$. Then from $p(n|S_{k,t},\bar{\theta}) = \exp(z_n^\top\bar{\theta})/(1 + \sum_{m\in S_{k,t}} \exp(z_m^\top\bar{\theta})) = \exp(x_n^\top(U_r\bar{\theta}))/(1 + \sum_{m\in S_{k,t}} \exp(x_m^\top(U_r\bar{\theta})))$, we can show that $\bar{H}_{k,\tau}(\bar{\theta}) \succeq \kappa V_{k,\tau}$, which implies $H_{k,\tau}(\bar{\theta}) \succeq \bar{H}_{k,\tau}(\bar{\theta}) \succeq \kappa V_{k,\tau}$.

Then we have

$$\|\hat{\theta}_{k,\tau} - \theta_k^*\|_2^2 \leq (1/\lambda_{\min}(V_{k,\tau}))(\hat{\theta}_{k,\tau} - \theta_k^*)^\top V_{k,\tau}(\hat{\theta}_{k,\tau} - \theta_k^*)$$

$$\leq (1/\kappa\lambda_{\min}(V_{k,\tau}^0))(\hat{\theta}_{k,\tau} - \theta_k^*)^\top H_{k,\tau}(\bar{\theta})(\hat{\theta}_{k,\tau} - \theta_k^*)$$

$$\leq (1/\kappa\lambda_{\min}(V_{k,\tau}^0))(\hat{\theta}_{k,\tau} - \theta_k^*)^\top H_{k,\tau}(\bar{\theta})H_{k,\tau}(\bar{\theta})^{-1}H_{k,\tau}(\bar{\theta})(\hat{\theta}_{k,\tau} - \theta_k^*)$$

$$\leq (1/\kappa^2\lambda_{\min}(V_{k,\tau}^0))(g_{k,\tau}(\hat{\theta}_{k,\tau}) - g_{k,\tau}(\theta_k^*))^\top V_{k,\tau}^{-1}(g_{k,\tau}(\hat{\theta}_{k,\tau}) - g_{k,\tau}(\theta_k^*))$$

$$\leq (1/\kappa^2\lambda_{\min}(V_{k,\tau}^0))\|g_{k,\tau}(\hat{\theta}_{k,\tau}) - g_{k,\tau}(\theta_k^*)\|_{V_{k,\tau}^{-1}}^2. \tag{48}$$

Then from $E_2$, we can conclude that

$$\|\hat{\theta}_{k,\tau} - \theta_k^*\|_2 \leq \frac{4}{\kappa}\sqrt{\frac{2r + \log(KTN/\delta)}{\lambda_{\min}(V_{k,\tau}^0)}}.$$

## A.10 PROOF OF PROPOSITION A.1

We first provide a lemma for a confidence bound. Let $\gamma_t(\delta) = c_1\sqrt{d}\log(L)\left(\log(t) + \sqrt{\log(t)\log(K/\delta)}\right)$ for some $c_1 > 0$.

**Lemma A.18** (Lemma 1 in Lee & Oh (2024)). *With probability at least $1 - \delta$, for all $t \geq 1$ and $k \in [K]$ we have*

$$\|\widehat{\theta}_{k,t} - \theta_k^*\|_{\mathcal{G}_{k,t}} \leq \gamma_t(\delta).$$

Let $\delta = 1/T$. From the above lemma, we define event $E = \{\|\widehat{\theta}_{k,t} - \theta_k^*\|_{\mathcal{G}_{k,t}} \leq \gamma_t \, \forall k \in [K] \text{ and } t \geq 1\}$, which holds with probability at least $1 - 1/T$. Then we provide a lemma for the optimism.

**Lemma A.19.** *Under $E$, for all $t \geq 1$, we have*

$$\sum_{k\in[K]} R_k(S_k^*) \leq \sum_{k\in[K]} R_{k,t}^{UCB}(S_{k,t}).$$

*Proof.* Under $E$, we have

$$|z_n^\top \widehat{\theta}_{k,t} - z_n^\top \theta_k^*| \leq \|z_n\|_{\mathcal{G}_{k,t}^{-1}}\|\widehat{\theta}_{k,t} - \theta_k^*\|_{\mathcal{G}_{k,t}} \leq \gamma_t\|z_n\|_{\mathcal{G}_{k,t}^{-1}},$$

which implies $z_n^\top \theta_k^* \leq z_n^\top \widehat{\theta}_{k,t} + \gamma_t\|z_n\|_{\mathcal{G}_{k,t}^{-1}} = h_{n,k,t}$. Therefore, from Lemma A.3 in Agrawal et al. (2017a), we have $R_k(S_k^*) \leq R_{k,t}^{UCB}(S_k^*)$. Then using definition of $S_{k,t}$ in the algorithm, we can conclude that

$$\sum_{k\in[K]} R_k(S_k^*) \leq \sum_{k\in[K]} R_{k,t}^{UCB}(S_k^*) \leq \sum_{k\in[K]} R_{k,t}^{UCB}(S_{k,t}).$$

$\square$

Now we provide a lemma which is critical to bound regret under optimism.

**Lemma A.20.** *Under $E$, for all $k \in [K]$, we have*

$$\sum_{t=1}^{T} R_{k,t}^{UCB}(S_{k,t}) - R_k(S_{k,t}) = O\left(r\sqrt{T} + \frac{1}{\kappa}r^2\right)$$

*Proof.* By following the proof steps in Theorem 4 in Lee & Oh (2024), we can show this lemma. $\square$

Then from Lemmas A.18 and A.20, we can conclude the proof for the regret as follows.

$$\mathcal{R}(T) = \mathbb{E}\left[\sum_{t\in[T]}\sum_{k\in[K]} R_k(S_{k,t}^*) - R_k(S_{k,t})\right]$$

$$\leq \mathbb{E}\left[\sum_{t=1}^{T}\sum_{k\in[K]}\left(R_k(S_{k,t}^*) - R_k(S_{k,t})\right)\mathbf{1}(E)\right] + \mathbb{E}\left[\sum_{t=1}^{T}\sum_{k\in[K]}\left(R_k(S_{k,t}^*) - R_k(S_{k,t})\right)\mathbf{1}(E^c)\right]$$

$$\leq \mathbb{E}\left[\sum_{t=1}^{T}\sum_{k\in[K]}\left(R_{k,t}^{UCB}(S_{k,t}) - R_k(S_{k,t})\right)\mathbf{1}(E)\right] + \sum_{t=1}^{T}\sum_{k\in[K]}\mathbb{P}(E^c)$$

$$= \tilde{\mathcal{O}}\left(rK\sqrt{T} + \frac{1}{\kappa}r^2 K\right) = \tilde{\mathcal{O}}\left(rK\sqrt{T}\right).$$

Now we discuss the computational cost. Since there exists $O(K^N)$ number of assortment candidate in $\mathcal{M}$, especially for $L \geq N$, the cost per round is $O(K^N)$ from Line 3.

### A.11 PROOF OF LEMMA A.10

Let $W(\pi) = V(\pi) + (1/rT_\tau)I_r$ and $g(\pi) = \max_{n \in \mathcal{N}_{k,\tau}} \|z_n\|^2_{(V(\pi)+(1/rT_\tau)I_r)^{-1}}$. Since $\pi_{k,\tau}$ is G-optimal, for $n \in supp(\pi_{k,\tau})$ we have that $z_n^\top W(\pi_{k,\tau})^{-1} z_n = g(\pi_{k,\tau})$ (otherwise, there exists $\pi'$ such that $g(\pi') \leq g(\pi_{k,\tau})$, which is a contradiction). Then we have $\sum_{n \in \mathcal{N}_{k,\tau}} \pi_{k,\tau}(n) z_n^\top W(\pi_{k,\tau})^{-1} z_n = g(\pi_{k,\tau})$. Therefore, we obtain

$$g(\pi) = \sum_{n \in \mathcal{N}_{k,\tau}} \pi_{k,\tau}(n) z_n^\top W(\pi_{k,\tau})^{-1} z_n = trace(\sum_{n \in \mathcal{N}_{k,\tau}} \pi_{k,\tau}(n) z_n z_n^\top W(\pi_{k,\tau})^{-1})$$

$$= trace((W(\pi_{k,\tau}) - (1/rT_\tau)I_d)W(\pi_{k,\tau})^{-1}) = d - (1/rT_\tau)trace(W(\pi_{k,\tau})^{-1}) \leq d.$$

Let $S = supp(\pi_{k,\tau})$. Then if $|S| > d(d+1)/2$ there are linearly dependent: $\exists v : S \to \mathbb{R}$ such that $\sum_{n \in S} v(n) z_n z_n^\top = 0$. Therefore, for $n \in S$, $z_n^\top W(\pi_{k,\tau})^{-1} z_n \sum_{n \in S} v(n) = trace(W(\pi_{k,\tau})^{-1} \sum_{n \in S} v(n) z_n z_n^\top) = 0$, which implies $\sum_{n \in S} v(n) = 0$. Define $\pi(t) = \pi_{k,\tau} + tv$, then we have $W(\pi(t)) = W(\pi_{k,\tau})$ for every $t$, which implies $g(\pi_{k,\tau}) = g(\pi(t))$. Let $t' = \sup\{t > 0 : \pi_{k,\tau}(n) + tv(n) \geq 0 \, \forall n \in S\}$. At $t = t'$, at least one weight becomes 0 (otherwise, there exists $t'' \geq t'$ s.t. $\pi_{k,\tau}(n) + t''v(n) \geq 0$ for all $n \in S$, which is a contradiction). Thus, we have an equally good design with $|S| - 1$ arms. Iterating the construction yields an optimal design $\pi$ with $|supp(\pi)| \leq d(d+1)/2$.

### A.12 AUXILIARY LEMMAS

**Lemma A.21** (Lemma E.2 in Lee & Oh (2024)). *For all $t \geq 1$ and $k \in [K]$, we have*

$$(i) \quad \sum_{s=1}^{t} \sum_{n \in S_{k,s}} p(n|S_{k,s}, \widehat{\theta}_{k,s}) p(n_0|S_{k,s}, \widehat{\theta}_{k,s}) \|z_n\|^2_{H_{k,s}^{-1}} \leq 2r \log\left(1 + \tfrac{t}{r\lambda}\right),$$

$$(ii) \quad \sum_{s=1}^{t} \max_{n \in S_{k,s}} \|z_n\|^2_{H_{k,s}^{-1}} \leq \tfrac{1}{\kappa} 2r \log\left(1 + \tfrac{t}{r\lambda}\right).$$

**Lemma A.22** (Lemma E.3 in Lee & Oh (2024)). *Define $\tilde{Q} : \mathbb{R}^{|S|} \to \mathbb{R}$ for $S \in [N]$, such that for any $\boldsymbol{u} = (u_1, \ldots, u_{|S|}) \in \mathbb{R}^{|S|}$, $\tilde{Q}(\boldsymbol{u}) = \sum_{n \in S} \frac{\exp(u_n)}{1 + \sum_{m \in S} \exp(u_m)}$. Let $p_n(\boldsymbol{u}) = \frac{\exp(u_n)}{1 + \sum_{m \in S} \exp(u_m)}$. Then for all $n \in S$, we have*

$$\left| \frac{\partial^2 \tilde{Q}}{\partial u_n \partial u_m} \right| \leq \begin{cases} 3p_n(\boldsymbol{u}), & \text{if } n = m \\ 2p_n(\boldsymbol{u})p_m(\boldsymbol{u}), & \text{if } n \neq m \end{cases}$$

### A.13 ADDITIONAL EXPERIMENTS

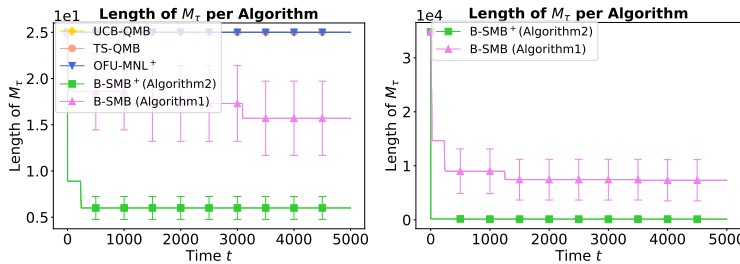

Figure 4: Cardinality of the active assignment set $\mathcal{M}_\tau$ over epochs for (left) $N = 3, K = 2$ and (right) $N = 7, K = 4$.

As shown in Figure 4, the size of the active assignment set $\mathcal{M}_\tau$ decreases rapidly across epochs. This demonstrates that elimination removes the vast majority of assignment candidates early on, greatly reducing the effective search space for the rare assortment-optimization steps. Consequently, the practical optimization cost is far smaller than the theoretical worst-case $O(K^N)$ bound.

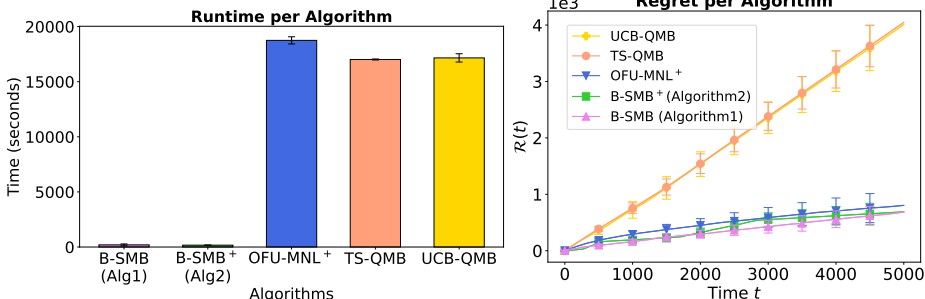

Figure 5: Experimental results with $N = 8$ and $K = 4$ for (left) runtime cost and (right) regret of algorithms. Notably, increasing $N$ from 7 to 8 (as opposed to Figure 2) causes the runtime of OFU-MNL$^+$ to exceed 15,000 seconds—up from 5,000 seconds—whereas our algorithms maintain runtimes under 1,000 seconds. In terms of regret performance, our algorithms achieve results comparable to OFU-MNL$^+$ while outperforming other benchmarks.

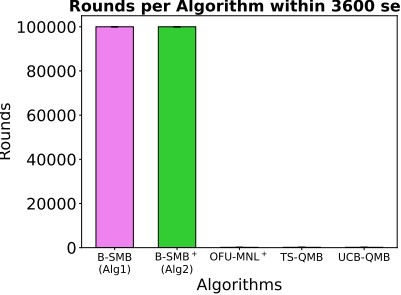

Figure 6: Computational overhead of benchmark algorithms prevents scaling to larger problem sizes, limiting experimental comparison. For example, with $N = 8$, $K = 5$, and $T = 100{,}000$, the figure reports the number of rounds completed by each algorithm within a 3600-second limit. Increasing $K$ from 4 to 5, similar to increasing $N$, significantly increases the runtime overhead of the benchmarks, allowing only a few completed rounds (barely visible in the plot). In contrast, our algorithms (B-SMB, B-SMB$^+$) successfully complete all 100,000 rounds within the time limit.

