# OpenReview forum: "Batched Stochastic Matching Bandits"
_ICLR.cc/2026/Conference — Submitted to ICLR 2026_

### Official Review · Reviewer_5UXB · 2025-10-30

**Soundness:** 3
**Presentation:** 2
**Contribution:** 3
**Rating:** 4
**Confidence:** 4

**Summary:**

This paper introduces a novel framework called Stochastic Matching Bandits (SMB), which generalizes the standard multi-armed bandit problem to a bipartite matching setting where arms stochastically select agents from assigned pools according to a Multinomial Logit (MNL) choice model with unknown preferences. The goal is to maximize cumulative reward from successful matches while learning these preferences online. A key challenge is the NP-hard computational complexity of the resulting combinatorial assignment problem in each round, which makes a naive approach prohibitively expensive (O(K^N) per round). To address this, the authors propose batched algorithms that limit the frequency of matching updates. Their main contributions are: (1) a batched algorithm (B-SMB) that achieves $\tilde{O}(\sqrt{T})$ regret with $O(1)$ amortized computational cost when a problem-dependent parameter $\kappa$ is known, (2) an improved algorithm that removes the need for knowing $\kappa$, and (3) empirical validation showing significant computational savings with comparable regret to non-batched baselines.

**Strengths:**

1. The SMB framework effectively models real-world matching markets (e.g., ride-hailing, job platforms) where choices are stochastic and preferences are latent, addressing a significant gap between traditional online matching theory and practical applications.
​​2. The batched approach is a clever and impactful contribution. By reducing the amortized cost to O(1) through limited updates $\Theta (\log\log T)$, the paper makes the problem tractable for large-scale systems without sacrificing theoretical regret guarantees.
​​3. The paper provides theoretical analysis, including regret bounds that match or improve upon existing lower bounds for related bandit problems. The analysis handles the complexity of multiple, interacting MNL models.

**Weaknesses:**

1. Strong Assumptions:​​ The requirement that the candidate set size |S_k,t| is bounded by a known constant L and that rewards r_{n,k} are known in advance may not hold in all real-world scenarios. These assumptions simplify the problem but limit generality.
2. ​​Empirical Scope:​​ While experiments demonstrate computational gains, the scale of the problems tested (N up to 8, K up to 5) is relatively small. It's unclear how the algorithms perform in massive markets (e.g., thousands of agents and arms), where the computational savings would be most critical.
3. ​​Practical Implementation Hurdles:​​ The algorithms involve solving G-optimal design problems and maintaining complex confidence structures. The practical implementation overhead (beyond just combinatorial optimization cost) is not thoroughly discussed.

**Questions:**

The proposed algorithms rely on a batched elimination approach that requires solving a combinatorial optimization over the active set M_τ at each update. While the number of updates is small, the cost per update could still be very high if the active set remains large. How does the size of the active set M_τ evolve during the elimination process in practice, and are there safeguards or heuristics to control its size and ensure that the individual batch updates remain computationally feasible, especially for very large N and K?

---

> ### Author Response · Authors · 2025-11-20
>
> We thank the reviewers for their thoughtful comments and constructive feedback. We have addressed all points below.
>
> - **Comment: The requirement that the candidate set size $|S_{k,t}|$ is bounded by a known constant L and that rewards $r_{n,k}$ are known in advance may not hold in all real-world scenarios. These assumptions simplify the problem but limit generality.**
>
>
>  We emphasize that both assumptions follow the standard MNL bandit framework and do not impose additional restrictions beyond those commonly used in prior work.
>
> **(1) Bounded candidate set size $|S_{k,t}| \le L$.**
> Our model is directly based on the MNL bandit framework. As in prior studies
> (Agrawal et al., 2017b; Oh \& Iyengar, 2019, 2021; Lee \& Oh, 2024),
> we introduce $L$ as the maximum assortment size, with the unconstrained case recovered by setting set $L = N$.
> In many practical platforms—such as ride-hailing, delivery services, or online marketplaces—each agent is presented with only a *small* subset of options due to user-interface constraints, cognitive load limits, or geo-spatial filtering.
> Thus, $|S_{k,t}| \le L \le N$ often reflects **operational reality** rather than a restrictive modeling assumption.
>
> **(2) Known rewards $r_{n,k}$.**
> Assuming known $r_{n,k}$ is also standard in the MNL bandit literature, where the primary challenge is learning latent user (or arm) preferences rather than estimating deterministic reward parameters.
> In real systems, platforms typically know prices or commissions beforehand—e.g., driver fares, delivery fees, or task payments—before making assignment decisions.
>
> The expected reward of a matching decision is
> $$
> \sum_{k \in [K]} R_k(S_k)
> = \sum_{k \in [K]} \sum_{n \in S_k} r_{n,k} \cdot p(n \mid S_k, \theta_k),
> $$
> where $p(n \mid S_k, \theta_k)$ captures the MNL acceptance probability, and $r_{n,k}$ represents the known payoff upon successful matching.
>
>
> - **Comment: Empirical Scope: While experiments demonstrate computational gains, the scale of the problems tested (N up to 8, K up to 5) is relatively small. It's unclear how the algorithms perform in massive markets (e.g., thousands of agents and arms), where the computational savings would be most critical.**
>
> **Response:**
> As discussed in Appendix A.13, existing benchmark algorithms (e.g., OFU-MNL$^+$ [24], UCB-QMB [20], TS-QMB [20]) incur substantial computational overhead, which prevents evaluation on larger problem sizes. For example, in Figure 6 (Appendix A.13) for $N = 8$, $K = 5$, and $T = 100{,}000$, baseline methods complete only a few rounds within a 3600-second limit due to their per-round optimization cost—often barely visible in the plot. In contrast, our algorithms (B-SMB and B-SMB$^+$) complete all $100{,}000$ rounds within the same time budget, demonstrating their computational advantages.
>
> We also note that the scalability of our method comes from its batched design: optimization is performed only $O(\log\log T)$ times, and the active set shrinks rapidly across epochs, making the effective search space far smaller than the theoretical worst case. These structural properties allow the algorithm to remain computationally feasible even as $N$ and $K$ grow.
>
>
>
>
>
>
>
>
>
>
>
>
>
>
>
>
> - **Comment: The algorithms involve solving G-optimal design problems and maintaining complex confidence structures. The practical implementation overhead (beyond just combinatorial optimization cost) is not thoroughly discussed.**
>
>
> **Response:** As discussed in Lines 305–313, the G-optimal design subproblem can be efficiently solved using the Frank–Wolfe algorithm (Ahipaşaoğlu et al., 2008), which provides a practical and scalable method for computing the required design vectors. The confidence structure in Algorithms 1.2 is also derived directly from the curvature of the MNL model and only requires maintaining the Gram matrix and its updates, which can be handled using standard incremental matrix-update techniques. The algorithms iteratively computes the inverse gram matrix using the Sherman-Morrison formula [Hager, William W., 1989], exploiting the fact
> that the matrix is a weighted sum of an identity matrix and rank-1 matrices.
>
> We will expand the discussion in the final version to make these implementation aspects clearer and to emphasize that the overhead beyond combinatorial optimization is modest and relies on well-established numerical routines.

---

> ### Author Response · Authors · 2025-11-20
>
> - **Comment:
> The proposed algorithms rely on a batched elimination approach that requires solving a combinatorial optimization over the active set $\mathcal{M}\_\tau$ at each update. While the number of updates is small, the cost per update could still be very high if the active set remains large. How does the size of the active set $\mathcal{M}\_\tau$ evolve during the elimination process in practice, and are there safeguards or heuristics to control its size and ensure that the individual batch updates remain computationally feasible, especially for very large N and K?**
>
> **Response:** **Our algorithm does not require enumerating the full matching space $O(K^N)$.**
> As noted in revised Remark 5.4, this quantity reflects the theoretical size of the worst-case action space—not the computational
> cost incurred by our method. In Algorithm 1, we do **not** optimize over all matchings.  The active set shrinks rapidly due to elimination. Eliminating suboptimal assignment pairs
> $(n,k)$ greatly reduces the candidate set after only a few batches. Empirically, most assignments are removed
> early, so the theoretical worst-case exponential dependence in $N$ is not encountered in practice. We've added the plot for the size of the active assignment set $\mathcal{M}_\tau$ (Line 14 in Algorithm 1, Line 41 in Algorithm 2) at Figure 4 in Appendix A.13 in our revision. This
> demonstrates that elimination removes the vast majority of assignment candidates early on, greatly
> reducing the effective search space for the rare assortment-optimization steps. Consequently, the
> practical optimization cost of $O(|\mathcal{M}\_\tau|)$ is far smaller than the theoretical worst-case $O(K^N)$ bound.
>
>
> As discussed in Appendix A.13, the scalability of benchmark algorithms (e.g., OFU-MNL$^+$[Lee \& Oh, 2024], UCB-QMB[Kim \& Oh, 2024], TS-QMB[Kim \& Oh, 2024]) is limited by their computational overhead. For example, in Figure 6 (Appendix A.13) with $N = 8$, $K = 5$, and $T = 100{,}000$, we report the number of rounds completed by each algorithm within a 3600-second time limit. Increasing $K$ from 4 to 5 (or $N$ likewise) leads to a substantial increase in runtime for the baselines—resulting in only a few completed rounds (barely visible in the plot). In contrast, our algorithms (B-SMB, B-SMB$^+$) successfully complete all 100,000 rounds within the time budget, demonstrating their computational efficiency and practical scalability.
>
>
> We highlight that **our algorithms perform the optimization only $O(\log\log T)$ times—not every round.**   As is well known in the combinatorial bandit literature  [Chen et al. (2013)], solving exact combinatorial optimization problems—without approximation—inevitably incurs exponential worst-case complexity. This cost is unavoidable unless approximation oracles targeting approximated regret are introduced.
> A key contribution of our work is the batched elimination design: the preference estimates $\hat\theta_{k,\tau}$
> are updated only at the endpoints of exponentially growing epochs. Even if each optimization step is costly,
> its **frequency** is extremely low. As shown in Remark 5.4, the amortized per-round complexity becomes $O(1)$ when $T$ is large enough.
>
> We emphasize that lowering the total computational cost of optimization from $O(K^N T)$ to $O(K^N \log\log T)$ constitutes a major improvement, offering significant benefits in both theoretical analysis and practical implementation. Even for moderately sized $T$, this yields a **substantial reduction in computational cost**.
> As shown in Figure 5 (Appendix A.13), this efficiency is significant. our algorithms complete 5,000 rounds within **1,000 seconds**, while other benchmarks require over **15,000 seconds** when $N=8$ and $K=4$.
>
>
> **In summary**, the apparent $O(K^N)$ complexity reflects the theoretical size of the action space, not the
> actual computational workload. Our algorithm performs optimization extremely rarely, over rapidly shrinking
> candidate sets. This performance advantage is conclusively demonstrated by both theoretical analysis and experimental results.

---

### Official Review · Reviewer_K7W9 · 2025-10-30

**Soundness:** 3
**Presentation:** 2
**Contribution:** 3
**Rating:** 6
**Confidence:** 3

**Summary:**

This paper introduces a novel "Stochastic Matching Bandits (SMB)" framework, designed to model online matching markets like ride-hailing. In this setting, $N$ agents (e.g., riders) are assigned to $K$ arms (e.g., drivers). The core novelty is that arms select stochastically from their assigned pool of agents according to an unknown Multi-nomial Logit (MNL) preference model. The paper identifies the primary challenge of this framework: the underlying combinatorial optimization problem is NP-hard, leading to a prohibitive $O(K^N)$ computational cost for a naive, per-round approach. The main contribution is a set of "batched" algorithms (B-SMB and B-SMB+) that strategically limit the frequency of these expensive matching updates to only $\Theta(\log\log T)$ times over a horizon of $T$ rounds.

**Strengths:**

An improved model with solid technical results. The core technical contribution is the elegant solution to the $O(K^N)$ computational bottleneck. The proposal to use a batched algorithm—which runs the NP-hard optimization only $\Theta(\log\log T)$ times—is a highly practical and innovative idea. Proving that this massive reduction in computational effort can still achieve an optimal $\tilde{O}(\sqrt{T})$ regret bound is a significant theoretical achievement.

**Weaknesses:**

1. Fundamental Mismatch in Problem Scale ($N$ and $K$). This is the most significant concern with the paper. The primary motivation is large-scale ride-sharing platforms (e.g., Uber, Lyft), where the number of agents ($N$, riders) and arms ($K$, drivers) are both massive, potentially in the thousands. However, the paper's entire technical framework is built around solving a core optimization problem that has a computational cost of $O(K^N)$ (as noted in Remarks 5.4). This complexity is exponential in $N$.This leads to a fundamental contradiction:The algorithm is only computationally feasible if $N$ is a very small constant (e.g., $N < 10$). The paper's own experiments, which use $N=7$ confirm this limitation. The proposed B-SMB algorithm could not even run its first step, making the $O(1)$ amortized cost irrelevant.This implies the paper's problem formulation (the $O(K^N)$ optimization) is fundamentally mismatched with its motivating application (the $N=$millions scenario). To make the algorithm's usability and relevance to large markets far stronger, could the authors provide an experiment with a larger $K$ and N  (e.g., $K, N ~ 10,000$)?

2. Theoretical Gap in Regret Bound: The paper itself (Discussion on Tightness) acknowledges a theoretical gap. The final regret bound of $\tilde{O}(K^{3/2}\sqrt{T})$ (for Algorithm 2) is worse than the derived lower bound of $\Omega(K\sqrt{T})$ by a $\sqrt{K}$ factor. While the authors provide a justification (the need to explore all potential matches during elimination), this remains a minor theoretical weakness.

**Questions:**

see weakness.

---

> ### Author Response · Authors · 2025-11-20
>
> We thank the reviewers for their thoughtful comments and constructive feedback. We have addressed all points below.
>
> - **Comment: The motivating scenario involves large-scale ride-sharing markets with thousands of riders and drivers, yet the core optimization step has worst-case complexity $O(K^N)$, which appears feasible only for very small $N$. This raises a potential mismatch between the application and the algorithm’s scalability. Can the authors demonstrate performance at larger $N, K$?**
>
> **Response:** **Our algorithm does not require enumerating the full matching space $O(K^N)$.**
> As noted in revised Remark 5.4, this quantity reflects the theoretical size of the worst-case action space—not the computational
> cost incurred by our method. In Algorithm 1, we do **not** optimize over all matchings.  The active set shrinks rapidly due to elimination. Eliminating suboptimal assignment pairs
> $(n,k)$ greatly reduces the candidate set after only a few batches. Empirically, most assignments are removed
> early, so the theoretical worst-case exponential dependence in $N$ is not encountered in practice. We've added the plot for the size of the active assignment set $\mathcal{M}_\tau$ (Line 14 in Algorithm 1, Line 41 in Algorithm 2) at Figure 4 in Appendix A.13 in our revision. This
> demonstrates that elimination removes the vast majority of assignment candidates early on, greatly
> reducing the effective search space for the rare assortment-optimization steps. Consequently, the
> practical optimization cost of $O(|\mathcal{M}\_\tau|)$ is far smaller than the theoretical worst-case $O(K^N)$ bound.
>
>
> As discussed in Appendix A.13, the scalability of benchmark algorithms (e.g., OFU-MNL$^+$[Lee \& Oh, 2024], UCB-QMB[Kim \& Oh, 2024], TS-QMB[Kim \& Oh, 2024]) is limited by their computational overhead. For example, in Figure 6 (Appendix A.13) with $N = 8$, $K = 5$, and $T = 100{,}000$, we report the number of rounds completed by each algorithm within a 3600-second time limit. Increasing $K$ from 4 to 5 (or $N$ likewise) leads to a substantial increase in runtime for the baselines—resulting in only a few completed rounds (barely visible in the plot). In contrast, our algorithms (B-SMB, B-SMB$^+$) successfully complete all 100,000 rounds within the time budget, demonstrating their computational efficiency and practical scalability.
>
>
> We highlight that **our algorithms perform the optimization only $O(\log\log T)$ times—not every round.**   As is well known in the combinatorial bandit literature  [Chen et al. (2013)], solving exact combinatorial optimization problems—without approximation—inevitably incurs exponential worst-case complexity. This cost is unavoidable unless approximation oracles targeting approximated regret are introduced.
> A key contribution of our work is the batched elimination design: the preference estimates $\hat\theta_{k,\tau}$
> are updated only at the endpoints of exponentially growing epochs. Even if each optimization step is costly,
> its **frequency** is extremely low. As shown in Remark 5.4, the amortized per-round complexity becomes $O(1)$ when $T$ is large enough.
>
> We emphasize that lowering the total computational cost of optimization from $O(K^N T)$ to $O(K^N \log\log T)$ constitutes a major improvement, offering significant benefits in both theoretical analysis and practical implementation. Even for moderately sized $T$, this yields a **substantial reduction in computational cost**. As shown in Figure 5 (Appendix A.13), this efficiency is significant. our algorithms complete 5,000 rounds within **1,000 seconds**, while other benchmarks require over **15,000 seconds** when $N=8$ and $K=4$.
>
> **In summary**, the apparent $O(K^N)$ complexity reflects the theoretical size of the action space, not the
> actual computational workload. Our algorithm performs optimization extremely rarely, over rapidly shrinking
> candidate sets. This performance advantage is conclusively demonstrated by both theoretical analysis and experimental results.
>
> - **Comment:
> The paper itself acknowledges a theoretical gap. While the authors provide a justification (the need to explore all potential matches during elimination), this remains a minor theoretical weakness.**
>
> **Response:** The regret term of $\widetilde{\mathcal{O}}(K^{3/2}\sqrt{T})$ arises from the need to ensure statistically sound elimination in a batched setting, under strict amortized efficiency constraints. Unlike fully adaptive methods that perform exact optimization at each round, our algorithm avoids solving the full combinatorial problem at every step. The extra $\sqrt{K}$ factor captures the cost of maintaining reliable learning without incurring prohibitive computation. While the lower bound $\Omega(K\sqrt{T})$ assumes round-by-round optimization, it remains open whether such a bound is achievable under our amortized and batch-limited setting.

---

### Official Review · Reviewer_682J · 2025-11-05

**Soundness:** 3
**Presentation:** 3
**Contribution:** 3
**Rating:** 4
**Confidence:** 3

**Summary:**

This paper investigates the stochastic matching bandit problem with batched updates. The market consists of N agents and K arms, where each arm’s preference is modeled by a multinomial logit (MNL) bandit. Specifically, when multiple agents select the same arm, the acceptance probability of each agent follows a softmax function. The objective is to determine an optimal assortment that maximizes the total expected reward across all arms. The paper proposes a batched learning algorithm and demonstrates that it requires only O(\log \log T) computational complexity while achieving a regret of order O(\sqrt{T}).

**Strengths:**

1. It is reasonable to incorporate randomness into the decision-making process of the arms.
2. The paper proposes a batched algorithm that effectively reduces the computational cost for an otherwise (per-round) computationally intractable problem.
3. The theoretical analysis is comprehensive, providing both upper bounds depending on $\kappa$ or not.

**Weaknesses:**

1. It would be helpful to include a motivating example in the problem-setting section. In particular, please provide more intuition about the roles of $x$, $\theta$, and $r$ to clarify how these quantities shape the model and its implications.
2. For batched learning, it is well known that $O(\log \log T)$ switching often suffices to achieve $O(\sqrt{T})$ regret in standard MAB settings. Your result aligns with this pattern for the multi-agent MNL bandit. I encourage the authors to elaborate on the technical challenges unique to this setting—relative both to existing batched bandit frameworks and to the single-player MNL bandit—highlighting what new ideas are required.
3. Though using a batched algorithm can reduce the computational cost, it still needs to solve the combinatorial optimization problem in some rounds, which requires exponential time complexity and is not feasible in large-scale markets.
4. Minor: Lines 162–163 contain several notational typos.

**Questions:**

Please see the last part.

---

> ### Author Response · Authors · 2025-11-20
>
> We thank the reviewers for their thoughtful comments and constructive feedback. We have addressed all points below.
>
> - **Comment: It would be helpful to include a motivating example in the problem-setting section. In particular, please provide more intuition about the roles of $x,\theta, r$
>  to clarify how these quantities shape the model and its implications.**
>
> **Response:**
>  Although our paper initially referenced ride-hailing dispatch systems as a motivating example (Lines 44–48), we provide additional motivating example  based on an online freelancer marketplace, where the platform matches a set of tasks (agents) to freelancers (arms). In this example:
>
>
> - $x_n$ denotes the feature vector of task $n$, representing its required skills and known in advance.
>
> - $\theta_k$ is the latent preference vector of freelancer $k$, capturing how well their expertise aligns with different task types.
>
> - $r_{n,k}$ is the revenue obtained when freelancer $k$ successfully completes task $n$.
>
> Under the MNL choice model, the platform recommends tasks to freelancers based on the success probability implied by the utility $x_n^\top \theta_k$, together with the corresponding revenue $r_{n,k}$. This example illustrates how $x_n$, $\theta_k$, and $r_{n,k}$ jointly determine both the likelihood and value of successful matches, shaping the platform’s learning and optimization objectives.
>
>
> - **Comment: I encourage the authors to elaborate on the technical challenges unique to this setting—relative both to existing batched bandit frameworks and to the single-player MNL bandit—highlighting what new ideas are required.**
>
> **Response:**
>  The core algorithmic and theoretical developments in this work are novel and tailored to the unique challenges of the multi-assortment MNL setting with combinatorially large action spaces.
>
>
>
> In contrast to prior work in (Batched) linear bandits [Ruan et al., 2021; Hanna et al., 2023; Han et al., 2020; Ren \& Zhou,
> 2024; Sawarni et al., 2024; Ren et al., 2024], our setting requires managing an exponentially large action space, making direct application of standard techniques intractable. To overcome this, we design a statistically efficient exploration strategy based on elimination of suboptimal matches. Instead of exploring all possible matchings—which is computationally prohibitive—we assess the eligibility of each assignment $(n, k)$ (i.e., agent $n$ and arm $k$) by constructing a representative assortment $\\{S\_{l,\tau}^{(n,k)}\\}\_{l \in [K]}$ from an optimistic perspective (Line 5 in Algorithm 1). This allows us to perform elimination over a compressed search space, reducing both statistical overhead.
>
> Furthermore, unlike prior MNL bandit literature  (Oh \&
> Iyengar, 2021; Lee \& Oh, 2024), which constructs confidence intervals on each latent utility within the MNL function, our approach places the confidence term outside the MNL structure, as shown and described with Eqs. (3) and (5).
> This modification is essential due to the need to incorporate both UCB and LCB indices in conjunction with the reward terms $r_{n,k}$.
>
> Lastly, we also propose an algorithm which does not depend on the problem-dependent term $\kappa$ without knowledge of it, by utilizing localized Gram matrix (4), which is different from the previous MNL bandit literature [Goyal \& Perivier, 2021; Lee \& Oh,
> 2024] as mentioned in Remark 6.1.  In [Goyal \& Perivier, 2021], the localized term introduces a dependency on non-convex optimization to achieve optimism, whereas our approach utilizes $\widehat{\theta}\_{k,\tau}$ without requiring such complex optimization. Meanwhile, [Lee \& Oh,
> 2024] incorporates all historical information of the estimator into the Gram matrix, which is not well-suited for the G/D-optimal design. In contrast, our method updates the Gram matrix using only the current estimator $\widehat{\theta}_{k,\tau}$, allowing for alignment with a rescaled feature space $\tilde{z}\_{n,k,\tau}$ that is specifically tailored to the combinatorial structure of the problem.
>
> Finally, as mentioned in Remark 6.2, our G/D-optimal design extends beyond existing methods in linear [Lattimore \& Szepesvari, 2020] and generalized linear bandits [Sawarni et al., 2024]. In those settings, the objective is typically based on individual actions, whereas our problem involves structured dependence on assortments, necessitating exploration over a rescaled feature space defined over the entire matching configuration (see Lines 24, 25, 34, and 39 in Algorithm 2).
>
> In summary, this work contributes new algorithmic designs, theoretical analyses, and implementation strategies that are not direct extensions of existing methods.

---

> ### Author Response · Authors · 2025-11-20
>
> - **Comment: Though using a batched algorithm can reduce the computational cost, it still needs to solve the combinatorial optimization problem in some rounds, which requires exponential time complexity and is not feasible in large-scale markets.**
>
>
> **Response:** **Our algorithm does not require enumerating the full matching space $O(K^N)$.**
> As noted in revised Remark 5.4, this quantity reflects the theoretical size of the action space—not the computational
> cost incurred by our method. In Algorithm 1, we do **not** optimize over all matchings.  The active set shrinks rapidly due to elimination. Eliminating suboptimal assignment pairs
> $(n,k)$ greatly reduces the candidate set after only a few batches. Empirically, most assignments are removed
> early, so the theoretical worst-case exponential dependence in $N$ is not encountered in practice. We've added the plot for the size of the active assignment set $\mathcal{M}_\tau$ (Line 14 in Algorithm 1, Line 41 in Algorithm 2) at Figure 4 in Appendix A.13 in our revision. This
> demonstrates that elimination removes the vast majority of assignment candidates early on, greatly
> reducing the effective search space for the rare assortment-optimization steps. Consequently, the
> practical optimization cost of $O(|\mathcal{M}\_\tau|)$ is far smaller than the theoretical worst-case $O(K^N)$ bound.
>
>
> As discussed in Appendix A.13, the scalability of benchmark algorithms (e.g., OFU-MNL$^+$[Lee \& Oh, 2024], UCB-QMB[Kim \& Oh, 2024], TS-QMB[Kim \& Oh, 2024]) is limited by their computational overhead. For example, in Figure 6 (Appendix A.13) with $N = 8$, $K = 5$, and $T = 100{,}000$, we report the number of rounds completed by each algorithm within a 3600-second time limit. Increasing $K$ from 4 to 5 (or $N$ likewise) leads to a substantial increase in runtime for the baselines—resulting in only a few completed rounds (barely visible in the plot). In contrast, our algorithms (B-SMB, B-SMB$^+$) successfully complete all 100,000 rounds within the time budget, demonstrating their computational efficiency and practical scalability.
>
>
> We highlight that **our algorithms perform the optimization only $O(\log\log T)$ times—not every round.**   As is well known in the combinatorial bandit literature  [Chen et al. (2013)], solving exact combinatorial optimization problems—without approximation—inevitably incurs exponential worst-case complexity. This cost is unavoidable unless approximation oracles targeting approximated regret are introduced.
> A key contribution of our work is the batched elimination design: the preference estimates $\hat\theta_{k,\tau}$
> are updated only at the endpoints of exponentially growing epochs. Even if each optimization step is costly,
> its **frequency** is extremely low. As shown in Remark 5.4, the amortized per-round complexity becomes $O(1)$ when $T$ is large enough.
>
> We emphasize that lowering the total computational cost of optimization from $O(K^N T)$ to $O(K^N \log\log T)$ constitutes a major improvement, offering significant benefits in both theoretical analysis and practical implementation. Even for moderately sized $T$, this yields a **substantial reduction in computational cost**. As shown in Figure 5 (Appendix A.13), this efficiency is significant. our algorithms complete 5,000 rounds within **1,000 seconds**, while other benchmarks require over **15,000 seconds** when $N=8$ and $K=4$.
>
>
> **In summary**, the apparent $O(K^N)$ complexity reflects the theoretical size of the action space, not the
> actual computational workload. Our algorithm performs optimization extremely rarely, over rapidly shrinking
> candidate sets. This performance advantage is conclusively demonstrated by both theoretical analysis and experimental results.
>
>
>
>
> - **Comment: Minor: Lines 162–163 contain several notational typos.**
>
> **Response:** We've corrected the typos in our revision.

---

### Official Review · Reviewer_2vzi · 2025-11-07

**Soundness:** 2
**Presentation:** 3
**Contribution:** 3
**Rating:** 4
**Confidence:** 3

**Summary:**

This paper presents a novel approach to solving the stochastic matching bandit problem with batched updates. In this framework, N agents are matched to K arms, where each arm has a preference modelled by a multinomial logit (MNL) model. The acceptance probability for each agent is determined by a softmax function, and the goal is to find an optimal assortment that maximizes the total expected reward. The paper proposes a batched learning algorithm with an amortized computational cost of O(1) and provides a regret bound of Õ(√T), demonstrating both computational and theoretical efficiency. The experimental results show that the proposed algorithm achieves competitive regret while reducing computational overhead compared to traditional methods.

**Strengths:**

Real-World Applicability: The problem is well-motivated by real-world applications such as ride-hailing or job platforms, where stochastic preferences play a crucial role. The paper's proposed framework extends beyond traditional deterministic matching models to a more realistic, stochastic setting.

Batched Approach: The introduction of a batched learning approach to reduce computational costs is a significant contribution, particularly in large-scale matching markets. This approach ensures scalability while maintaining theoretical guarantees.

Empirical Evaluation: The experimental results demonstrate that the batched approach performs efficiently, with computational cost improvements and comparable regret to existing baselines. The paper provides solid evidence for the practical impact of the proposed algorithms.

**Weaknesses:**

Lack of Intuition and Motivation: The paper would benefit from a clearer motivating example in the problem-setting section. Specifically, it would be helpful to provide more intuition about the roles of the different parameters and how they influence the model's performance.

Strong Assumptions: The requirement that the candidate set size |S_k,t| is bounded and that the rewards r_{n,k} are known in advance may not hold in all real-world scenarios. These assumptions simplify the problem but also limit the generalizability of the results. The paper should discuss the potential limitations of these assumptions.

Scalability Concerns: While the batched approach reduces computational cost, there is still a combinatorial optimization problem that requires exponential time complexity in some cases. The paper does not fully address how this might limit scalability in very large markets, with thousands of agents and arms.

**Questions:**

The proposed batched approach eliminates frequent updates, but the cost of solving the combinatorial optimization problem still seems to remain in certain rounds. How does the active set size evolve over time, and are there any safeguards or heuristics in place to ensure that these updates remain computationally feasible for very large values of N and K?

---

> ### Author Response · Authors · 2025-11-20
>
> We thank the reviewers for their thoughtful comments and constructive feedback. We have addressed all points below.
>
> - **Comment: The paper would benefit from a clearer motivating example in the problem-setting section. Specifically, it would be helpful to provide more intuition about the roles of the different parameters and how they influence the model's performance.**
>
>
> **Response:** Although our paper initially referenced ride-hailing dispatch systems as a motivating example (Lines 44–48), we provide additional motivating example  based on an online freelancer marketplace, where the platform matches a set of tasks (agents) to freelancers (arms). In this example:
>
> - $x_n$ denotes the feature vector of task $n$, representing its required skills and known in advance.
>
> - $\theta_k$ is the latent preference vector of freelancer $k$, capturing how well their expertise aligns with different task types.
>
> - $r_{n,k}$ is the revenue obtained when freelancer $k$ successfully completes task $n$.
>
> Under the MNL choice model, the platform recommends tasks to freelancers based on the success probability implied by the utility $x_n^\top \theta_k$, together with the corresponding revenue $r_{n,k}$. This example illustrates how $x_n$, $\theta_k$, and $r_{n,k}$ jointly determine both the likelihood and value of successful matches, shaping the platform’s learning and optimization objectives. We will incorporate this motivating example in the revision.
>
>
> - **Comment:  The requirement that the candidate set size $|S_{k,t}|$ is bounded and that the rewards $r_{n,k}$ are known in advance may not hold in all real-world scenarios. These assumptions simplify the problem but also limit the generalizability of the results. The paper should discuss the potential limitations of these assumptions.**
>
> **Response:**  We emphasize that both assumptions follow the standard MNL bandit framework and do not impose additional restrictions beyond those commonly used in prior work.
>
> **(1) Bounded candidate set size $|S_{k,t}| \le L$.**
> Our model is directly based on the MNL bandit framework. As in prior studies
> (Agrawal et al., 2017b; Oh \& Iyengar, 2019, 2021; Lee \& Oh, 2024),
> we introduce $L$ as the maximum assortment size, with the unconstrained case recovered by simply setting $L = N$.
> In many practical platforms—such as ride-hailing, delivery services, or online marketplaces—each agent is presented with only a *small* subset of options due to user-interface constraints, cognitive load limits, or geo-spatial filtering.
> Thus, $|S_{k,t}| \le L \le N$ often reflects **operational reality** rather than a restrictive modeling assumption.
>
> **(2) Known rewards $r_{n,k}$.**
> Assuming known $r_{n,k}$ is also standard in the MNL bandit literature, where the primary challenge is learning latent user (or arm) preferences rather than estimating deterministic reward parameters.
> In real systems, platforms typically know prices or commissions beforehand—e.g., driver fares, delivery fees, or task payments—before making assignment decisions.
>
> The expected reward of a matching decision is
> $$
> \sum_{k \in [K]} R_k(S_k)
> = \sum_{k \in [K]} \sum_{n \in S_k} r_{n,k} \cdot p(n \mid S_k, \theta_k),
> $$
> where $p(n \mid S_k, \theta_k)$ captures the MNL acceptance probability, and $r_{n,k}$ represents the known payoff upon successful matching.

---

> ### Author Response · Authors · 2025-11-20
>
> - **Comment:  Even with batching, the algorithm still requires solving an exponentially hard combinatorial optimization in some rounds. How does this scale to very large $N,K$ and how does the active set shrink to keep these updates feasible??**
>
>
> **Response:** **Our algorithm does not require enumerating the full matching space $O(K^N)$.**
> As noted in revised Remark 5.4, this quantity reflects the theoretical size of the worst-case action space—not the computational
> cost incurred by our method. In Algorithm 1, we do **not** optimize over all matchings.  The active set shrinks rapidly due to elimination. Eliminating suboptimal assignment pairs
> $(n,k)$ greatly reduces the candidate set after only a few batches. Empirically, most assignments are removed
> early, so the theoretical worst-case exponential dependence in $N$ is not encountered in practice. We've added the plot for the size of the active assignment set $\mathcal{M}_\tau$ (Line 14 in Algorithm 1, Line 41 in Algorithm 2) at Figure 4 in Appendix A.13 in our revision. This
> demonstrates that elimination removes the vast majority of assignment candidates early on, greatly
> reducing the effective search space for the rare assortment-optimization steps. Consequently, the
> practical optimization cost of $O(|\mathcal{M}\_\tau|)$ is far smaller than the theoretical worst-case $O(K^N)$ bound.
>
>
> As discussed in Appendix A.13, the scalability of benchmark algorithms (e.g., OFU-MNL$^+$[Lee \& Oh, 2024], UCB-QMB[Kim \& Oh, 2024], TS-QMB[Kim \& Oh, 2024]) is limited by their computational overhead. For example, in Figure 6 (Appendix A.13) with $N = 8$, $K = 5$, and $T = 100{,}000$, we report the number of rounds completed by each algorithm within a 3600-second time limit. Increasing $K$ from 4 to 5 (or $N$ likewise) leads to a substantial increase in runtime for the baselines—resulting in only a few completed rounds (barely visible in the plot). In contrast, our algorithms (B-SMB, B-SMB$^+$) successfully complete all 100,000 rounds within the time budget, demonstrating their computational efficiency and practical scalability.
>
>
> We highlight that **our algorithms perform the optimization only $O(\log\log T)$ times—not every round.**   As is well known in the combinatorial bandit literature  [Chen et al. (2013)], solving exact combinatorial optimization problems—without approximation—inevitably incurs exponential worst-case complexity. This cost is unavoidable unless approximation oracles targeting approximated regret are introduced.
> A key contribution of our work is the batched elimination design: the preference estimates $\hat\theta_{k,\tau}$
> are updated only at the endpoints of exponentially growing epochs. Even if each optimization step is costly,
> its **frequency** is extremely low. As shown in Remark 5.4, the amortized per-round complexity becomes $O(1)$ when $T$ is large enough.
>
> We emphasize that lowering the total computational cost of optimization from $O(K^N T)$ to $O(K^N \log\log T)$ constitutes a major improvement, offering significant benefits in both theoretical analysis and practical implementation.
> Even for moderately sized $T$, this yields a **substantial reduction in computational cost**.
> As shown in Figure 5 (Appendix A.13), this efficiency is significant. our algorithms complete 5,000 rounds within **1,000 seconds**, while other benchmarks require over **15,000 seconds** when $N=8$ and $K=4$.
>
> **In summary**, the apparent $O(K^N)$ complexity reflects the theoretical size of the action space, not the
> actual computational workload. Our algorithm performs optimization extremely rarely, over rapidly shrinking
> candidate sets. This performance advantage is conclusively demonstrated by both theoretical analysis and experimental results.

---

### Meta-Review · Area_Chair_nNPC · 2026-01-07

**Summary:**

This paper falls within the reject region but is somewhat close to the borderline. One serious concern the reviewers have raised is the scalability of the approach. In the rebuttal, the authors added new experiments and showed in these that scalability is not a major concern in that set of experiments. But that is in my opinion insufficient and the rebuttal is not the right place to add new experiments. In my opinion, the paper ideally needs some theoretical guarantees on the run-time performance under suitable conditions.

**Reviewer Concerns:**

Scalability was a major concern.

**Reviewer Scores:**

The concern wasn't sufficiently addressed and I don't think that the scores would have changed.

---

### Decision · Program_Chairs · 2026-01-26

Reject